# The Global Forest Fire Emissions Prediction System version 1.0

Kerry Anderson[1,2], Jack Chen[1], Peter Englefield[3], Debora Griffin[1], Paul A. Makar[1], Dan Thompson[4]

[1] Air Quality Research Division, Environment and Climate Change Canada, Toronto, Ontario, Canada
[2] Natural Resources Canada (emeritus)
[3] Natural Resources Canada, Edmonton, Alberta, Canada
[4] Natural Resources Canada, Sault St. Marie, Ontario, Canada

*Correspondence to*: Kerry Anderson (kerryanderson@shaw.ca)

**Abstract.** The Global Forest Fire Emissions Prediction System (GFFEPS) is a model that estimates biomass burning in near-real time for global air-quality forecasting. The model uses a bottom-up approach, based on remotely-sensed hotspot locations, and global databases linking burned area per hotspot to ecosystem-type classification at a 1-km resolution. Unlike other global fire emissions models, GFFEPS provides dynamic estimates of fuel consumption, fire behaviour and fire growth based on the Canadian Forest Fire Danger Rating System, plant phenology as calculated from daily global weather and burned are a estimates using near-real-time VIIRS satellite-detected hotspots and historical burned area statistics. Combining forecasts of daily fire weather and hourly meteorological conditions with a global land classification, GFFEPS produces fuel consumption and emission predictions in 3-hour time steps (in contrast to non-dynamic models that use fixed consumption rates and require collection of burned area to make post-burn estimates of emissions). GFFEPS has been designed for use in operational forecasting applications as well as historical simulations for which data are available. A study was conducted showing GFFEPS predictions through a six-year period (2015-2020). Regional annual total smoke emissions, burned area and total fuel consumption per unit area as predicted by GFFEPS were generated to assess model performance over multiple years and regions. The model's fuel consumption per unit area results clearly distinguished regions dominated by grassland (Africa) from those dominated by forests (Boreal regions), and showed high variability in regions affected by El Niño and deforestation. GFFEPS carbon emissions and burned area were then compared to other global wildfire emissions models, including GFAS, GFED4.1s and FINN1.5/2.5. GFFEPS estimated values lower than GFAS/GFED (80%/74%), and values similar to FINN1.5 (97%). This was largely due to the impact of fuel moisture on consumption rates as captured by the dynamic weather modelling. Model evaluation efforts to date are described – an ongoing effort is underway to further validate the model, with further developments and improvements expected in the future.

## 1 Introduction

Biomass burning from wildland fires and agricultural burning is a major source of carbon emissions and greenhouse gases globally. In 2021, estimates of emissions from wildland fire, deforestation and agricultural burning accounted for 2.062 Pg C yr$^{-1}$ (Kaiser and van der Werf, 2023). Compared to the total anthropogenic emissions of 11.0 Pg C yr$^{-1}$ for 2021 (40.2

Pg $CO_2$ yr$^{-1}$, Friedlingstein et al., 2022), biomass burning would equate to 19% of those from anthropogenic emissions; yet much of these emissions (1.75 Pg C yr$^{-1}$ averaged over 2012–2021; Friedlingstein et al., 2022) are recaptured by carbon uptake in the forests (afforestation, reafforestation and forestry) reducing their impact on global concentrations.

Unlike anthropogenic sources, emissions vary greatly from year to year as wildland fire is a dynamic and highly variable event. Estimates show that between 2003 to 2020, biomass burning accounted for 1.781−2.421Pg C yr$^{-1}$ (Kaiser and van der Werf, 2023). Recent events include:

- El Niño events (1997, 2006, 2015, 2019) that triggered extreme emissions from peat fires in Indonesia and southeast Asia (Field et al., 2009; Huijnen et al., 2016; Page and Hooijer, 2016; McPhaden, 2023);
- Australia's unprecedented fire season in 2019/2020, following its hottest, driest year on record (Abrams et al., 2021);
- California's record-breaking number of large fires in 2020, exceeding the previous record in 2018 (Keeley and Syphard, 2021);
- Canada's burned area reaching a record 15.0 Mha for 2023, exceeding the previous record of 6.7 Mha set in 1989 (Kolden et al. 2024).

Wildfires also emit significant quantities of shorter-lived atmospheric pollutants (e.g., nitrogen oxides, volatile organic gases, carbon monoxide, ammonia, particulate matter, heavy metals; cf. Akagi et al., 2011, Urbanski, 2014, Hatch et al., 2017, Wentworth et al., 2018, Hayden et al., 2022, Liu et al., 2023). Global forest fire emissions of particulate matter have been identified as one of the largest sources of atmospheric trace gases and aerosols (Knorr et al., 2012), and their global particulate matter emissions have been found to result in 65.6 million deaths annually (Chen et al., 2021). Respiratory and cardiovascular deaths have been found to be among the chief causes of global wildfire PM mortality (Chen et al., 2021; Barros et al., 2023; Matz et al., 2020) and the impacts on the heart have been found to extend over several days subsequent to wildfire emissions exposure (Barros et al., 2023). Accurate emission estimates of wildfires for smoke forecasting and inventory accounting are therefore of great importance from the standpoint of assessing their impacts on human health and the environment.

Efforts to model wildfire emissions globally have been on-going since the 1970s (Seiler and Crutzen, 1980). Currently, several global wildfire emissions models exist (Pan et al., 2020), such as the Global Fire Analysis System (GFAS, Kaiser et al., 2012), the Global Fire Emissions Database (GFED, Van der Werf et al., 2017), the Fire INventory from NCAR (FINN, Wiedinmyer et al., 2011, Wiedinmyer et al., 2023) and others. Carbon emissions from GFAS (calibrated to partly match GFED emissions) are routinely used to estimate annual carbon emissions from wildland fires for the American Meteorological Society's annual State of the Climate reports (Kaiser and van der Werf, 2023). Wildfire emissions models are also used in conjunction with global chemical transport models such as the Copernicus Atmospheric Monitoring System (CAMS), which uses GFAS emissions to provide concentration estimates that are linked to human health outcomes (Roberts and Wooster, 2021).

Emissions models follow one of two general methodologies: either a top-down or a bottom-up approach to modelling. The top-down approach used by GFAS is centered around satellite-based MODIS active fire products (MOD14/MYD14 Level-2) that provide instantaneous observations of actively burning fires and measurements of fire radiative energy (FRE, the time integral of fire radiative power (FRP); Mota and Wooster, 2018), and biome-specific conversion factors are used to determine combustion rates, which in turn are combined with emission factors to estimate emission rates. The bottom-up approach used by GFED is based on observed burned area (MODIS MCD64A1 mapping algorithm), landscape maps for fuels (MODIS MCD12Q1 land cover type), along with fuel loads, combustions completeness and emission factors per biome typically collected from the literature (van Leeuwen et al., 2014). Both top-down and bottom-up methodologies use satellite sensors for fire detection (MODIS and/or VIIRS) to identify fire locations spatially and temporally (Giglio et al., 2016).

Each approach has its limitations. The satellite-based fire detection used by both top-down and bottom-up methodologies are generally restricted by satellite-overpass times, sensor resolution, observational swath width, heavy smoke and cloud cover. The bottom-up approach is also limited by land-cover and burned-area mapping resolution as well as the accuracy of fuel load mapping and fuel consumption modelling. A methodology to extrapolate the contribution of small, undetected fires – especially important for capturing cropland burning – was presented by Randerson et al. (2012) and included in GFED4.1s. The effectiveness of this small fire boost to emissions has been questioned (Zhang et al., 2018; Gaveau et al., 2021; Ramo et al., 2021) and so GFED5 was developed with scalar corrections based on higher-resolution (non-global) datasets from Landsat and Sentinel-2 (Chen et al., 2023, Hall et al., 2024). Also, Van Wees et al. (2022) incorporated monthly water and temperature stress scalars to model the net primary production (NPP) of stem, leaf and root pools at a 500-m spatial resolution into a simplified version the GFED model, giving fuel loads a temporal variability.

A second limitation of current models is the use of static values for combustion completeness per biome. Fire behaviour is recognized as being dependent on fuels, weather and topography, with weather in the form of temperature, wind, humidity, precipitation, cloud cover and atmospheric stability, being the most variable (Countryman, 1972). The Canadian Forest Fire Danger Rating System (CFFDRS, Stocks et al., 1989) addresses these factors daily in the Canadian Forest Fire Weather Index (FWI) System (Van Wagner, 1987), a system that has seen uptake not only in North America but also New Zealand, Mexico, parts of Europe and southeast Asia (Taylor and Alexander 2006).

A third limitation of many of these models is the timeliness of their products. Certain models depend on remotely-sensed data to build burned areas, accumulated over the course of a month (Giglio et al., 2018, van der Werf et al., 2017, Chen et al., 2023). While such approaches may add precision to predictions, they are of limited benefit to operational air-quality forecasts.

The Canadian Forest Fire Emissions Prediction System (CFFEPS) is a model that predicts smoke emissions used in air quality forecasts for North America based on the CFFDRS. Driven by forecasted hourly meteorology at detected hotspot locations, the model estimates burned area, the hourly chemical components of fire emissions, the plume injection height and vertical distribution of emissions. Predicted smoke emissions are incorporated into Environment and Climate Change Canada's numerical weather and chemical transport model (the Global Environmental Multiscale – Modelling Air quality and

Chemistry model; GEM-MACH). The combined system of emissions, chemistry and transport is referred to as FireWork, an air quality prediction system that indicates how smoke from wildfires is expected to chemically transform and disperse across North America over the next 72 hours. The plume rise component of CFFEPS, as derived from modelled fuel consumption

and parameterized heat flux, has been validated using satellite plume height observations (Griffin et al., 2019). As part of FireWork, the CFFEPS model has been incorporated into ECCC's operational Air Quality Health Index (AQHI) forecasts for North America since 2019 (Chen et al., 2019). More recent work with CFFEPS has allowed its incorporation on-line into a research version of the GEM-MACH two-way coupled air-quality model, in turn accounting for aerosol feedbacks between wildfire emissions and regional weather to be simulated (Makar et al., 2020).

This paper describes the adaptation and extension of the methodologies used in the CFFEPS model to a global domain, as the Global Forest Fire Emissions Prediction System (GFFEPS) – a system that provides spatiotemporal fire emissions estimates for air-quality forecasting based on satellite hotspot retrievals, weather and fire behaviour modelling at the global scale. The motivation for this work was the recognized need in extending FireWork's current North American air-quality forecasting to the global domain, thus improving Canadian forecasts by introducing near real time global simulations of smoke

emissions external to the original North American domain. With increasing fire frequency, size and intensity, smoke can be injected aloft and transported across oceans. For example, smoke from the 2016 Fort McMurray Fire (a.k.a. the Horse River Fire, 2016) affected New York (Wu et al., 2018) and reached as far as the United Kingdom (Vaughan et al., 2018); similarly, the 2023 wildfires in Quebec were observed to transport across the Atlantic impacting air quality of many European communities. A recent study (Makar et al., 2020) showed the impact of forest fire smoke emissions from Eurasia on North

American meteorology and air quality forecasting, highlighting that un-nested continental-only scale air quality models show reduced skill during transoceanic smoke transport events. The impacts of intercontinental pollutant transport have also been demonstrated elsewhere in the literature (e.g., Huang et al., 2017).

This paper sets out to document the data, the methodology and resulting predictions of the GFFEPS model, comparing it to other published global fire emissions models. Chapter 1 provides an introduction with historical content and need for the

120 work. Chapter 2 provides the underlying theory of the model and foundational work. Chapter 3 outlines the external data required to drive the model while Chapter 4 describes the internal calculations and methodology. Results are presented in Chapter 5, discussion in Chapter 6 and conclusion in Chapter 7.

Two appendices are included. Comparisons of GFFEPS to field data is presented in Appendix A, where the GFFEPS methodology of calculating fuel consumption is compared to published field work in Canada, Siberia, Indonesia, African and

125 Brazilian savannah, and Australian eucalypt, and compared to values predicted by GFED. Appendix B provides a sensitivity analysis, examining the impact of landcover data sets, of agricultural burning and small fires and of daily weather.

**2 Theory**

A central problem of predicting smoke emissions is the estimation of the amount of forest fuel consumed by fire, which in turn is injected into the atmosphere. For the bottom-up approach, estimating the amount of fuel consumed involves estimating the total mass of biomass combustion, which is the product of fuel consumed per unit area (kg m$^{-2}$ or t ha$^{-1}$) and burned area (m$^2$, ha or km$^2$). The emissions of specific gas and particle species (collectively, "tracers") is estimated from final effective mass of fuel consumed multiplied by emission factors. Emissions factors are generally pre-determined values derived from measurements as mass of species emitted per unit mass of fuel consumed, typically grams of emitted species per kg of dry fuel consumed (Urbanski, 2014). Fuel consumed per time step is used to calculate heat flux from the combustion process and then used to calculate plume injection height, and parameterize the vertical distribution of the emitted tracers for distribution within a vertical atmospheric column. Species are distributed from the surface to the plume height based on maintaining a constant mixing ratio of smoke to clean air.

**2.1 The Canadian Forest Fire Danger Rating System**

The Canadian Forest Fire Danger Rating System (CFFDRS) has been an important part of forest protection operations in Canada since 1970 (Stocks et al., 1989). The two principal models of the CFFDRS are the Canadian Forest Fire Weather Index (FWI) System, which models fuel moisture and potential fire behaviour in the forest, and the Canadian Forest Fire Behaviour Prediction (FBP) System, which predicts physical fire behaviour in specific vegetative landscapes, referred to as fuel types.

The Canadian Forest Fire Weather Index (FWI) System (Van Wagner 1987) is a set of numerical codes and indices rating relative fire potential. Built on measurements from jack pine forests near Petawawa ON, the system is strictly weather dependent and independent of forest fuel type. Daily and hourly temperature, humidity, wind speed and precipitation are used to estimate the various FWI system indices. The FWI System consists of six components that account for the effects of weather on fuel moisture and potential fire behaviour. The first three components are the fuel moisture codes. These include the Fine Fuel Moisture Code (*FFMC*), the Duff Moisture Code (*DMC*) and the Drought Code (*DC*). These are numeric ratings, or indices, of the moisture content of the litter and other fine fuels, of the loosely compacted organic layers of moderate depth, and of the deep, compact organic layers respectively. Their values rise as moisture content decreases. The remaining three components – the Initial Spread Index (*ISI*), the Buildup Index (*BUI*) and the Fire Weather Index (*FWI*) – are fire indices. These indices represent respectively the rate of fire spread, the fuel available for combustion, and the frontal fire intensity; their values rise as the fire danger increases.

The FWI system is internationally-recognized and is used by several countries including Canada, certain US states, Mexico, ASEAN nations, New Zealand and a number of European nations (Taylor and Alexander, 2006). Daily maps in near-real time are routinely generated and displayed on the Canadian Wildland Fire Information System (https://cwfis.cfs.nrcan.gc.ca/home, last accessed 2024-05-28; Lee et al., 2002), the European Fire Information System

The Canadian Forest Fire Behaviour Predictions (FBP) System (Forestry Canada Fire Danger Group 1992; Wotton et al., 2009) is an extension of the FWI system. It captures the physical measures of fire behaviour within certain Canadian landscapes. The FBP system consists of a series of empirical models that predict fire behaviour conditions for 18 common fuel types in Canada (see Table 1). Using daily and hourly weather values and indices from the FWI system as inputs, the FBP

system predicts for the prescribed fuel types in Canada measurable physical variables including the forward rate of spread ($ROS$) in m min$^{-1}$, head fire intensity ($HFI$) in kW m$^{-1}$, surface, crown and total fuel consumptions ($SFC$, $CFC$, $TFC$) in kg m$^{-2}$ (where $TFC = SFC + CFC$) and crown fraction burned ($CFB$) as a fraction or percentage. It is worth noting that the FBP system was designed with a focus on the most hazardous fuels in Canada and under high fire behaviour conditions. Challenges will be present adapting FBP to broader, global landscapes and addressed in the methodology.

The fuel consumption values ($SFC$, $CFC$, $TFC$) predicted by FBP system are central to the wildfire emissions predictions in CFFEPS and in GFFEPS. It is assumed that the fuel consumed by the fire translates directly to emissions, and that components of tracer emissions, that in turn are injected into the atmosphere, directly contribute to wildfire smoke (i.e., one tonne of fuel consumed becomes one tonne of smoke emissions, including ash and soot). In forecast model applications, FWI values and FBP predictions can be calculated daily and hourly with outputs from numerical weather models, and tracer

emissions can be calculated in near-real time for fire locations as they are identified.

**Table 1. Canadian Forest Fire Behavior Prediction (FBP) System fuel types.**

| Group/Identifier | Descriptive name |
| --- | --- |
| *Coniferous* | |
| C-1 | Spruce-lichen woodland |
| C-2 | Boreal spruce |
| C-3 | Mature jack or lodgepole pine |
| C-4 | Immature jack or lodgepole pine |
| C-5 | Red and white pine |
| C-6 | Conifer plantation |
| C-7 | Ponderosa pine - Douglas-fir |
| *Deciduous* | |
| D-1 | Leafless aspen |
| D-2 | Aspen - green |
| *Mixedwood* | |
| M-1 | Boreal mixedwood - leafless |

| M-2 | Boreal mixedwood - green |
| M-3 | Dead balsam fire mixedwood - leafless |
| M-4 | Dead balsam fire mixedwood - green |
| *Slash* | |
| S-1 | Jack or lodgepole pine slash |
| S-2 | White spruce-balsam slash |
| S-3 | Coastal cedar - hemlock - Douglas-fir slash |
| *Open* | |
| O-1a | Matted grass |
| O-1b | Standing grass |

## 2.2 The Canadian Forest Fire Emissions Prediction System

GFFEPS follows the same methodology as its predecessor CFFEPS, which has been documented in recent publications (Makar et al., 2020, Chen et al., 2019).  CFFEPS uses fire weather conditions modelled by the FWI system and fire behaviour by the FBP system to determine fuel consumed per unit area per time step (1 hour in CFFEPS; 3 hours in GFFEPS).  Burned area (per day) in CFFEPS is based on annual ecoregion and vegetation-specific burned-area climatology normalized by the number of satellite-detected hotspots (Chen et al., 2019).  For CFFEPS, values of historical average burned area per hotspot

(i.e., burn-area climatology for 10 years from 2012-2021) were calculated by each fuel type and ecoregion for each province/territory, by relating recorded hotspots to annual burned area statistics as reported by provincial and territorial agencies (the National Burned Area Composite (NBAC): https://cwfis.cfs.nrcan.gc.ca/datamart/download/nbac, last accessed 2024-05-28). The process followed by CFFEPS/GFFEPS and unit convention used in the paper is illustrated in Fig. 1.

        The application of CFFEPS calculations is conducted on each satellite-detected hotspot.  Fire weather conditions are

interpolated to the hotspot location and fire behaviour is calculated based on the fuel type sampled at the hotspot location. Burned area per day, based on the burned-area climatology, is used and persistence of burned-area rate is assumed for the ensuing 24-hour forecast period.  Fuel consumption per time step is calculated using a diurnal pattern of area growth per hour.

        While CFFEPS has been demonstrated to be an excellent means of near-real-time wildfire emission estimate for air-quality forecast application within a North American context, several critical issues arise when expanding its utility to the

global scale.  These include expanding the FWI calculations to a global domain, establishing a global fuels map compatible with the FBP system, and determining the most relevant, compatible fuel type and fuel consumption equations within the CFFDRS framework to represent global landscapes.

**Figure 1. Structure of the Canadian Forest Fire Emissions Prediction System (CFFEPS), used by GFFEPS. Historical input data (parallelograms) are shown in blue. Current input data (parallelograms) and operational calculations (rectangles) are shown in green. Predictive models (rectangles) are shown in red (CFFDRS), purple (CFFEPS) and orange (burned area mapping). Units reflect those used in the text. The plume rise module and the emissions vertical distribution are not discussed in this paper.**

## 2.3 Global Models

There are several published global fire-emissions models (Pan et al., 2020). For this study we included

- Global Fire Assessment System (GFAS1.2, Kaiser et al., 2012),

- Global Fire Emissions Database (GFED4.1s, van der Werf et al., 2017),
- Fire Inventory from NCAR (FINN 1.5/2.5, Wiedinmyer et al., 2011, Wiedinmyer et al., 2023).

The GFED and FINN models use the bottom-up approach and estimate effective fuel consumption rate $E$ (mass of fuel consumed per unit area per time – kg m$^{-2}$ d$^{-1}$ or tonnes ha$^{-1}$ d$^{-1}$) based on the equation of Seiler and Crutzen (1980):

$$E = BA \times FL \times CC \times EF \tag{1}$$

where burned area ($BA$) is a measure of the spatial extent of fire activity over a period of time (ha d$^{-1}$), fuel load ($FL$) is the biomass of combustible fuels (t ha$^{-1}$) on the landscape, and combustion completeness ($CC$) is the percentage of the total available biomass consumed by fire (%). For final emission rates related to chemical component emissions such as CO, CH$_4$ and particulate matter, the effective fuel consumed ($BA$ x $FL$ x $CC$) is multiplied by species-specific emission factors ($EF$) in g of emissions per kg of dry fuel consumed. These factors are typically derived from field or laboratory measurements and can be specific to fuel type and burn conditions as measured by combustion efficiency (Urbanski 2014, Chen et al., 2019).

Expanding CFFEPS into the global domain, GFFEPS follows a similar methodology to GFED and FINN. An adjusted version of Eq. (1) is used in GFFEPS as the $FL$ x $CC$ term is replaced by the total fuel consumption ($TFC$) of the Canadian Forest Fire Behaviour Prediction (FBP) system. In doing so, daily fire behaviour is captured by using the FWI and FBP systems; we replace the static combustion completeness used by standard bottom-up models such as GFED and FINN with more dynamic parameterizations contained within the CFFDRS framework.

The global regions commonly used in global fire-emissions analyses are shown in Fig. 2.

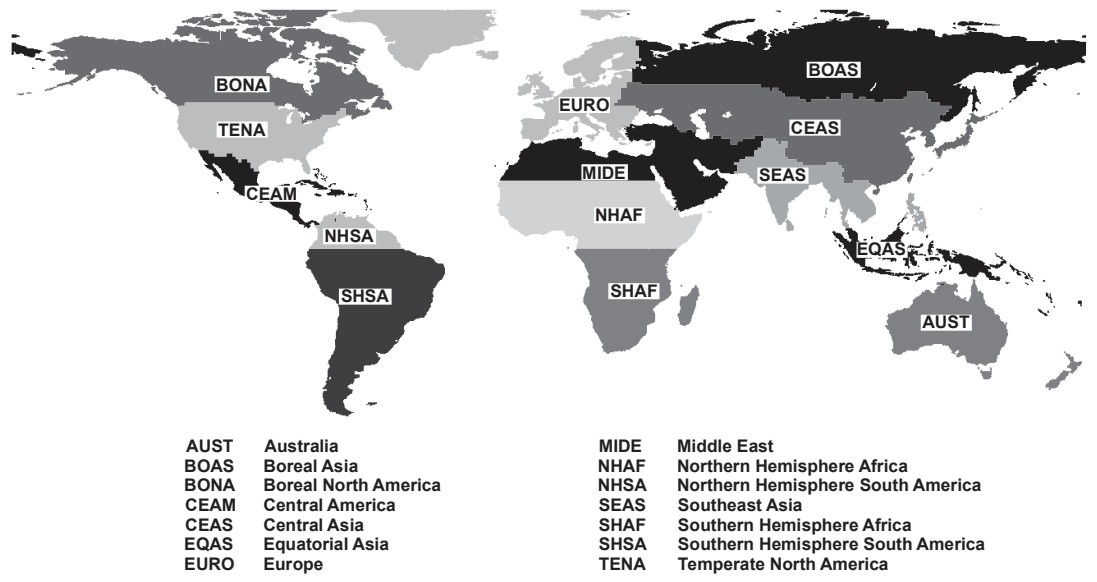

| AUST | Australia | MIDE | Middle East |
|------|-----------|------|-------------|
| BOAS | Boreal Asia | NHAF | Northern Hemisphere Africa |
| BONA | Boreal North America | NHSA | Northern Hemisphere South America |
| CEAM | Central America | SEAS | Southeast Asia |
| CEAS | Central Asia | SHAF | Southern Hemisphere Africa |
| EQAS | Equatorial Asia | SHSA | Southern Hemisphere South America |
| EURO | Europe | TENA | Temperate North America |

**Figure 2. Global regions and regional abbreviations used in this study following those defined in Giglio et al. (2006).**

## 3 Data

To calculate global fire emissions, critical input data are needed. These include global land classifications, satellite-detected fire locations (a.k.a. hotspots), daily global weather, plant phenology and agricultural burning statistics. These data sources are external to the daily operational running of GFFEPS (Fig. 1) and require preprocessing.

### 3.1 Land Classification

A land classification system is required to link tree species and landscapes at fire locations to fire behaviour, fuel consumption and emissions as predicted by FBP. Global climate models use a variety of vegetation classification systems. The Global Land Cover 2000 Project by European Commission (GLC2000, Bartholome and Belward, 2005) is such a product (Fig. 3 and Table 2) and was adopted in the development of this initial version of GFFEPS. Developed in collaboration with a network of partners around the world, the general objective of GLC2000 was to provide a harmonized land-cover database

over the whole globe for the year 2000. The year 2000 was selected as a reference year for environmental assessment in relation to various activities. While other land use databases are available (e.g., MODIS, etc.), GLC2000 was selected for its global spatial resolution with 1-km at the equator, for its level of detail in the number of land use types, for the national-level ground-truthing data used in its construction, for ease of data usage, accessibility and for consistency throughout our analysis. While acknowledging the 25-year age of the GLC2000 dataset, we note that land use changes occurring subsequent to the year

2000 are unlikely to result in a significant change in biomass burning emissions in an on-line model such as GFFEPS. For example, vegetation classes rarely change (e.g. deciduous forests rarely change into coniferous) and most land use changes, whether they were a result of disturbance (fires, deforestation) or urbanization, would result in landscapes less fire prone and this in turn would be reflected by a reduced number of hotspots. In turn, reduced hotspot detection would result in less smoke emissions, capturing the impact of the land use change. However, we note that the same methodology developed here using

GLC2000 may be used with other land-use databases, including time-varying databases such as those provided by satellite retrievals (e.g. MODIS). We present comparisons between GFFEPS configured for MODIS land use data versus GLC2000 in Appendix B.1.

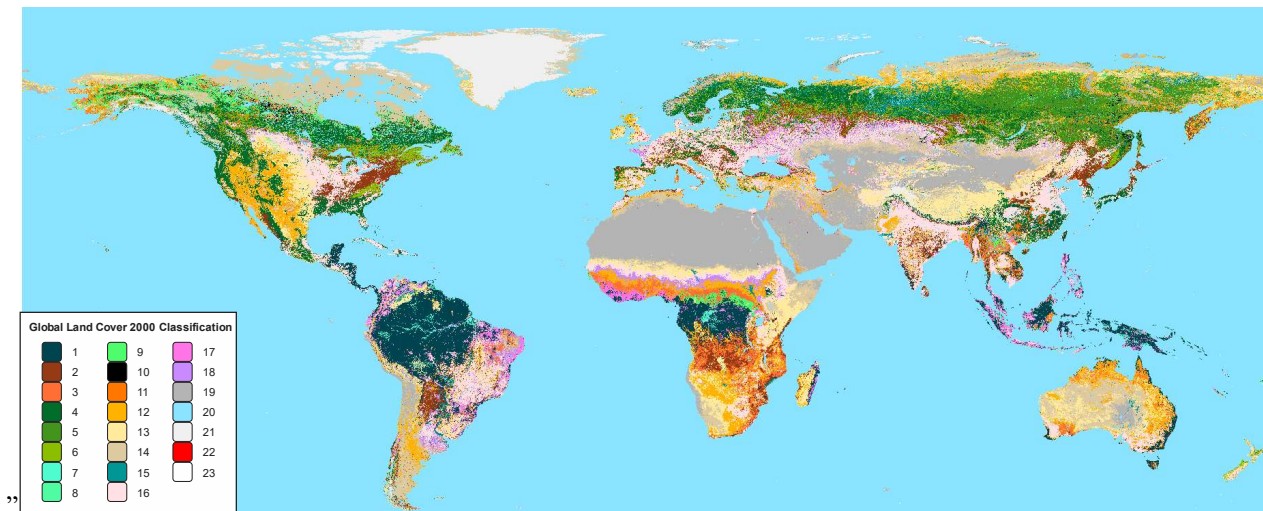

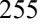

**Figure 3. Global Land Cover 2000 classification. See Table 2 for land classification descriptions for numbered values appearing in the legend.**

**Table 2. Global Land Classification 2000 (GLC 2000) codes and descriptions.**

| GCL 2000 | Description |
| --- | --- |
| 1 | Tree Cover, broadleaved, evergreen |
| 2 | Tree Cover, broadleaved, deciduous, closed |
| 3 | Tree Cover, broadleaved, deciduous, open |
| 4 | Tree Cover, needle-leaved, evergreen |
| 5 | Tree Cover, needle-leaved, deciduous |
| 6 | Tree Cover, mixed leaf type |
| 7 | Tree Cover, regularly flooded, freshwater (& brackish) |
| 8 | Tree Cover, regularly flooded, saline water |
| 9 | Mosaic: Tree cover / Other natural vegetation |
| 10 | Tree Cover, burnt |
| 11 | Shrub Cover, closed-open, evergreen |
| 12 | Shrub Cover, closed-open, deciduous |
| 13 | Herbaceous Cover, closed-open |
| 14 | Sparse Herbaceous or sparse Shrub Cover |
| 15 | Regularly flooded Shrub and/or Herbaceous Cover |
| 16 | Cultivated and managed areas |
| 17 | Mosaic: Cropland / Tree Cover / Other natural vegetation |

| | |
|---|---|
| 18 | Mosaic: Cropland / Shrub or Grass Cover |
| 19 | Bare Areas |
| 20 | Water Bodies (natural & artificial) |
| 21 | Snow and Ice (natural & artificial) |
| 22 | Artificial surfaces and associated areas |
| 23 | No data |

A review of regional descriptions of each land classification provided a means to assign FBP fuel types to all GLC2000 classifications present in each region based on expert opinions. The assigned fuel for specific classifications may vary between regions and confidence in assignments varies.  The resulting mapped FBP fuel types are shown in Fig. 4.

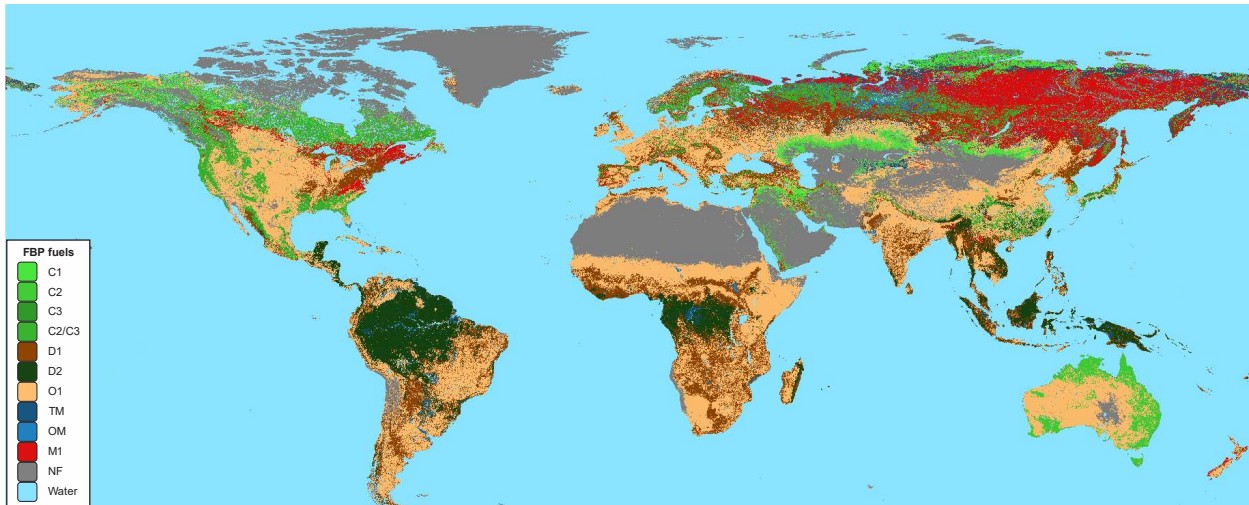

**Figure 4. Canadian Forest Fire Behaviour Prediction (FBP) System fuel types as assigned from the Global Land Classification 2000.  See Table 1 for descriptions of FBP fuel types appearing in the legend. TM and OM, not listed in Table 1, represent treed and open muskeg.**

We note that both the land use classification (GLC2000) and the region classification are used in determining the fuel assignment.  For example, peatlands in the tropics differ from those in northern latitudes (see 4.3.1 Peat Fires and Appendix section A.3); coniferous forests differ between North America, Eurasia and Australia (see Appendices A.1, A.3, section 4.3.2 and Appendix A.5).

Certain GLC2000 land cover classifications, such as peat lands (described as "regularly flooded"), do not have any corresponding fuel types in FBP.  Methods for representing these are discussed in the supplemental information section

described in Appendix A. Also, GLC2000 land classifications 16, 17 and 18 were assigned to agriculture regions and treated separately (see 3.5 Agricultural Burning). The resulting map (Fig. 5) presents the supplemental fuel types, which take precedence over the FBP fuel types where they occur.

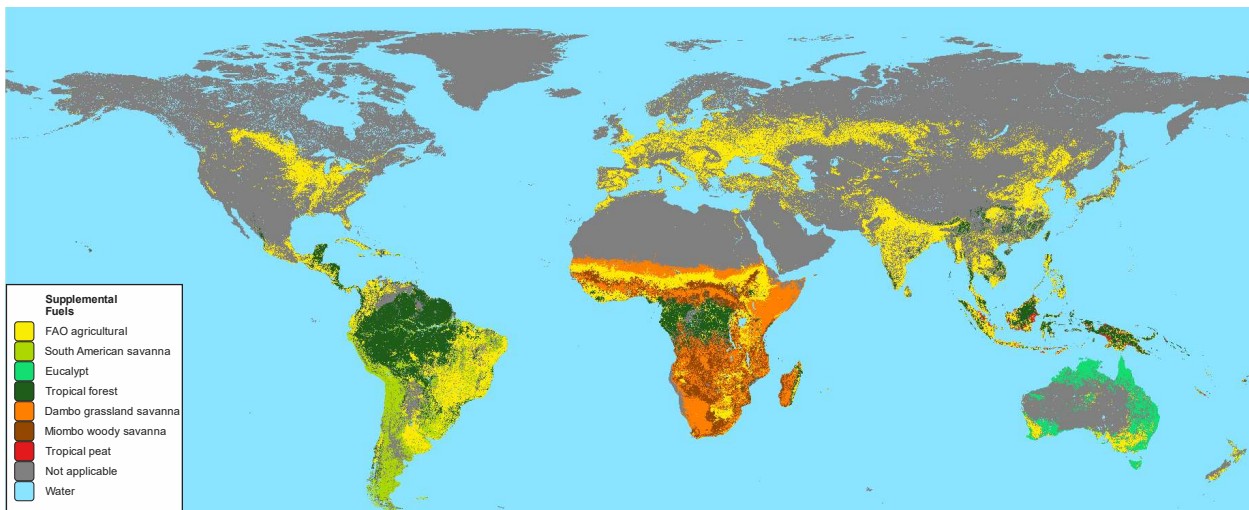

**Figure 5. Supplemental fuels as described in Chapter 3.5 and in Appendix A. Note that these fuels take precedence over the FBP fuel types presented in Fig. 4.**

In addition to the land-cover classification, GFFEPS requires surface fuel load, forest floor depth and bulk density data (https://www.ciffc.ca/publications/glossary, last accessed 2024-05-28) for FBP system calculations (derivation of these data

are discussed later in the methodology section). As part of the GFED3.x and 4.x wildfire emissions model, van Leeuwen et al. (2014) and van der Werf et al. (2017) collected detailed fuel load and consumption data from 201 and 591 sites respectively through literature reviews. However, the biomes used by GFED do not directly correspond to land classifications in the GLC2000; therefore, the biomes of these sites were matched to GLC2000 land classifications, with varying degrees of confidence depending on the number of sites within biome and the consistency of correspondence between biome and

GLC2000 land use classifications. Given the matches, fuel load values were then applied (see Appendix A).

**3.2 Satellite Hotspots**

GFFEPS requires the times and locations of active fires. Similar to most global fire emission models, these are obtained in the form of hotspots identified from infrared satellite imagery. GFFEPS uses hotspots detected by the Visible Infrared Imaging Radiometer Suite (VIIRS) sensor and obtained from the Fire Information for Resource Management System (FIRMS)

provided by NASA and the U.S. Forest Service. The VIIRS sensor was first launched on board the Suomi National Polar-Orbiting Partnership (S-NPP) satellite in 2011 (also onboard NOAA-20, NOAA-21 satellites since 2017 and 2022,

respectively) and provides coverage of every location on the globe at least twice daily, with higher frequency at high latitudes. Not all fires are detected; some are too small, some are short-lived and burn between satellite overpasses, and some burn under thick cloud cover or heavy smoke that render them invisible. In spite of these limitations, we selected VIIRS data because it is sub-daily, global, readily available, higher resolution than alternative sensors, available in near-real time, and expected to continue well into the 2030s.

Hotspot data from other sensors is available from FIRMS as well, including the Moderate Resolution Imaging Spectroradiometer (MODIS) and the Advanced Baseline Imager (ABI). VIIRS was selected because of its higher resolution (375m, compared with 1 km for MODIS and 2+ km, depending on latitude, for ABI) but in the future, data from other sensors could be incorporated as inputs to GFFEPS. Unlike other top-down approaches that use quantitative FRE/FRP to parameterize fuel consumption, GFFEPS does not use satellite sensor quantitative measurement; instead, only high-resolution hotspot location and ignition timing is required. This allows potential future expansion of GFFEPS to use other remote sensing data, including radiometric measurements such as Interferometric Synthetic Aperture Radar (InSAR) with the advantage of detecting fire through cloud and smoke at high spatial resolution (Ban et al., 2020; Goodenough et al., 2014).

## 3.3 Global Weather

Global weather conditions, essential in predicting FWI, are calculated with ECCC's Global Environmental Multiscale (GEM) model. GEM is the core numerical weather prediction (NWP) model of ECCC's operational weather prediction services. Global scale GEM currently provides gridded meteorological conditions at 15-km resolution every 3-hour time steps to calculate fire behaviour conditions and smoke emissions in GFFEPS. Extracted surface variables include windspeed, relative humidity, temperature and 24-hour accumulated precipitation to calculate FWI; additional variables including vapour pressure deficit, solar day length etc. were extracted for FBP.

Daily FWI is central to the GFFEPS system: noon values are used to calculate the FWI values, which are then used to predict fire behaviour and resulting smoke emissions. For example, the Buildup Index (*BUI*) is one of the FWI indices and a principal driver in calculating fuel consumption in the FBP system. Environment and Climate Change Canada now has FWI calculations incorporated as part of model product processes in regional weather forecasts. Figure 6 shows a sample global map of daily value of the Buildup Index (*BUI*) predicted for the reported hotspots for September 1, 2019.

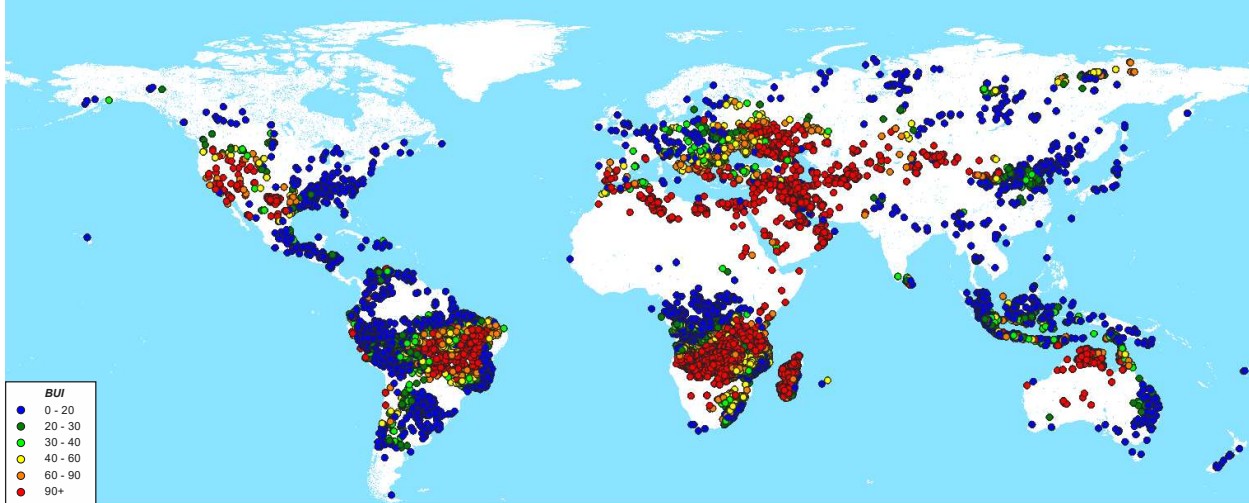

**Figure 6. Buildup Index (*BUI*) for September 1, 2019 as interpolated to the 63,566 hotspot locations observed on that date. The *BUI*, a principal driver in calculating fuel consumption in the FBP system, is calculated using meteorological data from Environment and Climate Change Canada's Global Environmental Multiscale (GEM) model.**

### 3.4 Plant Phenology

Seasonal cycles in plant characteristics, known as phenologies, significantly influence the timing and quantity of live vegetative growth. These phenological changes influence overall fuel moisture levels (considering both live and dead fuels) and consequently impact fire behaviour. In temperate and boreal ecosystems during spring, deciduous trees in the temperate zones emerge from winter dormancy, leafing out through the growing season before shedding leaves as they return to dormancy in autumn (Alexander, 2010a; Quintillio et al., 1991). Similarly, grasses green-up in the spring and reach maturity, then desiccate in the summer heat, either dying off or re-entering dormancy in warm temperate, Mediterranean, and tropical climates with a strong wet-dry seasonality such as Australia (Cheney and Sullivan, 2008). Grasses as well as trees in cool temperate and boreal regions are controlled by a combination of photoperiod and freezing temperatures initiating grass curing (Jolly et al., 2005). Coniferous crowns undergo an important seasonal dip in foliar moisture during the spring as needles transpire while the roots are still frozen that has impacts on the initiation of crown fire (Alexander, 2010a). These effects have an important impact on smoke emissions and hence have been addressed within GFFEPS.

### 3.4.1 Growing Season Index

Deciduous leaf-out (greenness), and grass dormancy (curing), are important factors in fire behaviour. The FWI system does not have a built-in method to predict these phenologies; instead, the FBP system relies on users to provide both grass

curing fraction as well as the leaf-out status of deciduous vegetation based on physical observations. To address this in GFFEPS, the Growing Season Index (*GSI*) by Jolly et al. (2005) was used as a surrogate to capture the seasonal dynamics of
345 deciduous leaf-out. This model uses simple threshold functions (zero below a minimum value, one above a maximum value and a linear relation from 0 and 1 between the minimum and maximum values; reversed in the case of vapour pressure deficit) based on the following three observable parameters:

- minimum temperature (linear response range between -2°C and 5°C),
- vapour pressure deficit (linear response range between 900 Pa and 4100 Pa),
- hours of daylight (linear response range between 10 hrs and 11 hrs).

A daily *GSI* is calculated as the product of these three output values; afterwards, a moving average over the previous 21-days' *GSI* values is applied to reduce abrupt daily variability thus better mimicking plant response (Jolly et al., 2005). Figure 7 shows a sample global map of daily value of the *GSI* predicted for the reported hotspots for September 1, 2019.

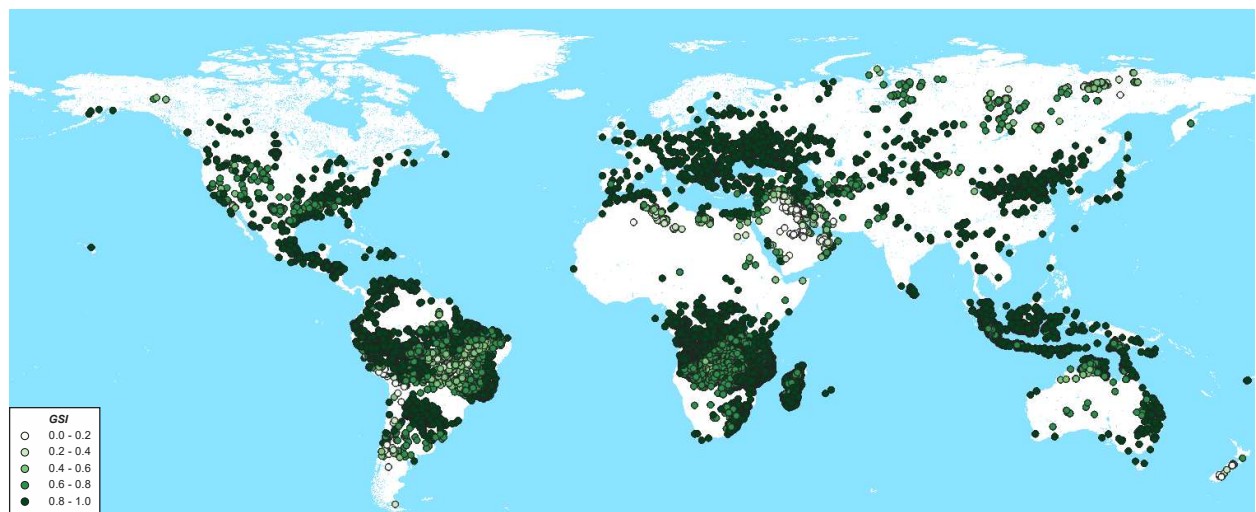

**Figure 7. Growing Season Index (*GSI*) for September 1, 2019 as interpolated to the 63,566 hotspot locations observed on that date. The *GSI* provides a method to estimate the greenness of deciduous forests and degree of grass curing, both important factors in fuel consumption. The 21-day average *GSI* is calculated using meteorological data from Environment and Climate Change Canada's Global Environmental Multiscale (GEM)**
**model.**

The *GSI* is a surrogate for the greenness of the Normalized Difference Vegetation Index (NDVI; Pettorelli, 2013), typically measured via remote-sensing approaches. GSI provides a continuous calculation of the greenness value, both spatially and temporally, easy for forecast application, while NDVI must be stitched and gap-filled from satellite data that is
365 significantly more complicated and laborious. The GSI is also currently used by the US Forest Service as part of the National

Fire Danger Rating System (https://www.firelab.org/project/national-fire-danger-rating-system – last accessed 2024-05-28). Nevertheless, as the operational system further develops, observed NDVI could one day be timely assimilated and replace the GSI calculations for grass curing and deciduous green-up.

### 3.4.2 Foliar Moisture Content

The Foliar Moisture Content ($FMC$) is another phenology required by the FBP system, and is defined as the moisture content of live needles in a conifer tree (Alexander, 2010a). On average, the $FMC$ of coniferous trees is 120% during the fire season in Canada, but in the spring as the ground thaws, the $FMC$ dips to 85%, reflecting a decrease as the foliage transpires while the roots are still frozen. This spring dip of $FMC$ increases the likelihood of crown fire initiation in conifer trees (fires rarely crown in deciduous trees); in turn, this affects crown fuel consumption ($CFC$) and thus emissions into the atmosphere.

The Julian date of the minimum $FMC$ is denoted as $D_o$.

The CFFDRS has a means of calculating the $FMC$, yet this is only valid in North America. To expand this to a global domain, a new set of equations was developed to calculate $FMC$ in Eurasia following the principles of the original approach. As the spring dip in the $FMC$ value is based on the assumption that the ground is frozen in the winter, $FMC$ calculations are limited to northern latitudes where $D_o$ exceeds 90 (i.e., minimal $FMC$ occurs on or after March 31); elsewhere, the default

$FMC$ value of 120% is used in the northern hemisphere In the southern hemisphere, a default $FMC$ value of 147% is used year-round, as used in New Zealand (Pearce et al., 2008; Alexander, 2010a), where coniferous trees rarely reach freezing conditions .

The $FMC$ used in the CFFDRS was based on observations from eight stations in Canada (Forestry Canada Fire Danger Group 1992). The assumption was that dates of minimum $FMC$, $D_o$, followed climatological isotherms along with elevation

adjustments of 0.026 days m$^{-1}$. Following the same rationale, climatological maps of isotherms for March, April and May were collected for Eurasia. Assuming a parabolic shape for the April 0°C isotherm, the resulting equation for the latitude-longitude contour of the date of the minimum FMC becomes

$$LATZ = (65 - 47)/(30 - 120)^2 \times (LON - 120)^2 + 47 = 0.0022 \times (LON - 120)^2 + 47. \qquad (2)$$

The resulting curve corresponds to $D_o$, the contour of the Julian day of minimum $FMC$, chosen as 151. In Canada, $D_o$

drops off at approximately 2 days per degree latitude so in Asia,

$$D_o = 151 + 2.286 * (LAT - LATZ). \qquad (3)$$

Using 146°W and 12°W as the lines dividing Eurasia from North America, $D_o$ can be calculated for the northern hemisphere (not required for the southern hemisphere as noted above). Figure 8 illustrates the global map of the month when the day of minimum $FMC$ ($D_o$) occurs.

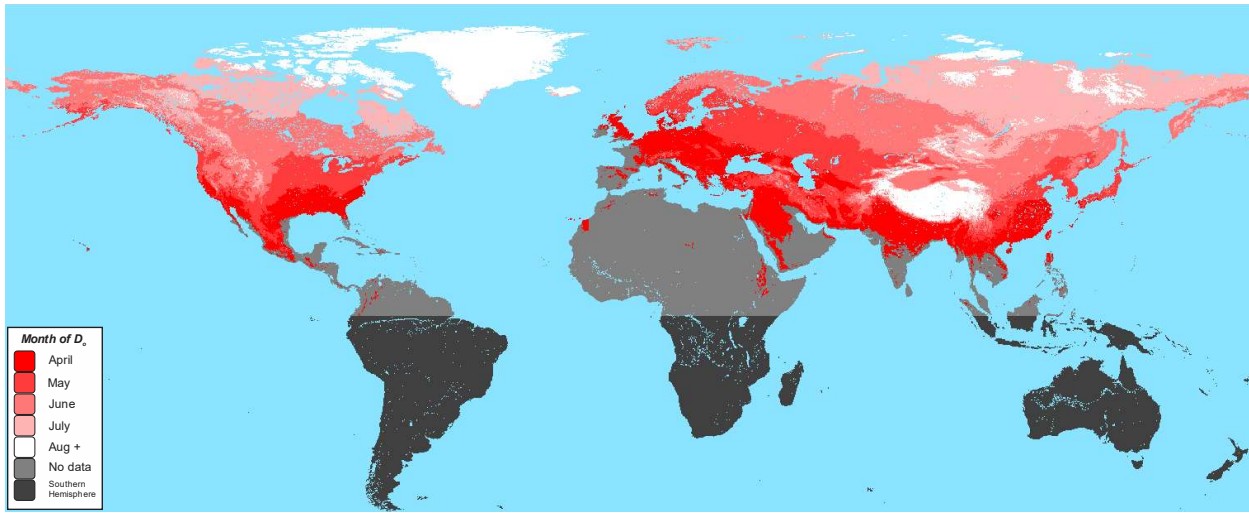

**Figure 8. Month of occurrence of the day of minimum Foliar Moisture Content ($D_o$), when the *FMC* dips to 85% from 120%. The no data zone and the southern hemisphere are assumed to have no spring dip and thus *FMC* values are set to a constant 120% and 147% respectively.**

**3.5 Agricultural Burning**

Agricultural burning is governed by a variety of processes that differ from those encompassed by Eq. (1).  These must be addressed separately in order to properly model the contribution of the agricultural sector to global fire emissions. Croplands cover 12% of the ice-free land surface.  It is estimated that residue burning accounts for 5% of global emissions (Cassou, 2018; Bond et al., 2013). Depending on the time of year, farmers may burn the residue after harvest. There are a wide

variety of crops but the three principal crops whose post-harvest residues are typically burned are maize, rice and wheat. Pouliot et al. (2017) provided fuel loads for these and several other crop residues in the USA. These are as much as twice the default fuel load of the FBP system's standing grass FBP O-1 (0.35 kg m$^{-2}$), indicating the need to differentiate agricultural burning from grass fuel.

In order to include these effects into GFFEPS, agricultural burning calculations were applied to GLC2000 land

classifications 16, 17 and 18 (see Fig. 3 and Table 2).  Streets et al. (2003) presented the following equation to estimate the total mass of crop residue burned in the field ($R$) as

$$R = P \times N \times D \times B \times F \tag{4}$$

where

- $P$ = crop production;

- $N$ = crop-specific production-to-residue ratio;

- $D$ = dry-matter-to-crop ratio;

- $B$ = the percentage of dry matter residues that are burned in the field;
- $F$ = the crop-specific burn efficiency ratio.

The first four terms in Eq. (4) are provided in data collected by the United Nations' Food and Agriculture Organization (FAO). The FAO collects global agricultural production, presenting national amounts on their FAOSTAT page (https://www.fao.org/faostat/en/#data/GB; last accessed 2024-05-28). This page provides past crop residue burning values by nation per year. This value, $E_{FAO}$, captures dry weight of crop production ($P$), crop-specific production-to-residue ratio ($N$), dry-matter-to-crop ratio ($D$) and percentage of dry matter residues that are burned in the field ($B$) as a single value, simplifying the application of Eq. (4) to

$$R = (P \times N \times D \times B) \times F = E_{FAO} \times F. \qquad (5)$$

Values for burning efficiencies ($F$) for specific crops were then taken from Turn et al. (1997).

Annual statistics of biomass burned (dry matter) from the FAOSTAT were compiled for each country for the years 2012, 2015, 2018, 2019 and 2020. Similarly, the number of VIIRS hotspots occurring within the GLC2000 land classifications assigned to agricultural burning were counted for each country for the same years. National statistics were then grouped according to regions outlined by Giglio et al. (2006, Fig. 2). From this, historical average biomass burned per hotspot was calculated for each region's agricultural zone, which were then used in subsequent GFFEPS estimates of emissions from agricultural burning. Emissions per time step for agricultural fires was based on a diurnal curve for agricultural burning approximating a Gaussian curve centered on 15:00 LST (Eyth et al., 2022, McCarty et al., 2009).

We note that this approach is a significant departure from the method used in other global fire emission models. Agricultural burning is typically conducted at small scales and short durations and, as a result, are difficult to detect with satellite-based remote sensing. GFED4.1s simulates these undetected agricultural fire emissions by extrapolating the agriculture areas burned detected by remote sensing (Randerson et al., 2012, van der Werf et al., 2017). Hall et al. (2024) furthers this by calculating crop-specific burned area conversion factors based on detailed cropland mapping. The FAO statistics approach used by GFFEPS avoids the small fire detection issue associated with agricultural burning by using country-specific report data from FAO to capture all biomass burned, including small fires, in agricultural landscapes. The approach does assume small fires in other, non-agricultural, landscapes are inconsequential, which we see as acceptable. This is certainly the case in Canada, where the National Forestry Database (https://cwfis.cfs.nrcan.gc.ca/ha/nfdb, last accessed 2024-05-28) indicates that between 1980-2021, fires less than 1 ha, which constituted 73% of fires, account for only 0.03% of the burned area nationally; that fires less than 10 ha, which account constituted 87% of fires, account for only 0.18% of the burned area.

In principle, the data and methodology outlined for GFFEPS captures all biomass burned in croplands, regardless of fire size, statistically accounted for and reported by individual countries. With that said, the Tier 1 methodology used by the FAO to determine this value may not be rigorous in developing countries (Tubiello et al., 2014) or where illegal agricultural burning is widespread (Hall et al., 2021); nevertheless, its application in GFFEPS seemed a direct and practical solution for real-time smoke forecasting while addressing the small fire issue specific to agriculture activities.

  **4 Methodology**

GFFEPS follows the same methodology as CFFEPS (Chen et al. 2019) but uses additional data sets and alterations described in this section. Likewise, GFFEPS follows Eq. (1) and the section titles described under the methodology follow each of the Seiler and Crutzen equation variables: burned area ($BA$), fuel load ($FL$), combustion completeness ($CC$) and emission factors ($EF$). GFFEPS is then run daily, using observed satellite-detected hotspots and historical average burned area per hotspot to calculate burned area ($BA$), and interpolated fire weather and fuel characteristics at each hotspot to determine fuel consumption ($FL \times CC$) and then daily smoke emissions.

Global fire emissions produced by GFFEPS were first examined for interannual and interregional variability, then a multi-model comparison was conducted between GFFEPS and four other published wildfire emissions models to test its general performance.

**4.1 Burned Area**

GFFEPS requires an estimate of burned area ($BA$) for each hotspot. Historical data for 2012-2019 was obtained from the MODIS burned area product (Giglio et al., 2018), which provides gridded monthly burned area for the globe. Total burned area and VIIRS hotspot count were determined for each combination of region (Fig. 2), month, and land-cover type (Fig. 3 and Table 2).

Dividing total burned area by number of hotspots provides a simple estimate of historical average burned area per hotspot. However, in some cases hotspots were found in the same location two or more days in a row, or within a short period of time. This could represent a pixel partially or incompletely burned, as would be the case if a fire was moving slowly, or burning in episodes separated by smouldering. In other cases, hotspots were occurring in a location repeatedly for several months or even years, indicating a non-fire heat source, usually an industrial facility. Whatever the underlying reason, it was decided that these hotspots should not be assigned the same burned area as lone or isolated hotspots.

For each hotspot, the number of times burned, $T$, was calculated as the number of hotspots that occurred in the last 6 months within the VIIRS I-band pixel (375m) centered on that hotspot. As the current hotspot was included in the count, $T$ was always at least 1. The 6-month time frame reflects our assessment that a completely burned vegetation is unlikely to regrow quickly enough to be susceptible to fire again within that time.

Total burned area and the sum of $1/T$ were derived for each combination of month, region and land-cover type to derive a burned area estimate for lone hotspots:

$$E_L = BA / \sum_j (1/T_j) \tag{6}$$

where $E_L$ is the area-burned estimate for single (lone) hotspots, $BA$ is the total burned area, and $T_j$ is the times burned metric and $j$ is the hotspot number. Note that for repeatedly burned pixels, the use of Eq. (6) prevents their burned area, and consequently their emissions, from being overestimated.

For region-month-landcover combinations with fewer than 1000 hotspots, the resulting $E_L$ values were not statistically significant; in this case, a larger dataset region-month combinations were used instead, combining all the land-cover types together within a region and month.

In subsequent emissions calculations, the number of times burned is similarly calculated. Area estimates for lone hotspots ($T = 1$) were set equal to $E_L$. For hotspots in previously burned pixels, the burned area estimate was set to $E_L$ divided by the times burned metric:

$$E = E_L/T \tag{7}$$

This method reduces the burned area in multiple-hotspot locations, preventing the same fuel from being burned multiple times during emissions calculations. Hotspots generated by industrial heat sources remain in the dataset, but they are assigned a very small burned area; as a result, their impact on emissions estimates is minimized.

## 4.2 Fuel Load

The fuel loads ($FL$) used in GFFEPS are based on values collected from the literature review by van Leeuwen et al. (2014) and from van der Werf et al. (2017) as used in GFED. From these data, fuel load values were assigned to surface, crown and grass fuel loads ($SFL$, $CFL$, $GFL$) and averaged across sites with a common GLC2000 land classification. The source data had fields ranging from simple totals to very specific fuel component descriptions per site. When this ancillary data was available, certain heavier fuels were excluded from fuel loads, such as live stems and branches with diameters greater than 10 cm, and were deemed inflammable (as residual snags).

Following this initial classification, attention was given to regional differences, many of which are summarized in Appendix A, especially for high-emitting regional land classifications. These include boreal forests, tropical forests, tropical peat, wooded and open savanna grasslands, and Australian eucalypt forests.

## 4.3 Combustion Completeness

The combustion completeness ($CC$) used in GFFEPS is captured by the total fuel consumption ($TFC$) as calculated by FBP, which is equal to the product of Seiler and Crutzen's fuel load and combustion completeness ($FL \times CC$). The forecasted weather and FWI described earlier are combined with the FBP fuel types as derived from the GLC2000 land classification to provide the necessary inputs for the FBP calculations. Total fuel consumption per time step is then calculated assuming a diurnal pattern of area growth per hour (Chen et al., 2019).

In implementing a global system, adjustments to the original FBP fuel loads and fuel consumption equations were required. The FBP system was designed specifically for Canadian fuel types (Table 1); extrapolating these to a global environment for fuels outside Canada was necessary. In this process, a few critical limitations in the Canadian-specific FBP system were also recognized and addressed, specifically:

- Surface fuel loads ($SFL$) were used to adjust the surface fuel consumption ($SFC$) equations within FBP; replacing the original 1.5 and 5.0 kg m$^{-2}$ present in most FBP fuel consumption equations.

- Grass fuel consumption (*GFC*) was separated from surface fuel consumption (*SFC*); along with crown fuel consumption (*CFC*), expressing a new total fuel consumption (*TFC*):

$$TFC = GFC + SFC + CFC. \tag{8}$$

GFC was adjusted to account for the degree of curing, the process by which grass dries over the season, by multiplying the grass fuel load by the grass curing adjustment factor (*C*):

$$GFC = C \times GFL. \tag{9}$$

- Grass curing adjustment factor was based on the equation derived for dormant grass in savanna grasslands using drought code (*DC*) values (see Appendix A.3):

$$C = 100\% \times (1 - e^{-0.0027\,DC}). \tag{10}$$

- Green-up (leaf-out) of deciduous forests (FBP fuel type D-1 and D-2, Table 1), normally a dichotomous process in FBP (Alexander, 2010b), was set to a fractional scale and using this the surface fuel consumption (*SFC*) of green deciduous (D-2) was derived by adjusting the surface fuel consumption for FBP class of leafless deciduous (D-1) by (1 - *GSI*):

$$SFC(\text{D-2}) = (1 - GSI) \times SFC(\text{D-1}). \tag{11}$$

### 4.3.1 Peat Fires

Fires in equatorial Asia (Indonesia, Malaysia and New Guinea) in GLC2000 land classifications 7, 8, 9, 11 and 14 (see Table 2) were assumed to be peat fires, which require special consideration. Field et al. (2004) determined that most severe haze events from peat fire smoke in Indonesia occurred at a Drought Code (*DC*) value of 388.2 and higher. They accordingly assigned boundaries between moderate-high and high-extreme categories at *DC* values of 264.4 and 346.9. Based on these values, a logistic equation was constructed to mimic these conditions

$$SFC = 105.6/\left[1 + e^{-0.1(DC-388.2)}\right] \tag{12}$$

where 105.6 kg m$^{-2}$ reflects the fuel load of the tropical peatland fuels (van Leeuwen et al., 2014). A detailed description of this derivation is provided in Appendix A.3.

Outside of equatorial Asia, boreal peatlands were assessed as treed (shaded, enclosed) or open peat lands, based on, respectively, the *Tree Cover, regularly flooded, freshwater* or *Regularly flooded Shrub and/or Herbaceous Cover* descriptions in the GLC2000 land classification (Table 2). Depending on current FWI conditions, a treed peatlands fuel type is assigned to boreal mixedwood forest with 50% conifer trees (fuel type M-1/2) when the *DC* is above 330, and to fuel type D-2 at lower *DC* values; open peatlands are assigned a non-fuel type until the *DC* reaches 650, at which point they assumed to burn as fully cured standing grass (O-1b). These thresholds are prescribed following earlier study of Thompson et al. (2019).

### 4.3.2 Eucalypt

Over 22% of Australia is forested, of which 78% is eucalypt, also known as jarrah (Sullivan et al., 2012). Eucalypt does not fit any fire behaviour reflected in the FBP system so an effort was made to create a fuel consumption model specific to eucalypt from the published literature (Hollis et al., 2010). A sigmoidal consumption completeness curve was developed similar in structure to those used in the FBP System (Appendix A.5). An upper limit of 90% was used as it was assumed that standing snags would likely be left after a fire-front passage. The resulting equation is

$$CC = 90\% \times [1 - e^{(0.01976\,BUI)}]^3 \tag{13}$$

where $CC$ is the combustion completeness (%). Total fuel consumption for eucalypt is achieved by multiplying combustion completeness by a eucalypt fuel load of 7.8 kg m$^{-2}$ as used in GFFEPS (Sullivan et al., 2012).

Additionally, Oliveira et al. (2015) examined fire activity in tropical savannas in northern Australia. They described the landscapes as open woodlands, woodlands and open forests with forest protective covers of <10%, 10-30% and 30-70% respectively. Average values of these fractions were used in GFFEPS, with the balance as grass fuels. A detailed description of this derivation is provided in Appendix A.5.

### 4.4 Emissions Factors

For emission factors per chemical species ($EF$), GFFEPS uses the values presented in Chen et al. (2019) and Urbanski (2014). Combustion is divided into three classes based on the crown, surface and grass fuel consumptions. Surface fuel is further divided into litter (0-1.2 cm), upper (1.2-7 cm) and lower (7-18 cm) duff layers following fuel-based depths and fuel-dependent bulk densities (mass of fuel per unit volume in g cm$^{-3}$, Anderson, 2000; https://www.ciffc.ca/publications/glossary, last accessed 2024-05-28). The fuel consumed in each layer is burned in succession through flaming, smoldering and residual combustion stages, which are then convolved with area growth over time. Emission rates per chemical species emitted are defined through each stage of combustion by combining emission factors for flaming, smoldering, and residual with FBP's $CFC$, $SFC$ model values. By modelling the total fuel consumption per unit area (kg m$^{-2}$), emissions per species are calculated based on species emission factors (g kg$^{-1}$) as defined in Chen et al. (2019).

In the current initial application of GFFEPS, for direct assessment of fuel consumption values, and comparison with other global fire emissions inventory, a simple unit emission factor is first presented for estimating smoke emissions, followed by an application of standard emission factor of 500 g kg$^{-1}$ for estimating total carbon emissions (Thomas and Martin, 2012).

## 5 Results

The GFFEPS model was run for six consecutive years (2015 to 2020) to examine the quantitative fire emissions globally for each year, and interannual variability predicted by the model. Model output was measured in total smoke emissions released from fires. This equals the total fuel consumed by fire assuming a unit emission factor (1 kg kg$^{-1}$), thus allowing for

a direct comparison to the source FBP calculations. Afterwards, a multi-model comparison using carbon emissions factors (500 g kg$^{-1}$) was conducted between GFFEPS and four other published wildfire emissions models and Inventories. Results in both sections were broken down into the 14 regions following Giglio et al. (2006). See Fig. 2 for the region descriptions and the abbreviations used.

**5.1 GFFEPS Total Smoke Emissions**

Figure 9 shows the regional, annual values of (a) smoke emissions, (b) burned area and (c) average total fuel consumption per unit area. Total smoke emissions and burned area are directly estimated by the GFFEPS model. Average total fuel consumption per unit area was calculated as smoke emissions over burned area for each of the analysis regions, allowing a comparison of regional model results to the original FBP fuel consumption calculations.

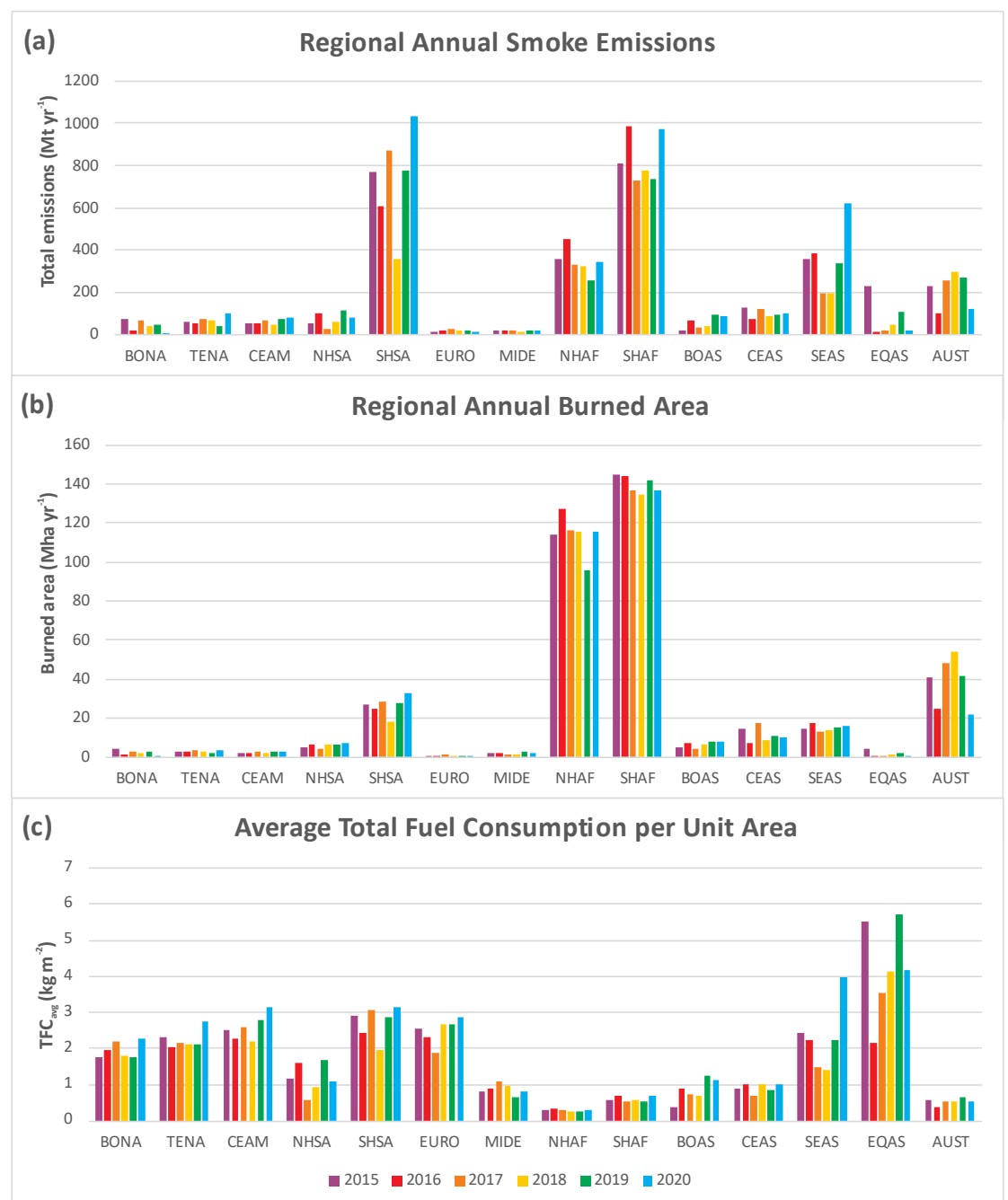

**Figure 9. (a)** Regional annual smoke emissions (Mt yr$^{-1}$), **(b)** regional annual burned area (Mha yr$^{-1}$) and **(c)** average total fuel consumption per unit area (kg m$^{-2}$) as predicted by GFFEPS for 2015-2020. Smoke emissions reflect all emissions released from fires with an emission factor equal to the total fuel consumed (1 kg kg$^{-1}$). See Fig. 2 for descriptions of regional abbreviations.

Total smoke emissions over the six consecutive years, as shown in Fig. 9(a), indicate the largest emitters being SHAF and SHSA, with average annual smoke emissions of 834 and 736 Mt, respectively. Interannual values are relatively consistent through most regions, with the largest range (maximum/minimum) occurring in EQAS (225/12 Mt), BONA (71/6 Mt), BOAS (94/18 Mt), NHSA (109/24 Mt) and SEAS (619/195 Mt). El Niño likely drives the variability in EQAS and SEAS, while fire weather conditions likely determine the variability in the two boreal regions. The figure also shows possible impacts of El Niño (strong in 2015/16, weak in 2018/19 and early 2020; McPhaden, 2023) and changing deforestation legislation in Brazil, affecting South American emissions.

Figure 9(b) shows the burned area per region per year. Sub-Saharan Africa (NHAF+SHAF, but excluding MIDE) dominates the global burned area at 254 Mha (69%) of the global average 368 Mha burned annually. This is followed by 38.5 Mha in AUST and 32 Mha in South America (NHSA+SHSA).

The regional burned area predicted by GFFEPS can be compared to national statistics reported by certain countries. Model results indicate that on average 2.06 Mha yr$^{-1}$ in BONA (Canada and Alaska) during the 6 study years. For the same period, Canada's National Forest Database reported 2.19 Mha (http://nfdp.ccfm.org/en/data/fires.php, last accessed 2024-05-28) while Alaska Department of Natural Resources reported 0.64 Mha (https://forestry.alaska.gov/firestats/index, last accessed 2024-05-28). The sum of the two reported values being 2.83 Mha yr$^{-1}$, which exceeds the GFFEPS prediction by 0.77 Mha. Similarly, GFFEPS predicted on average 2.77 Mha yr$^{-1}$ in TENA, while US agencies reported 3.18 Mha yr$^{-1}$ in the lower 48 states for the same six years (https://www.nifc.gov/fire-information/nfn, last accessed 2024-05-28). While GFFEPS estimates only 73% and 87% of the observed values respectively, a correlation between modelled and reported annual values for the six years is strong in each region ($r^2 = 0.968$ in BONA; $r^2 = 0.914$ in TENA, not shown). This suggests the methodology for estimating burned area used by GFFEPS is appropriate, though with a bias. On the other hand, reported national statistics of burned area have their own sources of error. For example, the level of rigour in mapping varies between Canadian provincial and territorial agencies, where unburned areas within fire perimeters may be captured by some agencies and not by others. This variable quality is then passed onto the national statistics. Similar issues are likely occurring in US statistics. The issue of mapping irregularities was also recognized by Fraser et al. (2004), who indicated the coarse resolution burned-area (approx. 1-km) provided by SPOT VEGETATION and NOAA AVHRR imagery produced burned-area estimates 72 percent larger than the crown fire burned area mapped at 30 m using Landsat TM (11,039 versus 6,403 ha average area). This bias was attributed to spatial aggregation effects. In summary, it is difficult to make clear conclusions from national statistics but these indicate the GFFEPS methodology is producing realistic results.

Average total fuel consumption per unit area by year and region, as shown in Fig. 9(c), was calculated as smoke emissions over burned area from the annual results. Globally, the average is 0.81 kg m$^{-2}$, while regional results vary from 0.30 kg m$^{-2}$ in NHAF to 4.21 kg m$^{-2}$ in EQAS. Figure 9(c) clearly show regions dominated by forest (e.g., BONA, TENA and CEAM) as having higher fuel consumption per unit area on a global basis compared to those dominated by grasslands (e.g., NHAF and SHAF). The figure also shows regions strongly affected by El Niño events (EQAS, SEAS) with annual consumption rates doubling in El Niño years (strong in 2015/16, weak in 2018/19 and early 2020; McPhaden, 2023).

Figure 10 shows the regional daily smoke emissions for the six study years by day of year. Largest emissions occur in SHAF during the region's dry season (mid-May to mid-September) and in SHSA at the end of the dry season (August to mid-October). The latter would be consistent with deforestation burning (Pereira et al., 2022).

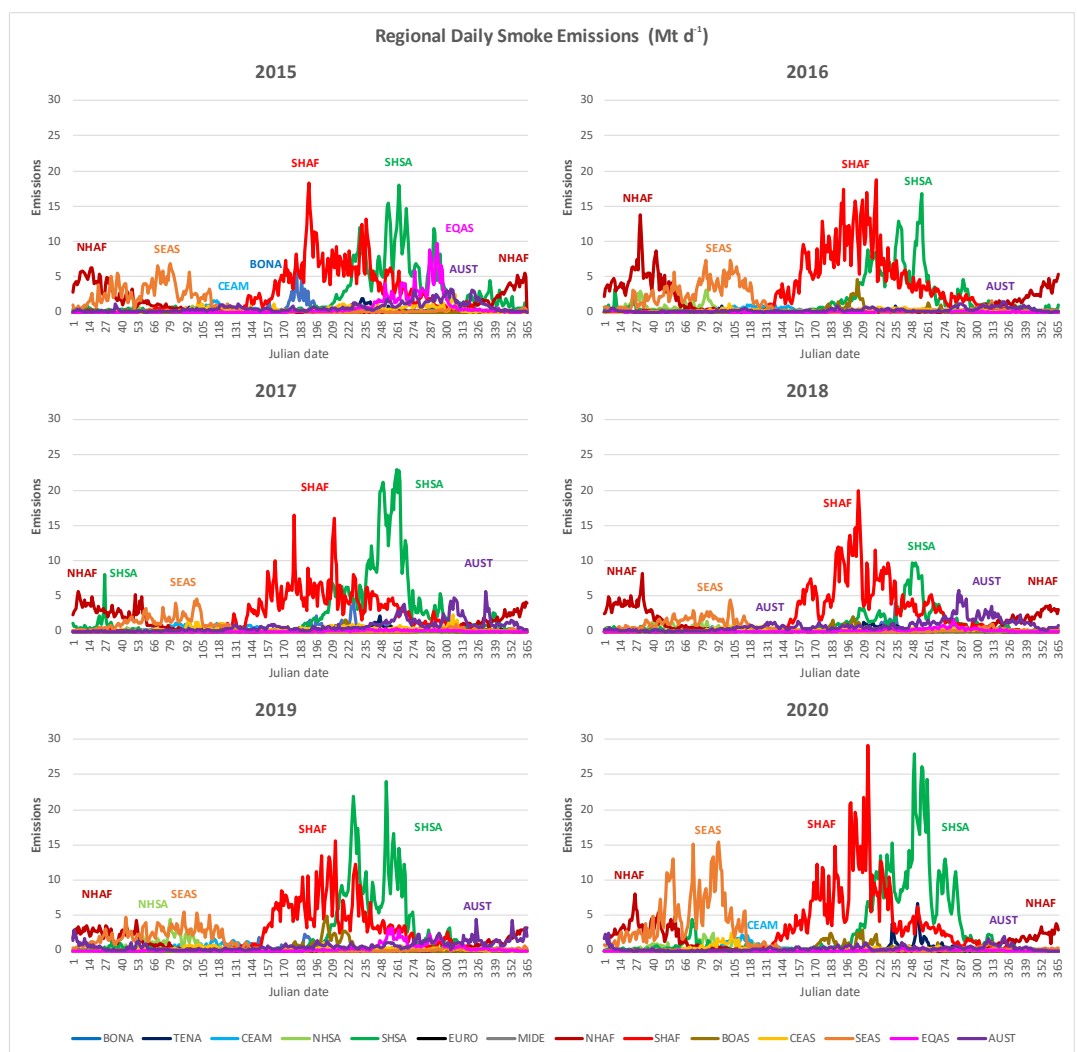

**Figure 10. Regional daily smoke emissions (Mt d⁻¹) for GFFEPS for the six study years as predicted by GFFEPS.**
**Smoke emissions reflect all emissions released from fires with an emission factor equal to the total fuel consumed**
**(1 kg kg⁻¹). See Fig. 2 for descriptions of regional abbreviations.**

### 5.2 Comparison of GFFEPS to other Wildfire Emissions Models and Inventories

        As noted above, the GFFEPS model was run for six consecutive years (2015 to 2020). Results for global carbon
emissions were compared to published results for

- GFAS (Kaiser et al., 2012, https://www.ecmwf.int/en/forecasts/dataset/global-fire-assimilation-system , last accessed 2024-05-27)
- GFED4.1s (van der Werf et al., 2017, https://www.geo.vu.nl/~gwerf/GFED/GFED4/, last accessed 2024-05-27),
- FINN version 1.5 (Wiedinmyer et al., 2011, https://www.acom.ucar.edu/Data/fire/, last accessed 2024-05-27),
- FINN version 2.5 (Wiedinmyer et al., 2023, https://rda.ucar.edu/datasets/ds312.9/dataaccess/, last accessed 2024-05-27).

Annual values of global carbon emissions for all five models are presented in Fig. 11. Results show a wide range of values from 1166 Tg C $yr^{-1}$ in 2018 by GFFEPS to 4231 Tg C $yr^{-1}$ in 2019 For FINN 2.5. In half of the years, GFFEPS produced the lowest results with values ranging from 1166 to 1789 Tg C $yr^{-1}$. Compared to the other models, GFFEPS estimated values lower than GFAS/GFED (80%/74%), while it estimated values similar to FINN1.5 (97%). The lower values are largely attributed to the inclusion of daily fire behaviour in the combustion completeness calculations, not accounted for in the other models.

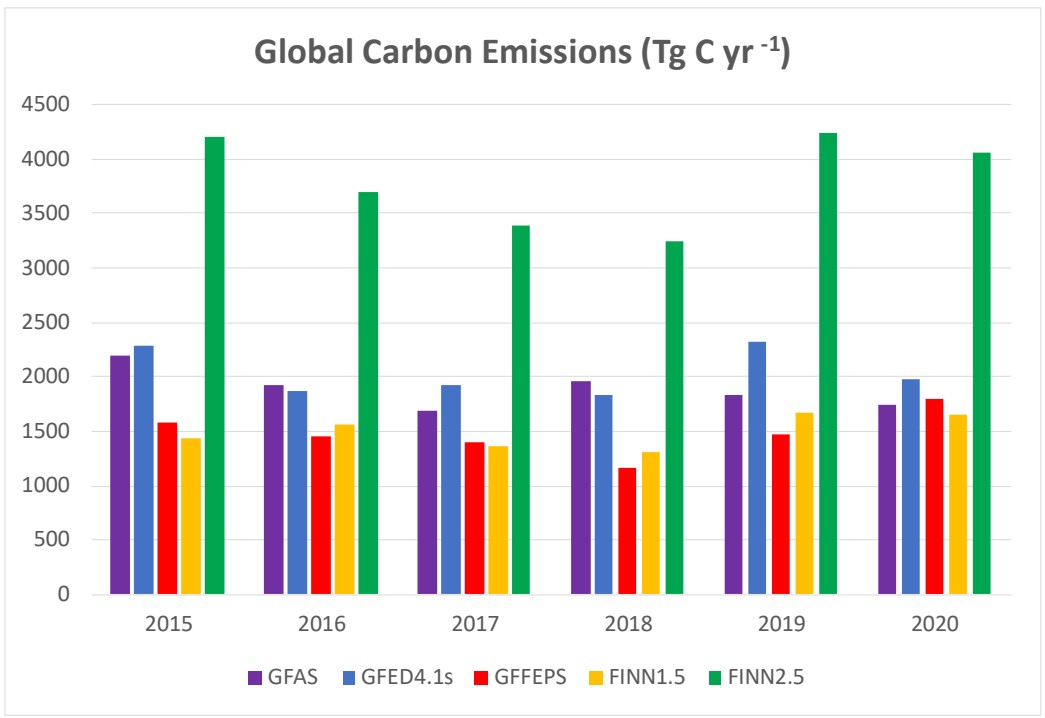

**Figure 11. Annual carbon emissions (Tg C $yr^{-1}$) of GFFEPS and other global wildfire emissions models included in this study.**

Figure 12 shows a comparison of average annual regional carbon emissions from GFED4.1s and GFFEPS (regional values were not readily available for the other models). The regions of largest GFED emissions are much lower in GFFEPS.

Sub-Saharan Africa (NHAF+SHAF) accounting for 1007 Tg C (49.5% of the total global emissions) in GFED, is reduced to 588 Tg C (39.8%) in GFFEPS. On the other hand, South America (NHSA+SHSA) increases from 304 Tg C (14.9%) in GFED to 403 Tg C (27.2%) in GFFEPS. Also, GFFEPS has greater emissions in six of the 14 regions: CEAM, SHSA, EURO, MIDE, CEAS and SEAS. These are areas dominated by agricultural burning, highlighting the impact of using FAO's crop-burning statistics.

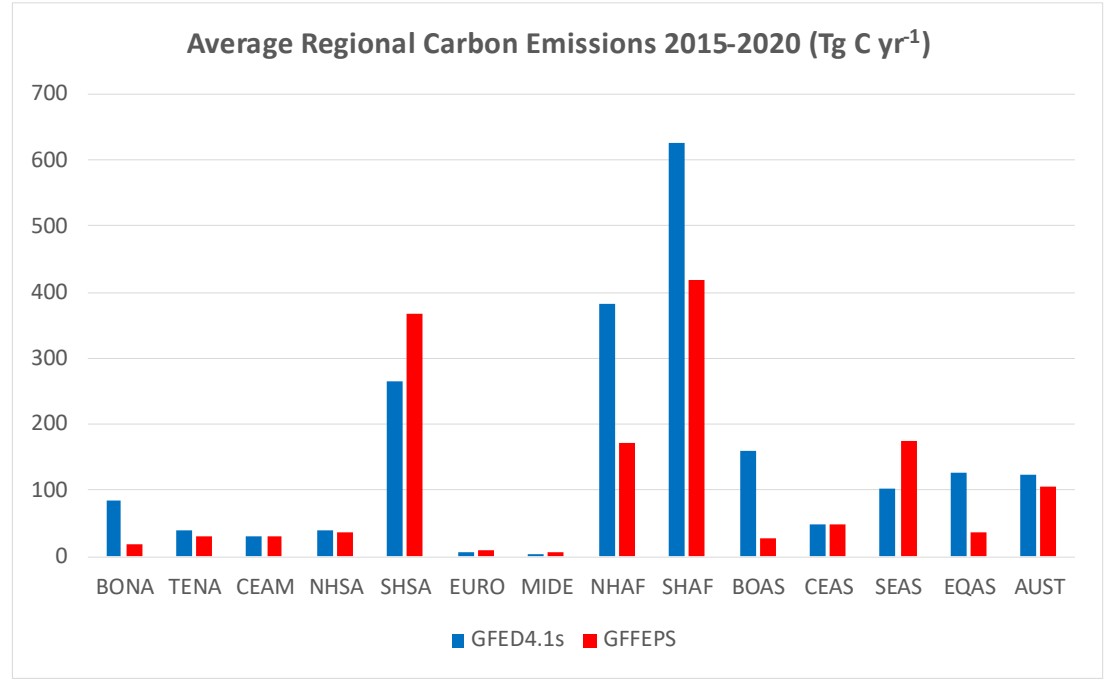

**Figure 12. Average annual emissions (Tg C yr$^{-1}$) by region for GFED4.1s and GFFEPS. See Fig. 2 for descriptions of regional abbreviations.**

Figure 13 shows the annual burned area from the MODIS burned area (MCD64A1) that is used by GFED prior to incorporating small fires, GFFEPS and FINN 1.5/2.5. FINN 1.5 calculates burned area based on active fire pixels detected by the MODIS Aqua and Terra satellites at 1 km$^2$ (0.75 km$^2$ in grasslands/savannas) per detection, which is then adjusted by percent tree, non-tree vegetation, and bare cover at 500m as provided by MODIS Vegetation Continuous Fields (VCF). FINN 2.5 (Wiedinmyer et al., 2023) uses a more sophisticated approach, aggregating VIIRS hotspots to create burned area polygons. GFFEPS is in line with most area-burned statistics including the MODIS burned area (MCD64) and FINN 1.5, while FINN 2.5 appears to estimate twice the burned area of the other models.

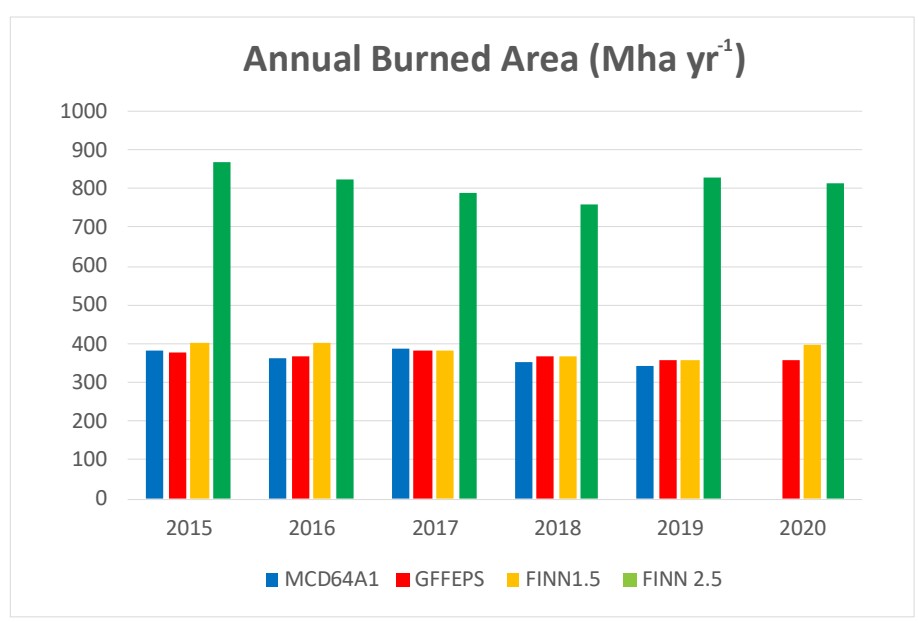

**Figure 13. Annual burned area (Mha yr⁻¹) of GFFEPS and other global wildfire emissions models included in this study.  The MCD64A1 data reflects the burned area data used by GFED (prior to small fire adjustments).**

Daily burned-area values are available in FINN products allowing a comparison between GFFEPS and the two FINN implementations.  Figure 14 shows a sample comparison (2017) between GFFEPS and FINN 1.5/2.5.  This pattern is similar to other years.  GFFEPS shows lower area-burned amounts during Feb-Mar and higher during Oct-Nov.  This may be occurring during harvest periods when small fires dominate some landscapes.

        A comparison of daily area-burned values suggests a pattern of results where GFFEPS burned area consistent with

675 FINN 1.5 (MODIS-based), while FINN 2.5 is predicting twice the burned area. Simple regressions indicate correlations (not shown) of $r^2$=0.61 between GFFEPS and FINN 1.5 and 0.71 between GFFEPS and FINN 2.5 (when the intercept is forced to zero, the correlations increase to 0.92 and 0.94 respectively).  The close agreement between GFFEPS and FINN 1.5 is of interest as FINN1.5 differs from GFFEPS in its method of calculating area burned.  On the other hand, FINN 2.5 approach, using aggregated VIIRS hotspots to create burned area polygons, increases the burned area by a factor of two, which is reflected

in the higher carbon emissions shown in Fig. 11.  These values are in-line with the global annual emissions estimate of 774 Mha yr⁻¹ produced by most recent GFED5 (Chen et al., 2023).  A similar approach is currently being considered for GFFEPS.

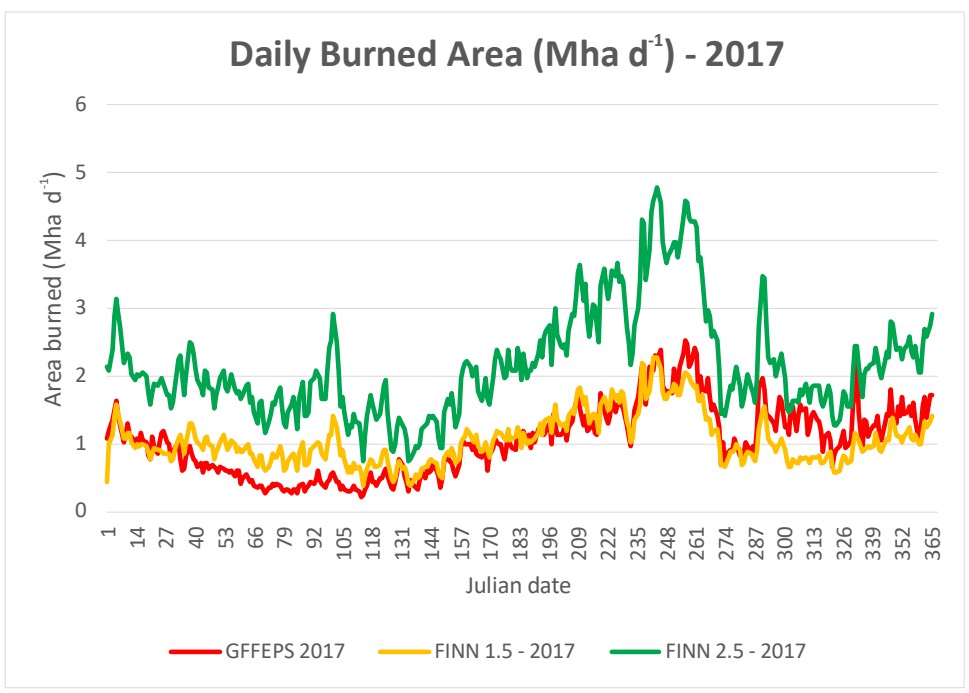

**Figure 14. Daily burned area globally (Mha d$^{-1}$) for GFFEPS and FINN1.5/2.5 for the study year 2017.**

## 6 Discussion

There are no direct measurements of global fire emissions and thus there is no definitive answer as to which of the five models/versions examined in this study provides the best estimate of fire emissions on a global scale.  Based on the principles

of fire, physics and remote sensing, we demonstrated that the GFFEPS global fire emission estimates are reasonable and realistic.  Pan et al. (2020) demonstrates the range of predictions from six models, while this manuscript shows the range of predictions among three published models and GFFEPS.

Results from six consecutive years of emissions comparisons show that the GFFEPS model is in general agreement with well-established models.  Each of these models emphasizes one aspect over the others in the Seiler and Crutzen equation

(Eq.1): GFED places its efforts on accurately predicting area burned, while FINN focuses on emissions factor estimates for a large number of chemical species.  The methodology presented in GFFEPS focuses on the dynamic predictions of fire behaviour, fuel consumption and emissions on a daily basis.

With regards to the similarities between GFFEPS and GFED4.1s, this should not be a surprise as the GFFEPS methodology and input data is similar to that used in GFED4.1s.  Nonetheless, the key essential differences between the two

models are that GFED4.1s uses static fuel loads and consumption completeness per biome, while GFFEPS models these

dynamically, both spatially and temporally, achieved by using the well-established CFFDRS with FBP fuel consumption driven by FWI fire weather; that GFFEPS considers plant phenology not explicitly recognized in GFED; and that GFFEPS calculates real-time burned area based current hotspots and historical statistics, while GFED uses burned area data accumulated over the course of a month from remotely sensed data. While the underlying CFFEPS system was designed for Canada and
North America, model results show that the approach making use of CFFDRS parameters is robust and adaptable to conditions beyond North America.

The benefit of producing the three components of Fig. 9 is important as that it helps to validate the GFFEPS calculations. While we cannot directly measure global emissions, we can measure certain components. The burned area (Fig. 9(b)) can be directly compared to national statistics where available, while the total fuel consumption per unit area (Fig. 9(c)) appear to fit
within expected values for various landscapes. Together, they indicate the calculated global emissions (Fig. 9(a)) produced by GFFEPS are realistic. Further refinement of the burned area and fuel consumption models will help to improve model accuracy.

Figure 9 also helps to illustrate the source of variability in global emissions. For example, the figure shows the magnitude of smoke emissions (Fig. 9(a)) in sub-Saharan Africa is primarily a result of the burned area (Fig. 9(b)) by low-
intensity fires, as indicated by the low value for the total fuel consumption per unit area (Fig. 9(c)). Conversely, higher fuel consumptions (Fig. 9(c)) are shown in the forested regions in North America while variable consumptions in southeast and equatorial Asia reflect the impact of El Niño on the regions.

When compared to other models, differences in estimated carbon emissions appear between the models within and across regions. Indeed, each of the models may be superior at modelling emissions in specific regions, while weaker in others.
Evaluating regional variability is beyond the scope of this study. Other factors appear in the inter-annual results such as possible impacts of changing deforestation burning policies in Brazil as emissions vary from year to year (Fig. 9(a); Schmidt and Eloy, 2020). El Niño events have been linked to global fire activity and emissions, and representation of this in weather data used by models can vary and appear linked to emission differences as impacts to southeast and equatorial Asia in 2015/2016, 2018/2019 and early 2020.

The GFFEPS model is largely based on the well-established CFFDRS system of fire behaviour and fuel consumption and the regional CFFEPS fire emissions model. The inclusion of the CFFDRS system allows for a clear and scientific method to directly incorporate NWP model forecasted meteorological conditions, near-real-time fire location measurements, and fuel moisture estimates as driving forces in daily fire activity accounting and emission calculations. Of the models presented in this study, GFAS and FINN provide comparable, near-real-time products, yet they do not address the near-real-time dynamic
fuel moisture and fire behaviour captured by the CFFDRS as used in GFFEPS.

Extending the CFFDRS to a global environment was a challenge, and in this initial global application exercise, several important assumptions were made. One such assumption was the introduction of the *GSI* as a means for modelling plant phenology responses in predicting seasonal leaf-out of deciduous forests, and grass curing though a *DC*-based approach. Applying these effects on fuel consumption was understandably unaddressed in the original, Canadian-focused, FBP system.

Canada's fire danger group focused much of its attention on hazardous fuels, capturing spread rates and fire behaviour in the situations that threaten fire fighter and community safety; little attention was made for the aftermath of fire activity in terms of accounting for smoke and carbon emissions in the 1970s through 1990s when the Canadian FBP system was developed. Also, green grass and leaf-out deciduous posed little threat and thus received cursory assumptions.

Another issue in extending CFFEPS to a global domain was the lack of data from field experiments and measurements
outside of Canada encompassing more diverse environment conditions. This was required not only for validation, but also for building a parameterization to expand the FBP approach to modelling fire behaviour in a broader domain (as presented in Appendix A). Papers such as Hoffa et al. (1999) and Shea et al. (1996) were invaluable in understanding fires in African savannah. There again, the authors focused their attention on the dry season and highest flammability, and this may influence GFFEPS results outside of these high-burning seasons.

The methodology of assigning burned area per hotspot with the burned-area climatology dataset was an early assumption of CFFEPS carried over into GFFEPS. It provides a means of predicting burned area in near-real-time for model forecast operational applications as compared to the hindcast, retrospective approach used in most other global fire emissions models. Discrepancies did arise, as evident in current 2016-2020 comparison where GFFEPS underestimates burned area in boreal and temperate North America relative to nationally-reported statistics.

Incorporating small fires was recognized as a non-negligible issue. Researchers developing the GFED model focused efforts into extrapolating burned area by small fires from coarser-resolution data, whereas, the use of United Nations' FAO crop-burning statistics for agricultural regions in GFFEPS provided an alternate route, following methods commonly applied in anthropogenic emission inventory assessments (Streets et al., 2003). While small fires may have some impact on fires outside of the agricultural zone, it was deemed an acceptable route given the relative contribution of agricultural fires compared
to wildfire emissions.

GFFEPS follows the satellite-based fire detection methodology and is faced with the traditional issues associated with that approach, namely restrictions due to satellite-overpass times, sensor resolution, observational swath width, heavy smoke and cloud cover. Other limitations of the bottom-up approach used by GFFEPS include land-cover and burned-area mapping resolution as well as the accuracy of fuel load mapping and fuel consumption modelling.

The GFFEPS model as presented in this manuscript has demonstrated the ability to model fuel consumption dynamically and its utility for forest fire emissions simulations, particularly in near-real-time forecasting applications, on a global level. There is the potential for future improvements. Many of the spatial components, specifically FBP fuels and fuel load assignments, need more rigorous examination and validation. A number of assumptions and generalizations were made to allow the model to function using global input data. Further effort could improve on and validate these initial findings. The
model as developed is adaptable and open to improvements.

Efforts are currently underway to validate GFFEPS against TROPOMI measurements, similar to Canada-only plume rise (Griffin et al., 2020) and CO emissions (Griffin et al., 2023) exercises that have yielded favourable results. Other regional studies may provide additional validation data through remote sensing, particularly on a regional or individual fire basis. For

example, Nguyen and Wooster (2020) estimated biomass burning in Africa using geostationary fire radiative power (FRP) and aerosol optical depth (AOD); Hayden et al. (2022) conducted airborne measurements of 193 compounds from 15 instruments, including 173 non-methane organics compounds (NMOG) downwind of a small peat-dominated wildfire at La Loche, Saskatchewan, as part of the Alberta oil sands field study;  Adams et al. (2019) used remote sensing to directly measure CO, $NH_3$ and $NO_2$ from the 2016 Horse River fire near Fort McMurray, Canada, while Stockwell et al. (2022) conducted similar measurements over western US fires.  Applying such approaches on a global scale would be beneficial for further validation of GFFEPS as well as assessing the feasibility in further applications with global chemical transport models.

Future direction of the GFFEPS model includes integration with the global GEM-MACH chemical transport model, and running the model operationally to provide boundary data and input for the regional FireWork model utilizing CFFEPS. This would allow for the transcontinental transport of smoke and further refine the regional air-quality forecasts for Canada. Efforts are underway to link CFFEPS with a predictive fire-growth model (Anderson et al., 2009) and coupling the impact of smoke plumes generated by CFFEPS on ground temperatures as presented in public forecasts (Makar et al., 2020).  Finally, steps have begun to link GFFEPS to the Canadian Earth System Model (CanESM5; Swart et al., 2019) and the Canadian fourth generation atmospheric global climate model (CanAM4; von Salzen et al., 2019) for integrated study of climate driven impacts on regional wildfire risks, and air quality analysis.

**7 Conclusion**

This paper presents the Global Forest Fire Emissions Prediction System (GFFEPS) as a model to estimate emissions of smoke from biomass burning globally.  Based on the regional Canadian Forest Fire Emissions Prediction System (CFFEPS), the methodology has been extended to a global environment.  Both systems are based on the well-established Canadian Forest Fire Danger Rating System.  By using forecasted 3-hour meteorological conditions produced by Environment and Climate Change Canada's Canadian Global Elemental Multiscale (GEM) model, daily fire weather calculated with FWI, and fire behaviour, area growth, fuel consumption estimated from FBP, the GFFEPS model is shown to produce estimates fire emissions in an operational setting.

The model uses a bottom-up approach and is based on remotely-sensed hotspot locations and predicted burned area. Using forecasted meteorological conditions, daily fire weather, historical burned area per hotspots and a global land classification at a 1-km resolution, GFFEPS provides dynamic estimates of fuel consumptions and area growth in near-real time, differentiating it from other global emissions models.

A study was conducted running GFFEPS through a six-year period (2015-2020).  Results were compared to other global emissions models including GFAS, GFED4.1s and FINN1.5/2.5.  GFFEPS estimated values lower than GFAS/GFED (80%/74%), while it estimated values similar to FINN1.5 (97%).  Differences are largely due to its inclusion of daily weather as predicted by the GEM model and fire behaviour modelling provided through the CFFDRS.

This manuscript presents the initial release of the GFFEPS model. Its development is on-going and future avenues are recognized and being pursued, including incorporating the model to existing air-quality models, coupling CFFEPS/GFFEPS with predictive fire-growth models, and linking the model to global climate models. This paper presents the methodology currently used in the model, and shows it providing realistic results in line with other models. Efforts are underway to continue validation of the model, improve its sub-components and expand its use to other global air-quality and climate models.

## 8 Appendix A. Supplemental Information: Fuel Consumption Models

        Efforts to validate fuel consumption models used in GFFEPS were conducted using data from published studies. These studies documented observed weather, fire behaviour and fuel consumption associated with prescribed fires in specific landscapes and forest stands. These results are compared with fuel consumption predicted by GFFEPS and by GFED4.

GFFEPS follows the Canadian Forest Fire Danger Rating System (CFFDRS, Stocks et al., 1989), specifically, the Canadian Forest Fire Weather Index (FWI) system (Van Wagner 1987) and the Canadian Forest Fire Behavior Prediction (FBP) system (Forestry Canada Fire Danger Group 1992; Wotton et al., 2009). To calculate fuel consumption, GFFEPS requires:

- a fuel model compatible with the FBP system,
- FWI values on the date of the fire,

- latitude, longitude, and Julian date for Foliar Moisture Content (*FMC*) calculation,
- daylength and vapour pressure deficit for Growing Season Index (*GSI*) calculation (Jolly et al., 2005).

        GFFEPS uses the Global Land Cover 2000 Project (GLC2000, Bartholome and Belward, 2005) to determine fuel models. GLC2000 provides spatial land cover classifications for the globe at a 1-km resolution. For the purposes of validating fuel consumption, a representative GLC2000 classification, shown in italics (e.g., *needle-leaved, evergreen*), was selected for

each study landscape.

        Historical fire weather values were taken from a high-resolution (0.25°) global re-analysis of fire weather conditions from 1979 to 2018 (McElhinny et al. 2020), except when values were included in specific studies (Alexander et al., 1991, Stocks et al., 2004, Stocks 1989, Stocks 1987a, Stocks 1987b, Quintilio et al., 1991). Duff moisture Codes (*DMC*) and Drought Codes (*DC*) were retrieved and from these, Buildup Indexes (*BUI*) were calculated following the FWI system equations. For

the purposes of this study, daily values of *GSI* were used in place of 21-day averages as historical weather to calculate a 21-day average were not readily available (historic papers typically included meteorological values the day of observed burns alone).

        Given the input represented or derived from observed data in each individual study, predicted GFFEPS fuel consumption was calculated using the FBP system equations. Fuel loads, largely based on van Leeuwen et at. (2014), were

used as global default values in the FBP calculations (see 4.2 Fuel Load); regional fuel load values presented in this

supplemental information section replace global defaults. Consumption rates following the GFED methodology are also presented for comparison. Note that GFED values are based on version 4.1s fixed fuel loads and consumption rates per region and fuel with no allowance for variable meteorology and fire weather (see 1.0 Introduction).

## A.1 Boreal Forest

The Canadian Forest Fire Behaviour Prediction (FBP) System is based on case studies of fire behaviour in the Boreal Forest (Table A.1). These studies include fuel loads and depths, noon weather observations (temperature, relative humidity, wind speed, etc.) as well as the calculated FWI values (*FFMC*, *DMC*, etc.). Fuel loads used in GFFEPS were based on default values in the FBP manual.

**Table A.2~~1~~. Canadian Forest Fire Behavior Predictions (FBP) System fuel types included in this study.**

| FBP | Fuel description | Reference | Surface fuel load (kg/m$^2$) | Crown fuel load (kg/m$^2$) |
|---|---|---|---|---|
| C-1 | Spruce–Lichen Woodland | Alexander *et al.* 1991 | 1.5 | 0.75 |
| C-2 | Boreal Spruce | Stocks *et al.* 2004 | 5.0 | 0.8 |
| C-3 | Mature Jack Pine | Stocks 1989 | 5.0 | 1.15 |
| C-4 | Immature Jack Pine | Stocks 1987a | 5.0 | 1.20 |
| M-3/4 | Dead Balsam Fir Mixedwood–Leafless | Stocks 1987b | 5.0 | 0.8 |
| D-1 | Leafless Aspen | Quintilio *et al.* 1991 | 1.5 | * |

*Crown fuel load for D1 is not applicable

## A.1.1 Coniferous

The GLC2000 lacks the detail required to distinguish all the fuels presented in these studies. Instead, *needle-leaved, evergreen* land cover classification is represented in GFFEPS simply as a C-2 (boreal spruce) fuel type for North America.

GFFEPS thus uses the C-2 surface fuel consumption calculation with the default C-2 surface fuel load of 50 t ha$^{-1}$ and an average crown fuel load of 10 t ha$^{-1}$ as documented in the FBP manual.

Figure A.1 shows the scatter plot of observed versus predicted total fuel consumption. Predicted values are based on GFFEPS calculations, assuming all fuels as C-2 (boreal spruce) fuel type, while using the observed using weather conditions from the source papers. The resulting correlation coefficient ($r^2$) was 0.416. Forcing the regression through the origin, we

find the predicted data is overpredicting the observed fuel consumption by only 2.5%.

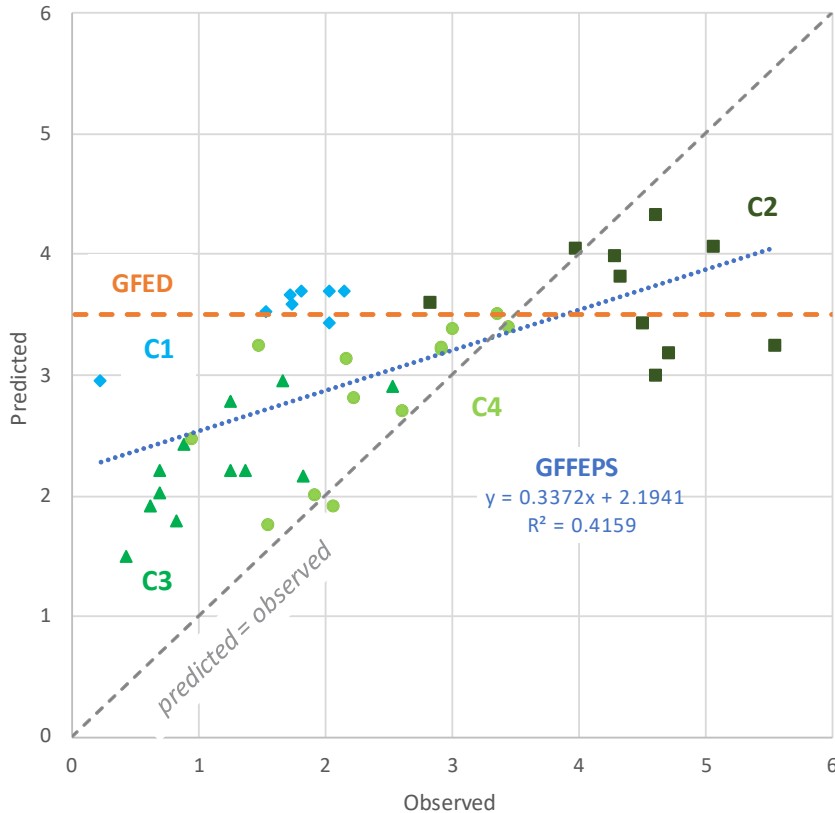

**Figure A.1. Observed total fuel consumption vs GFFEPS predictions for Boreal coniferous forests assuming all fuel as C-2. Points coloured to reflect the fuel type from each study. The constant value of GFED predictions (3.5 kg/m²) is shown as an orange dashed line.**

Using a fuel load of 69 t ha⁻¹ with a combustion completeness of 51%, GFED predicts a fixed fuel consumption 3.5 kg m⁻² for Boreal Forest, regardless of season, and does not distinguish between conifer and deciduous (van Leeuwen et at., 2014).

**A.1.2 Deciduous**

Deciduous stands in the Boreal Forest are represented by aspen in the CFFDRS. Quintillio et al. (1991) documented
spring fires in leafless aspen stands in central Alberta. Note that one reported burn was removed from this comparison. As the authors wrote:

"Two of the plots were jointly reburned, and, among other data, a 10-fold increase in fire intensity was documented, due largely to aspen mortality in 1972 and the subsequent increase in fuel load."

These two reburned plots (their 3b&c) were reported as a single data point with fuel consumption of 3.402 kg m$^{-2}$, which exceeded the default fuel load of 15 t ha$^{-1}$ (1.5 kg m$^{-2}$). The frontal fire intensity of this fire was 57,261 kW/m. Including this point would skew the regressions and thus were removed.

Using the original study results (less the removed plots), observed fuel consumptions were compared to those predicted by GFFEPS. The default FBP surface fuel load of 15 t ha$^{-1}$ was used in the GFFEPS calculations. The Growing Season Index (*GSI*) was then introduced as a modifier to the predicted fuel consumption, with *GSI* values ranging from 0.0 to 0.55 with an average of 0.18 (see Eq. (9) under 4.3 Combustion Completeness).

Figure A.2. shows the scatter plot of observed fuel consumption versus that predicted by GFFEPS. Including *GSI* in the calculations changed the regression from a negative correlation ($r^2 = 0.037$) to positive ($r^2 = 0.221$).

There is no clear fuel type in GFED that represents North American aspen forests (van Leeuwen et at., 2014). They report a fuel consumption of 3.5 kg m$^{-2}$ for the Boreal Forest, and 5.8 kg m$^{-2}$ for the temperate forest (fuel load of 115 t ha$^{-1}$ and combustion completeness of 61%), both of these values exceed all observed values in Quintillio et al. (1991).

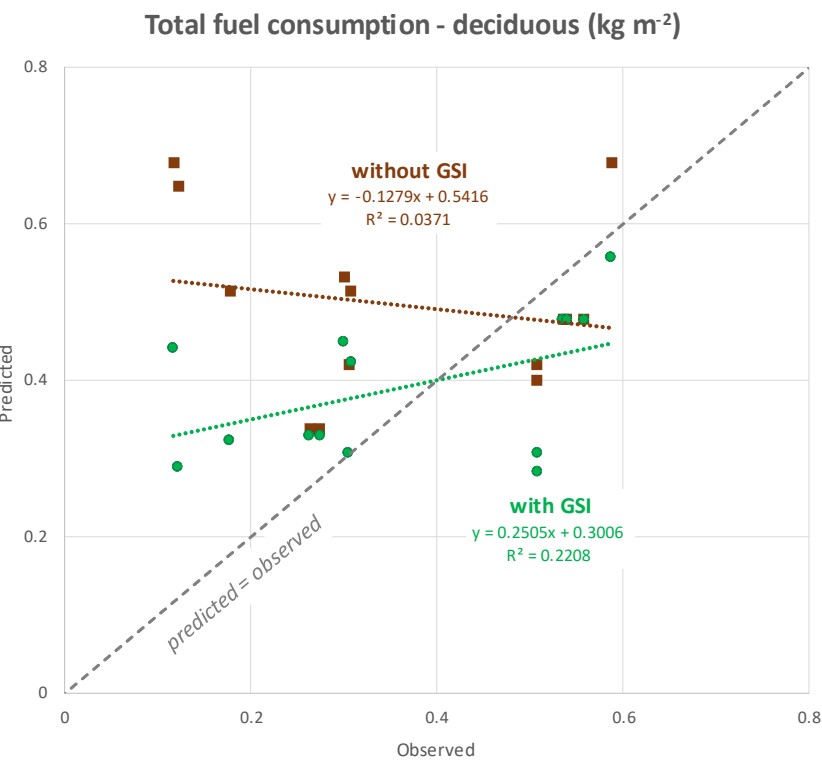

**Figure A.2. Observed total fuel consumption vs GFFEPS predictions for Boreal deciduous forests. Green and brown indicates the use of *GSI* as a modifier for greenup in fuel consumption calculations. Dotted lines show regressions through respective datasets.**

Note that as the only deciduous fuel type in the FBP system, D-1/2 (leafless/ leafed) aspen fuel type was used globally to represent a number of *broadleaved* land cover types in GLC2000 used by GFFEPS. Fuel loads and greenness varied between regions and classifications.

### A.1.3 Siberia

McRae et al. (2006) studied fire behaviour in Scotch Pine forests in central Siberia. Following the same methodology as Canadian forests, study results were compared to predictions based on GFFEPS. Foliar moisture content (*FMC*) equations developed for Eurasia were used as described in the manuscript (Chapter 3.4.2).

The reported results were compared to each of the seven FBP coniferous fuel types as well as the M-3/4 - Dead Balsam Fir mixedwood fuel type. Table A.2 summarizes the regression results. Immature jack pine (C-4) provided the best fit to the data ($r^2$ = 0.921) while mature jack pine (C-3) provided the fit closest to unity ($a$ = 1.036) and C-2 was closest to intercepting the origin ($b$ = 0.165). Figure A.3 shows scatter plots of the study data against GFFEPS predictions using fuel types with the best results.

**Table A.2. Summary of correlation results of study-based observed fuel consumptions in Scotch pine versus GFFEPS predictions using various FBP fuel types. Best fits per column are shown in bold.**

|  | Surface fuel load (kg/m$^2$) | Crown fuel load (kg/m$^2$) | $r^2$ | $a$ (slope) | $b$ (intercept) |
|---|---|---|---|---|---|
| C-1 | 1.5 | 0.75 | 0.300 | 0.351 | 0.51 |
| C-2 | 5 | 0.8 | 0.852 | 0.846 | **0.165** |
| C-3 | 5 | 1.15 | 0.873 | **1.036** | 0.449 |
| C-4 | 5 | 1.2 | **0.922** | 1.28 | -0.763 |
| C5 | 5 | 1.2 | 0.894 | 0.731 | -0.337 |
| C6 | 5 | 1.8 | 0.894 | 0.731 | -0.337 |
| C7 | 1.75* | 0.5 | 0.620 | 0.469 | 1.638 |
| M-3/4 | 5 | 0.8 | 0.805 | 0.704 | 0.787 |

*Surface fuel load of C7 is a blend of forest floor (2.0 kg m$^{-2}$) and woody fuel loads (1.5 kg m$^{-2}$).

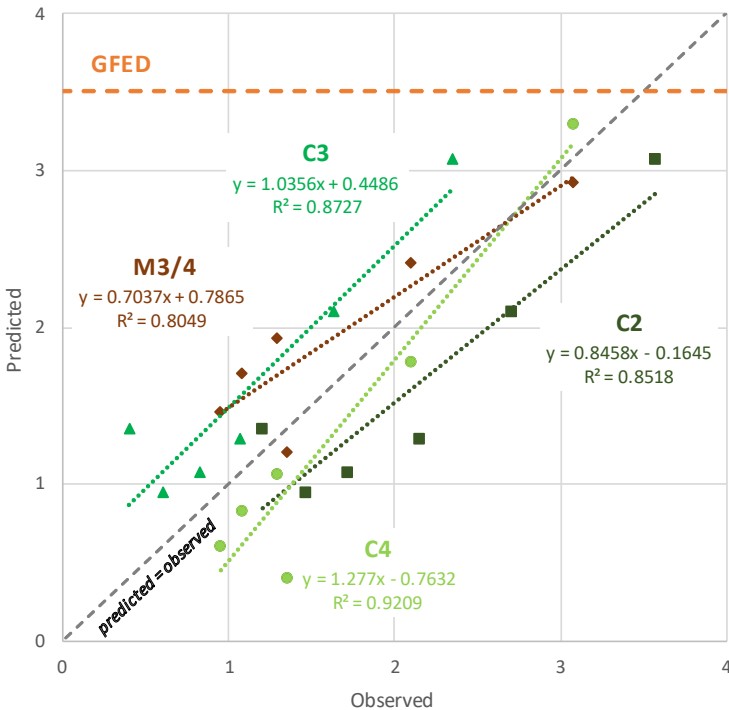

**Figure A.3. Observed total fuel consumption for central Siberian Scotch pine forest vs GFFEPS predictions using various FBP fuel types. Dotted lines indicate regression lines for respective fuel types. The constant value of GFED predictions (3.5 kg/m$^2$) is shown as an orange dashed line.**

It is expected that Scotch pine forests are best represented by the FBP mature and immature pine fuels found in Canada. With that said, Siberia, like Canada, is covered by a wide variety of coniferous and deciduous forests. A large component of these are larch forests that lose their needles every winter. No studies were found for comparative purposes.

For GFFEPS purposes, a C-2 – boreal spruce fuel type was used for *tree cover, needle-leaved, evergreen* in northern

Eurasia, northeastern Europe and North America; C-3 – mature jack pine in remaining areas. Fuel loads varied between regions.

**A.2 Tropical Forest**

A validation of model calculations against original source material was conducted for tropical fires in the Amazon. Source materials used were readily available papers referenced by van Leeuwen et al., 2014 (Carvalho et al., 1995; Fearnside

et al., 1993; Fearnside et al., 2001; Guild et al., 1998; Kauffman et al., 1993; Kauffman et al., 1998; Ward et al., 1992). Fires in these studies were all land clearing, conducted for agricultural use. Trees were typically felled at the onset of the May-

September dry season and burned at the end of the dry season. Natural fires in uncleared lands in the Amazon are rare (but are now increasing) and when they occur, they burn in the understorey, likely undetected by remote sensing (Withey et al., 2018).

The most representative classification of tropical rainforest in the GLC2000 land classification categories is *Tree cover, broadleaved evergreen.* Sampling the fire locations on the GLC2000 spatial dataset:

- 8 fires occurred in *Tree Cover, broadleaved, evergreen* (Carvalho et al., 1995; Fearnside et al., 1993; Fearnside et al., 2001; Guild et al., 1998; Kauffman et al., 1998)

- 2 fires occurred near *Tree Cover, broadleaved, evergreen* (Kauffman et al., 1993; Ward et al., 1992),

• 3 fires occurred in *Bare Areas*, but described in the text as 12-year regrowth after slash-and-burn (Kauffman et al., 1993),

- 4 southern fires in *Herbaceous Cover, closed-open* were described as savanna and left out of analysis (Ward et al., 1992),

where near is defined as having an adjacent cell categorized as *Tree Cover, broadleaved, evergreen* on the 1km resolution
dataset.

    The D-1 - Leafless Aspen FBP fuel type was used for downed trees (hence, greenup was deemed unnecessary). Various slash fuels in the FBP system were also examined but did not improve on the following results.

    Figure A.4 shows the scatter plot of observed total fuel consumption versus that predicted by GFFEPS. Points have been colour-coded based on their general land classification. Including all data points produces a poor correlation ($r^2 = 0.04$)
but by removing the outliers associated with burns after recent regrowth and those classified as near, but not within, broadleaf evergreen, increase the correlation to $r^2 = 0.732$.

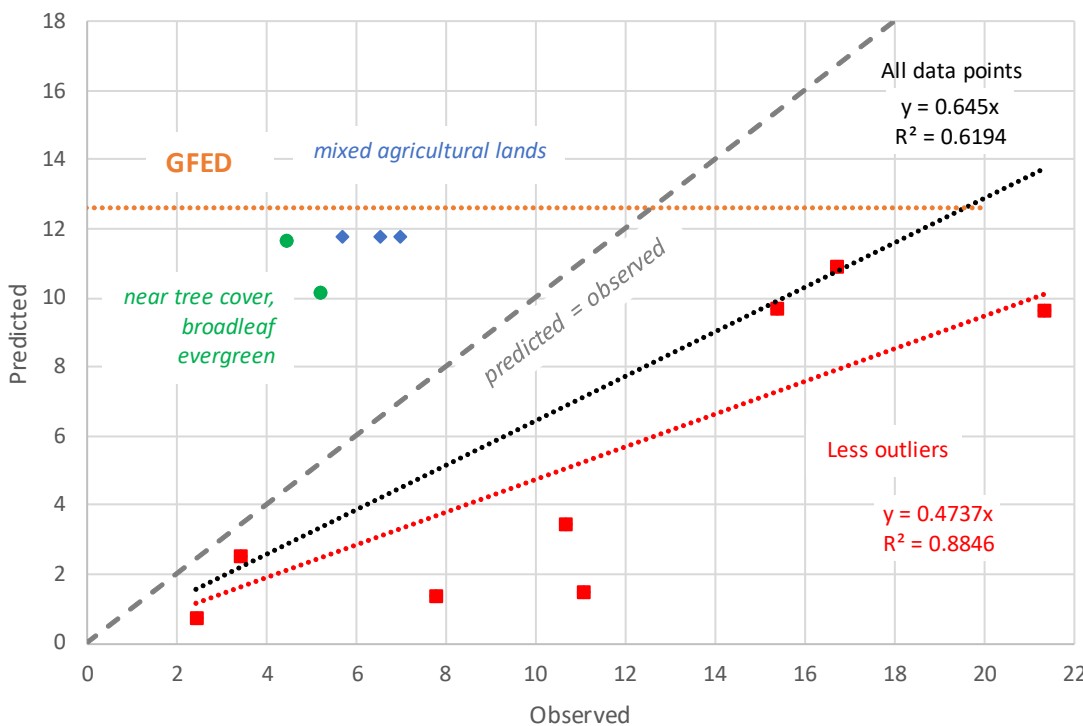

**Figure A.4. Observed total fuel consumption vs GFFEPS predictions for tropical forests assuming all fuel as D-1. Points coloured to reflect the general land classification from each study. The constant value of GFED predictions (12.6 kg/m²) is shown as an orange dashed line.**

For tropical forests, GFED uses a fuel load of 285 t ha⁻¹ and a combustion completeness of 49%, yielding a constant fuel consumption of 12.6 kg m⁻². Fuel loads for GFFEPS were calculated following data collected by van Leeuwen et al. but heavier fuels (20.5 cm diameter) were left out (assumed to be uncombusted) to give a fuel load of 117.9 t ha⁻¹. Adjusting this value by a bias correction of 155%, the fuel load becomes to 182.8 t ha⁻¹. The bias correction was based on a decision to include all points. This was made to avoid extreme overpredictions in the fringe areas, in this case representing 5 of the 13 points. All points covered site characteristics inconsistent over the eight published reports and while some studies produced outliers, their overall results were deemed valuable.

The Buildup Index (*BUI*) of the FWI system was compared directly to percent fuel consumed as shown in Fig. A.5. This supports the weather-based approach used by GFFEPS. Lower consumption (<60%) in *tree cover, broadleaved, evergreen* supports excluding heavier fuels from the analysis.

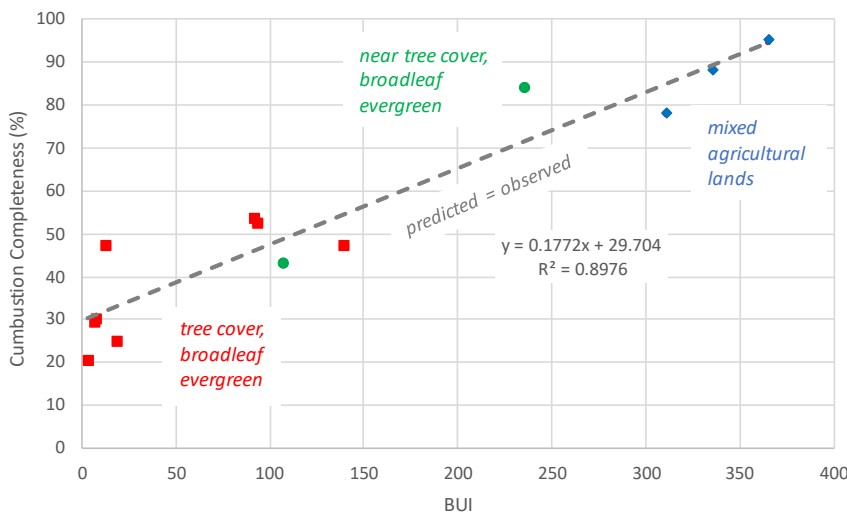

**Figure A.5. Observed Buildup Index (*BUI*) and combustion completeness at published fire sites. Points coloured to reflect the general land classification from each study.**

**A.3 Tropical Peat**

Field et al. (2004) studied air quality in western Indonesia using the Drought Code (*DC*) to predict visibility. In their study, a nonlinear regression model was developed relating visibility and *DC*. Based on their model, a logistic model for fuel consumption, *FC* (kg m$^{-2}$), was built using their point of inflection (*DC* = 551) and shape scale controlling the curvature (*S* = 123.7)

$$FC = 105.6 \,/(1 + e^{\frac{551-DC}{123.7}})$$
(A.1)

where 105.6 kg m$^{-2}$ (1056 t ha$^{-1}$) is the fuel load from van Leeuwen et al. (2014) for tropical peat.

2015 was an exceptional year for smoke emissions in the region. Kaiser et al. (2016) estimated that over 15% of 2015 global emissions were from fires in tropical Asia. To examine this, hotspots were collected between 0° and 4°S latitude and 965 112°E and 116°E longitude for 2015. Fuel consumption based on our logistic model was calculated using the daily average *DC* values of these hotspots (based on the GEM model FWI as described in the manuscript), which ranged from 5.15 to 458.1 and averaged 116.3.

Figure A.6 shows a comparison of daily hotspots and calculated fuel consumption. Both show peak activity in the fall, though the predicted fuel consumption spread is wider than the principal hotspot activity. A background fuel consumption of 970 1.213 kg m$^{-2}$ results when *DC* = 0. This could be removed in the future but in the absence of hotspots, this may be immaterial.

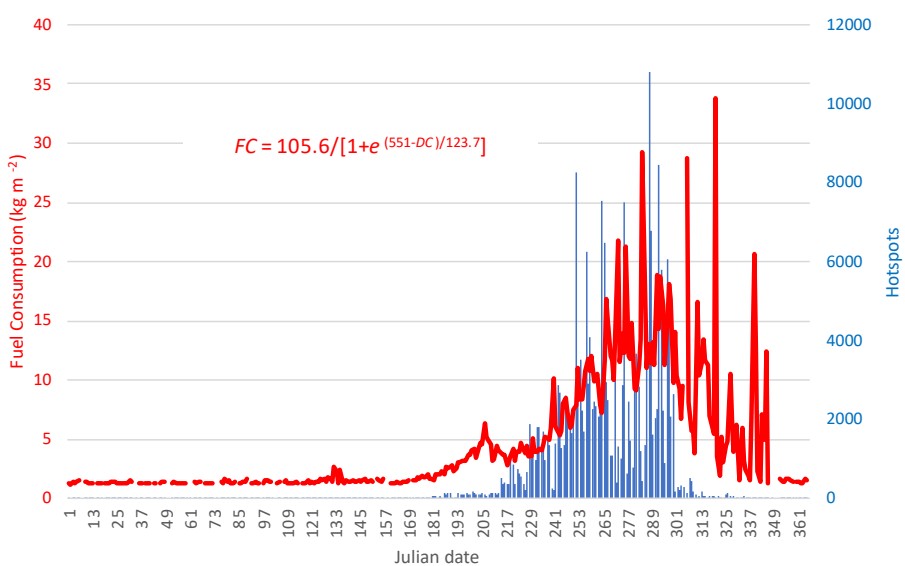

**Figure A.6. Fire characteristics in southern Kalimantan for 2015. Hotspots (blue) represent the daily number of hotspots observed between 0º and 4ºS latitude and 112ºE and 116ºE longitude. Fuel consumption (red) based on logistic model and average DC values of hotspots occurring in the region.**

Graham et al. (2022) evaluated fire behaviour in drained tropical peatlands, examining smouldering peat fires at five locations in Kalimantan during August and September, 2015. This provided data to validate our logistic model. Fuel consumption was calculated using *DC*s from the reanalysis data (McElhinny et al. 2020) with all five locations occurring in the same reanalysis grid cell. Choosing representative *DC* values was an issue as a precipitation event appears to have occurred, as on August 28, 2015 the *DC* dropped from 443 to 123 in the reanalysis data. This was not noted by Graham et al. and may not have happened at any of the study sites. To test the impact of this event, the adjacent reanalysis cell to the east where the precipitation did not occur was included for comparison. A second alternative was used based on the daily average *DC* values for hotspots occurring in the study area (between 2.2064 ºS and 2.5226ºS latitude and between 114.39 ºE and 114.63175 ºE longitude) based on an ECCC GEM-MACH model run. These values ranged from 147 to 291, which were higher than average *DC* of 116.3 for 2015.

Figure A.7 shows a scatter plot of the study data versus GFFEPS predictions. Fuel consumption based on the reanalysis data produced a negative trend, while results based on the reanalysis cell to the east produced consumption values 3 to 5 times higher than those using the average *DC*s of the hotspot in the area. Linear regressions of the latter two produced correlation coefficients ($r^2$) of 0.801 and 0.822, suggesting GFFEPS performed well for this tropical peatland location (given its few data points).

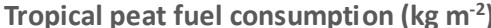

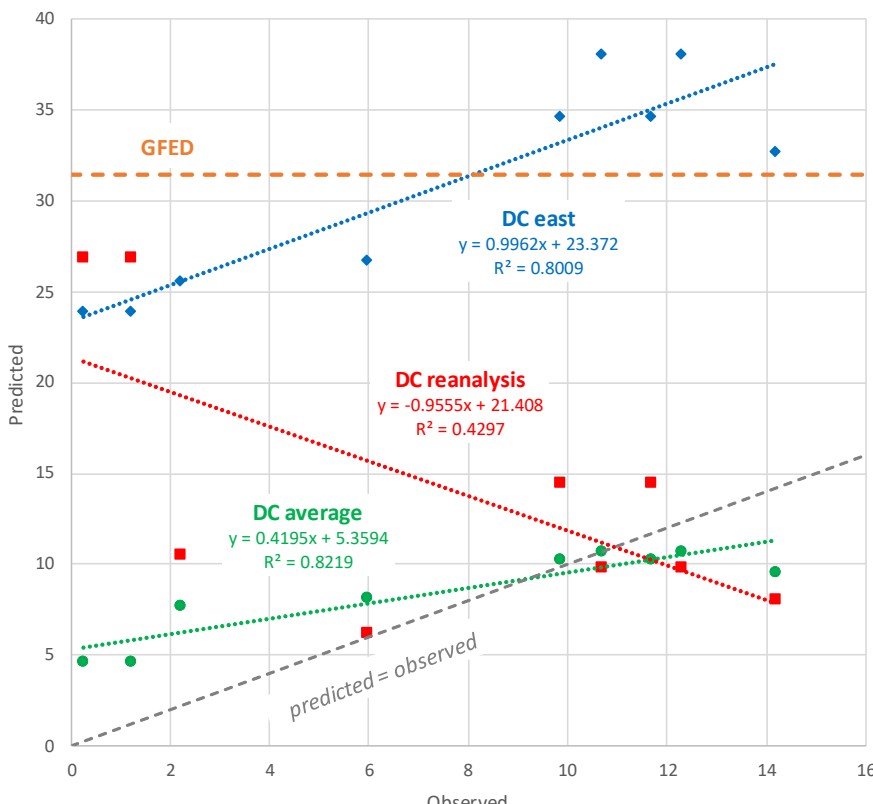

**Figure A.7. Observed fuel consumption in tropical peatlands in southern Kalimantan, Indonesia vs GFFEPS predictions. Predicted consumption for the nine data points using the reanalysis DC data (red), using DCs from the reanalysis grid cell 27 km due east (blue) and using daily DCs averaged from the hotspots occurring in the study area (green). The constant value of GFED predictions (31.4 kg/m²) is shown as an orange dashed line.**

For tropical peatland, GFED uses a fuel load of 1056 t ha$^{-1}$, combustion completeness of 27%, yielding a fuel consumption rate of 31.4 kg m$^{-2}$. This value is close to consumption rates observed east of the study.

The observed values were highly variable and this was acknowledged by Graham et al. (2022). The August 28 precipitation event played a significant role, as shown by the data. The *DC* average likely shows the general impact of precipitation on the sites while *DC* east shows the conditions without. In the *DC* reanalysis results, the two outlying points with low observed values (<2 kg m$^{-2}$) and high predicted values (>25 kg m$^{-2}$) may reflect a discontinuity in timing the transition from dry to wet conditions. This is certainly a possibility given these points were from one site sampled on August 20. It is possible that the site received precipitation prior to the August 28 event, yet without on-site weather observations, this is only speculation.

In terms of GFFEPS validation, it appears the predicted values of the *DC* average follow the observed data closely, with a correlation of 0.8219. The dry conditions shown by *DC* east match well with the GFED value but that may be due to the common fuel load value used by both models. Unfortunately, there is no reported precipitation data to be certain as to what happened at the study site. Closer examination of tropical peat fires is in order but such studies are not available in the current literature.

**A.4 Wooded and Open Savanna Grasslands**

Savanna fires were examined based on original work by Hoffa et al. (1999), Shea et al. (1996) and de Castro and Kauffman (1998) as referenced and used in van Leeuwen et al. (2014). Hoffa et al. (1999) studied 13 prescribed burns conducted in the early dry season (June to August) in Kaoma Local Forest 310, western Zambia (14°52'S, 24°49'E); as part of the South African Fire-Atmosphere Research Initiative (SAFARI) project, Shea et al. (1996) documented 10 fires in Kruger National Park, South Africa (31°14'00"E, 25°15'13"S), three fires in Kasanka National Park , Zambia (12°35'S, 30°21'E) and one near Choma, Zambia (16°50'S, 26°59'E) ; and de Castro and Kauffman 1998 examined fires in the Brazilian Cerrado, a mosaic of savanna and forests near Brasilia, at the *Reserva Ecológica do Instituto Brasileiro de Geografía e Estatística* (IBGE) and the *Jardim Botânico de Brasília* (JBB) (15°51'S, 47°63'W).

Dambo is an African grassland, seasonally flooded during the rainy season. It occupies 10% of Zambia. Miombo is an open-canopy, semideciduous woodland with a grass and shrub understory. It covers 12% of Africa and 80% of Zambia. In Shea et al. (1996) 12 burns were conducted in dambo grasslands, 2 in miombo woodlands; in Hoffa et al. (1999), 7 burns were conducted in dambo and 6 burns in Miombo. The four Cerrado sites in de Castro and Kauffman (1998) were conducted across a range of densities: *campo limpo* (pure grassland), *campo sujo* (a savanna with a sparse presence of shrubs), and two variants of Cerrado *sensu stricto* (a dominance of trees with scattered shrubs and a grass understorey).

Grass curing, a measure of percent dead/dormant/dry as opposed to live/growing/green grass, is a driving factor in the rate of spread in grass fuels in the FBP system. The system assumes complete consumption of grass fuels – a generalization made by those who developed the system (see 6. Discussion). An alternative approach used by GFFEPS is that grass fuel consumption is related to grass curing following the same relationship as used for rate of spread. Grass typically follows a seasonal pattern of growth during the spring (or rainy season) followed by drying and mortality during the summer (or dry season). Figure A.8 shows the relationship of grass curing (reported as % dormancy) at the burn sites in the three publications and the *DC* from the FWI system as interpreted from the global re-analysis of fire weather conditions (McElhinny et al. 2020). A power law relationship was derived with a correlation of 0.2515. *GSI* was considered as a possible predictor to grass curing but the correlation was negligible in these studies.

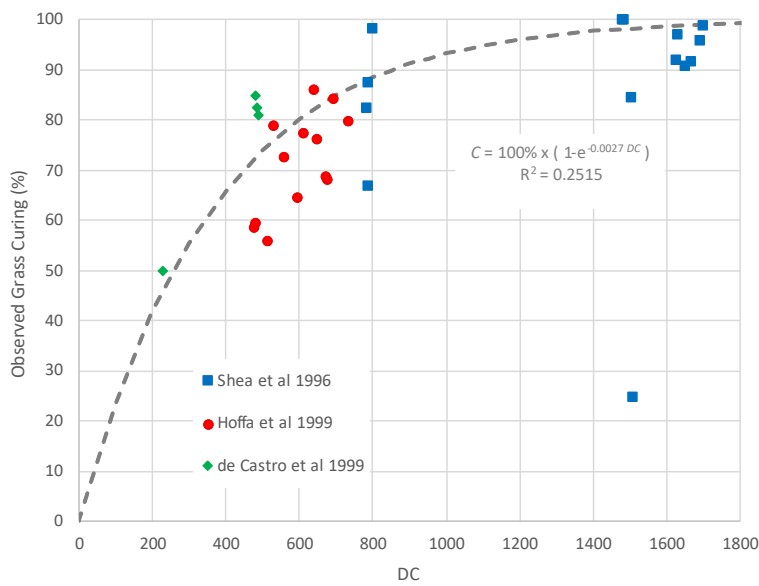

**Figure A.8 The relationship of grass curing (% dormancy) to Drought Code (*DC*) observed at the sites in the three publications.**

Figure A.9 shows the scatter plot of observed total fuel consumption versus that predicted by GFFEPS. Following GFFEPS methodology, dambo grassland savanna was assigned a standing grass open fuel type (O-1B) with an average total fuel load of 4.0 t ha$^{-1}$ based on the average total biomass reported in Hoffa et al. (1999) and Shea et al. (1996). Fuel consumption was calculated as the product of the grass fuel load and the percent curing. Miombo woody savanna was assigned a leafless aspen fuel type (D-1), given the predominance of down and dead fuels. Fuel load of 9.2 t ha$^{-1}$ was used based on the average total fuel loads. The *DC*s required for grass curing and *BUI*s required for D-1 calculations were based on McElhinny *et al.* (2020) global reanalysis (with overwintering). Correlation values ($r^2$) were 0.312 for dambo grassland and 0.673 for miombo woodland though both were far from the line of equality.

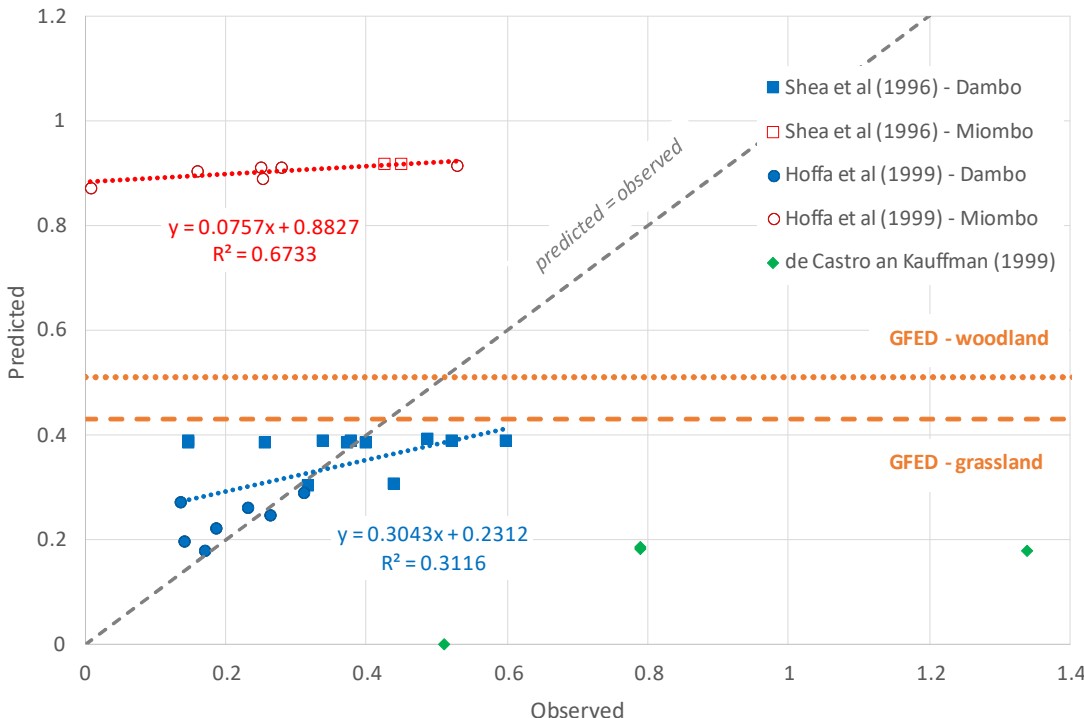

**Figure A.9. Observed fuel consumption in dambo grasslands and in miombo woodlands compared to predictions. Three separate studies are shown. The constant values of GFED predictions (0.43 and 0.51 kg/m²) are shown as orange dashed and dotted lines.**

Data from de Castro and Kauffman (1998) was intentionally left out of calculations given the broad range of site descriptions. Also, two outliers (due to their heavier fuel loads) tended to dominate and influence the correlations. Their points are shown on the graphs for comparative purposes.

An alternative approach was conducted, calculating grass fuel consumption and surface (non-grass) fuel consumption separately and then combining these afterwards. In dambo landscapes, the average grass fuel load was 2.18 t ha⁻¹ and surface fuel load 1.83 t ha⁻¹. In miombo, the average grass fuel load was 1.06 t ha⁻¹ and the surface fuel load 8.13 t ha⁻¹. While this approach improved the correlations, the separation from the line of equality remained (not shown). To better match to the average fuel consumption values, the fuel loads were adjusted to correct for the bias, as shown in Figure A.10, bringing the predictions in line with the observed values. Correlation values ($r^2$) were 0.330 for dambo grassland and 0.709 for miombo woodland.

The GFED model describes dambo as grassland savanna and uses a 5.3 t ha$^{-1}$ fuel load with an 81% combustion completeness resulting in 0.43 kg m$^{-2}$ fuel consumption. It describes miombo as woody savanna with a 11 t ha$^{-1}$ fuel load, 58% combustion completeness and 0.51 kg m$^{-2}$ fuel consumption. These relations are shown as horizontal lines of constant prediction for comparison purposes. Admittedly, the GFFEPS predictions are a modest improvement over the constant values for the GFED predictions, but this is a result of the high variability of the fuel loads in the source material. It does indicate GFED predictions are 10 to 20% or more higher than GFFEPS. Given the frequency of fire on the African savanna, such a difference would amount to substantially higher emissions in GFED predictions. Finally, we note the need for further studies of fire behaviour over a wider range of conditions in this region, in order to evaluate weather-based models such as ours.

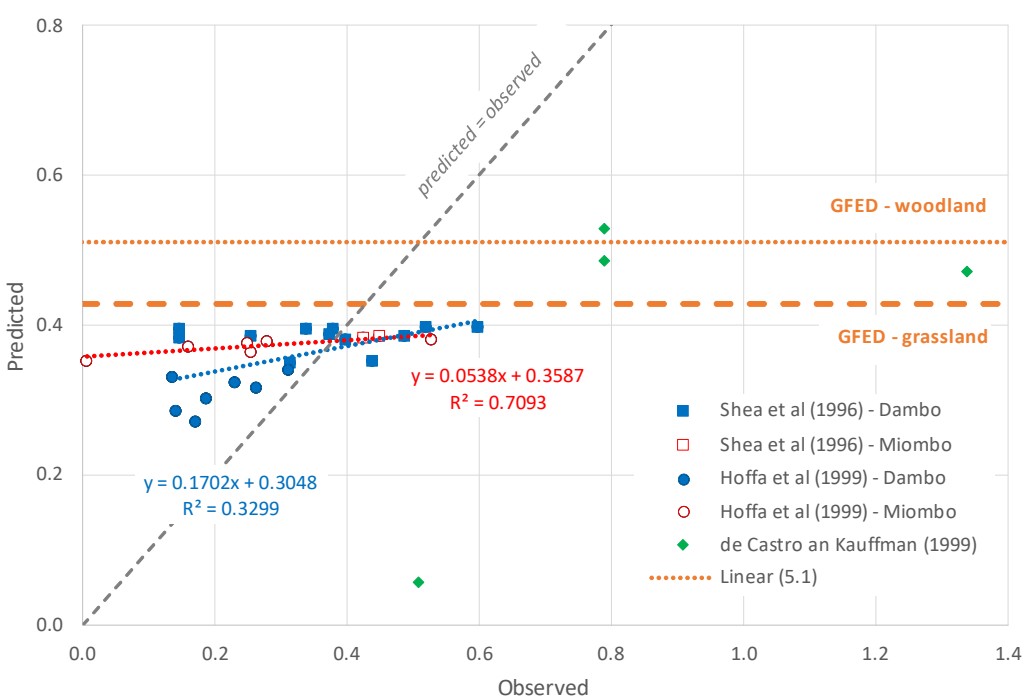

**Figure A.10. Observed fuel consumption in dambo grasslands and in miombo woodlands compared to predictions using the modified fuel consumption approach. Three separate studies are shown. The constant values of GFED predictions (0.43 and 0.51 kg/m²) are shown as an orange dashed and dotted lines.**

**A.5 Australia Eucalypt Forests**

Over 22% of Australia is forested, of which 78% is Eucalypt (Sullivan et al. 2012). Eucalypt (Jarrah) does not fit the typical fire behaviour reflected in the Canadian system so an effort was made to create a fuel consumption model specific to eucalypt from the published literature. In 1983, Australian agencies conducted the Aquarius project. This project studied a

number of aspects of fire in dry eucalypt forests, including fire behaviour, fire line productivity and workers' safety and health (Budd et al. 1997).

Hollis et al. (2010) summarized woody fuel consumption in eucalypt fires for 18 of the 32 fires of project Aquarius (among other fires) at McCorkhill forest block (33°56′38"S, 115°31′52"E as reported in Burrows et al, 2019).  Dates for these fires were collected from Cheney et al. (2012) and from Gould (pers. comm. 2022). *BUI*s were then ascertained from 1983 re-analysis data (McElhinny et al. 2020).  Sigmoidal curves similar in structure to those used in the Canadian Forest Fire Behaviour Prediction (FBP) System were used.  An upper limit of 90% was used as it was assumed that standing snags would

likely be left after a fire-front passage.  This is supported by the highest reported observation in the Aquarius studies.  As sigmoid curve fitting is inexact, four models were constructed based on successive power increments and a minimization of the sum of residuals.  Figure A.11 shows the chosen, resulting curve.  The choice of best model fit is speculative, given the spread of the data and the closeness of the curves.  Total fuel consumption for eucalypt is achieved by multiplying combustion completeness by a eucalypt fuel load of 7.8 kg m$^{-2}$ as used in GFFEPS (Sullivan et al., 2012).

**Figure A.11. Consumption completeness (%) in dry eucalypt forests based on Project Aquarius observations.  A linear regression through the origin shown as a dashed line.  A sigmoidal curve shows the chosen fit based on successive power increments. The constant value of GFED predictions (68.1%) is shown as an orange dashed**

**line.**

For eucalyptus, GFED reports an average combustion completeness of 68.1% (shown on the figure) and fuel consumption of 7.9 kg m$^{-2}$.

It is worth noting that the fire sites reported in Project Aquarius reflect the coarse woody debris left from forest management practices. This is evident in some of the other sites reported by Hollis et al. (2010), with pre-fire woody fuel loads in excess of 1000 t ha$^{-1}$ at Warra, Tasmania – a wet eucalypt forest site. Sullivan et al. (2012) reports a typical fuel load of 78 t ha$^{-1}$ in Jarrah (tall understorey), matching the average of all dry eucalypt sites in Hollis et al. (2010). This value was then assumed for all Australian forests.

**Appendix B. Sensitivity Analysis**

A sensitivity analysis was conducted to test the extent to which input parameters and methodologies used by GFFEPS affects the output emissions estimates. The analysis focused on three factors: land cover maps, agricultural burning and daily weather. Each of these specific factors was examined separately while maintaining the integrity of the remaining GFFEPS calculations. Results are presented as total smoke emissions, which are twice the carbon emissions (500 g kg$^{-1}$).

**B.1 Land Cover**

The GFFEPS model, as presented in this study, uses the GLC2000 dataset as land cover classification system. The decision to use GLC2000 was made in the early stages of GFFEPS model development. We needed a global land use of sufficient resolution that was easy to employ and GLC2000 was well suited for this purpose, providing a single map global coverage at a 1-km resolution. An important benefit of using the GLC2000 was the national expertise and ground truthing involved in the generation of that dataset. While the GLC2000 dataset is now 25 years old, this was seen as less critical as vegetation rarely changes (deciduous forests rarely change into coniferous) and most subsequent changes, whether they were a result of disturbance (fires, deforestation) or urbanization, would result in landscapes less fire prone - and this would be reflected by a reduced number of hotspots in these areas. For example, there should be fewer hotspots (if any) appearing in a burn scar. Consequently, the potential for post-2000 land changes to significantly affect model output is reduced, despite the 25-year age of GLC2000.

However, to confirm this hypothesis, a test was conducted, comparing GFFEPS model predicted smoke emissions for 2019 using the GLC2000 land cover scheme against predicted emissions instead using the Moderate Resolution Imaging Spectroradiometer (MODIS) Land Cover Type (MCD12C1) Version 6. The MODIS dataset is a product of the USGS presenting land cover at a 0.05 degree (5,600 m) spatial resolution. It is produced annually is a spatially aggregated and reprojected version of the tiled MCD12Q1 Version 6 (500 m) data product. Both follow the International Geosphere–Biosphere Programme (IGBP) for its land classifications. The MODIS dataset thus is less likely to be subject to age-of-dataset issues.

Implementing the IGBP land classification in the GFFEPS model was achieved by matching IGBP land classification categories (as provided in the MCD12C1 map product) to GLC2000 categories. A cross tabulation of IGBP versus GLC2000 land classification occurrences as reported in the daily observed hotspot data was used to find matching classifications. Observation dates selected were January 1, April 1, July 1 and October 1, 2019 (40,227, 57,639, 68,824 and 53,350 hotspots respectively) to account for any seasonal variation. Table B.1 shows the matching IGBP and GLC2000 land classifications achieved looking at the entire set of 220,040 hotspots, globally, for the four days. However, issues with this initial assessment were discovered. For example, the boreal forest, primarily a coniferous forest, was largely described by the MODIS data set as *Woody savannas* and thus initially matched with *Tree Cover, broadleaved, deciduous, closed* in GLC2000, a description more typical in Africa. This was rectified by conducting cross tabulation for each of the 18 geographic regions in the GLC2000 data set (not shown in the table). Subsequently, the GFFEPS model was run, sampling the 2019 MCD12C1 land cover category at each detected hotspot and replacing it with a regional matched GLC2000 land classification. Results were then compared to the original GFFEPS results. In doing so, the spatial representation of the MCD12C1 is captured while maintaining the fuel and fire behaviour associated with GCL2000 land classification categories.

**Table B.1 Matching IGBP and GLC2000 land classifications globally (regional specific matches may differ).**

| IGBP | Description | GLC2000 | Description |
|------|-------------|---------|-------------|
| 1 | Evergreen Needleleaf Forests | 4 | Tree Cover, needle-leaved, evergreen |
| 2 | Evergreen Broadleaf Forests | 1 | Tree Cover, broadleaved, evergreen |
| 3 | Deciduous Needleleaf Forests | 5 | Tree Cover, needle-leaved, deciduous |
| 4 | Deciduous Broadleaf Forests | 2 | Tree Cover, broadleaved, deciduous, closed |
| 5 | Mixed Forests: | | |
| | outside Africa | 6 | Tree Cover, mixed leaf type |
| | inside Africa | 2 | Tree Cover, broadleaved, deciduous, closed |
| 6 | Closed Shrublands | 12 | Shrub Cover, closed-open, deciduous |
| 7 | Open Shrublands | 14 | Sparse Herbaceous or sparse Shrub Cover |
| 8 | Woody Savannas | 2 | Tree Cover, broadleaved, deciduous, closed |
| 9 | Savannas | 3 | Tree Cover, broadleaved, deciduous, open |
| 10 | Grasslands | 12 | Shrub Cover, closed-open, deciduous |
| 11 | Permanent Wetlands | 15 | Regularly flooded Shrub and/or Herbaceous Cover |
| 12 | Croplands | 16 | Cultivated and managed areas |
| 13 | Urban and Built-up Lands | 22 | Artificial surfaces and associated areas |
| 14 | Cropland/Natural Vegetation Mosaics | 17 | Mosaic: Cropland/Tree Cover /Other natural vegetation |
| 15 | Permanent Snow and Ice | 21 | Snow and Ice |
| 16 | Barren | 19 | Bare Areas |

Figures B.1 and B.2 present the resulting daily values of global emissions shown as a time series and as a scatter plot, respectively.   The time series shows a similar pattern for the two models with GLC2000 predicting lower values than

MCD12C1 in the winter and higher values in the summer.   The scatter plot shows near equality between the two model predictions (a slope of 0.98) when forced through the origin, with an $r^2$ of 0.93. Total annual emissions were 2,957 and 3,028 Mt as predicted by GLC2000 and MCD12C1 respectively.   That is, on a global basis, the relative impact of the updated land use information is relatively small.

A factor contributing to the residual differences would be the data resolution.   The MCD12C1 has a 0.05 degree (~5.6

1150    km) spatial resolution, while GLC2000 has a 1 km resolution.   This suggests 31 GLC2000 cells would occur in each MCD12C1.   Spatial aggregation may thus account for some of the variation.

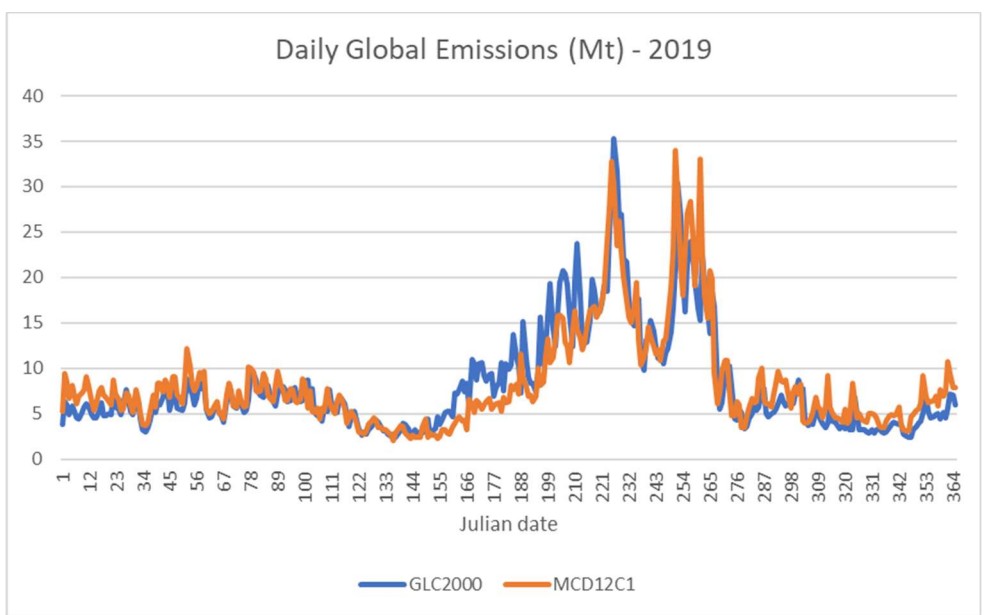

**Figure B.1. Time series of daily global emissions for 2019 using the GLC2000 versus the MODIS MCD12C1 land**
**classification.**

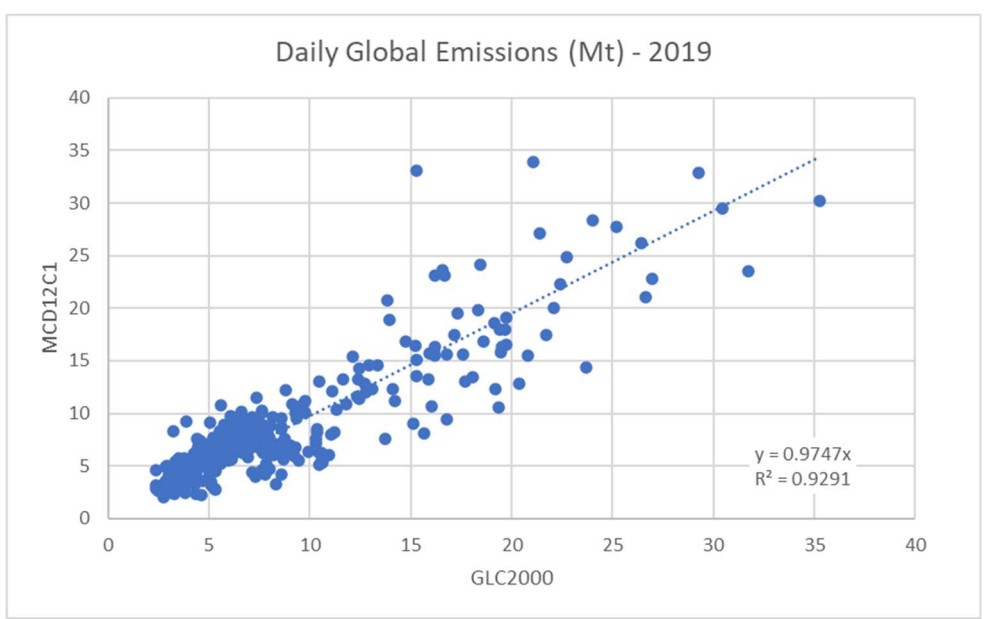

**Figure B.2. Scatter plot of daily global emissions for 2019 using the GLC2000 versus the MODIS MCD12C1 land classification.**

Figure B.3 shows the annual total emission values regionally, where GFFEPS differences associated with the two land use datasets become more apparent.  Largest differences occurred in EQAS, NHAF and BONA, where GLC2000 predictions were 61%, 65% and 67% of those for MCD12C1, while in SHAF GLC2000 predictions where 166% of those for MCD12C1.  These differences are likely to poor matching of coniferous versus deciduous forests, a distinction not captured in MCD12C1 classifications Savannas and Woody savannas (as previously described).  The difference between coniferous and deciduous

fuels is critical in the FBP fire behaviour calculations and any misclassification would have an impact on predictions.  Also, difficulties mapping fire emissions and land classifications in Africa have been discussed in various papers (Ramo et al., 2021; Nguyen and Wooster, 2020; Zhang et al. 2018), possibly accounting for the discrepancy shown in this comparison.

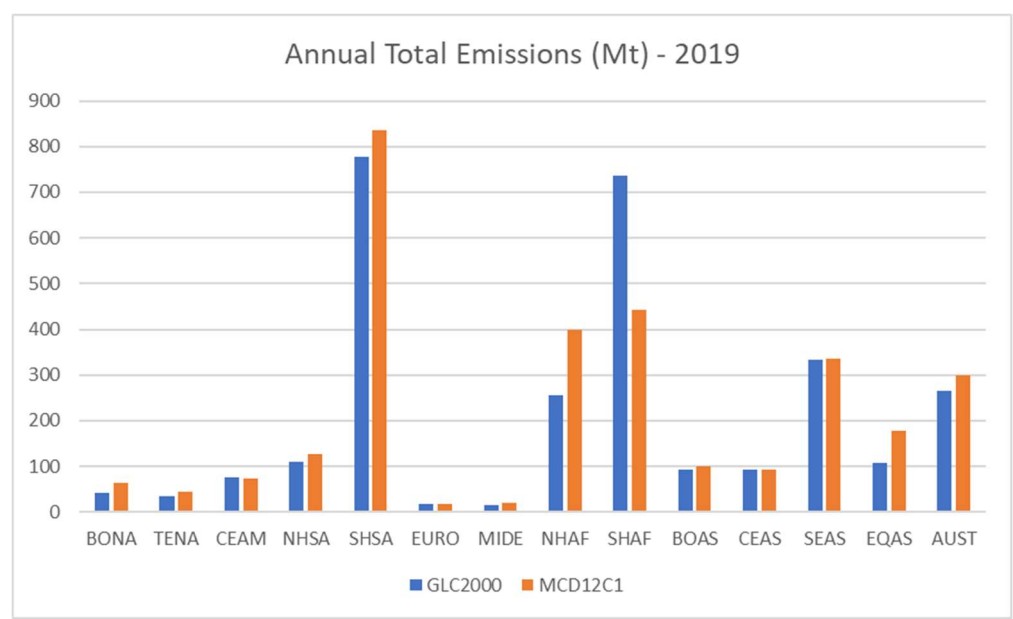

**Figure B.3. Regional annual emissions for 2019 using the GLC2000 versus the MODIS MCD12C1 land classification.**

### B.2 Agriculture

The sensitivity of the GFFEPS model to agricultural burning and small fires was examined. As presented in chapter 3.2, our approach used FAO agriculture burning statistics to predict emissions in cultivated zones. Using national annual values of biomass of residual crops burned divided by the number of hotspots that occurred per nation per year, a historical average biomass burned per hotspot was determined. This was then applied to future, observed hotspots to predict biomass burned from agricultural burning. The benefit of this method is that national statistics as reported to the FAO should account for all biomass burned, including that from small fires, which are undetected by satellite observation.

The sensitivity of the FOA approach within GFFEPS was assessed by replacing the FAO agricultural burning with grassland fires at a fixed grass fuel load (*GFL*) of 0.60 kg m$^{-2}$, a value equal to the average crop residue fuel produced by different crops in the US (Lal 2004). Then a historical average burned area per hotspot was calculated by the method described in chapter 4.1. No allowance for small fires was included in these fixed *GFL* calculations. The sensitivity test with fixed fuel loads is used to demonstrate the relative impact of small fires as well as the details of the agricultural fires parameterization on model results.

Figures B.4 and B.5 present the daily values of global emissions following the FAO approach versus the fixed *GFL* shown as a time series and as a scatter plot. These figures show a close agreement between the two predictions with an $r^2$ of 0.996 and a slope of 0.991. This indicates that for 2019, and likely other years, agricultural burning had an insignificant impact on global emissions beyond being modelled as a grass fuel and that small fires were inconsequential on a global scale.

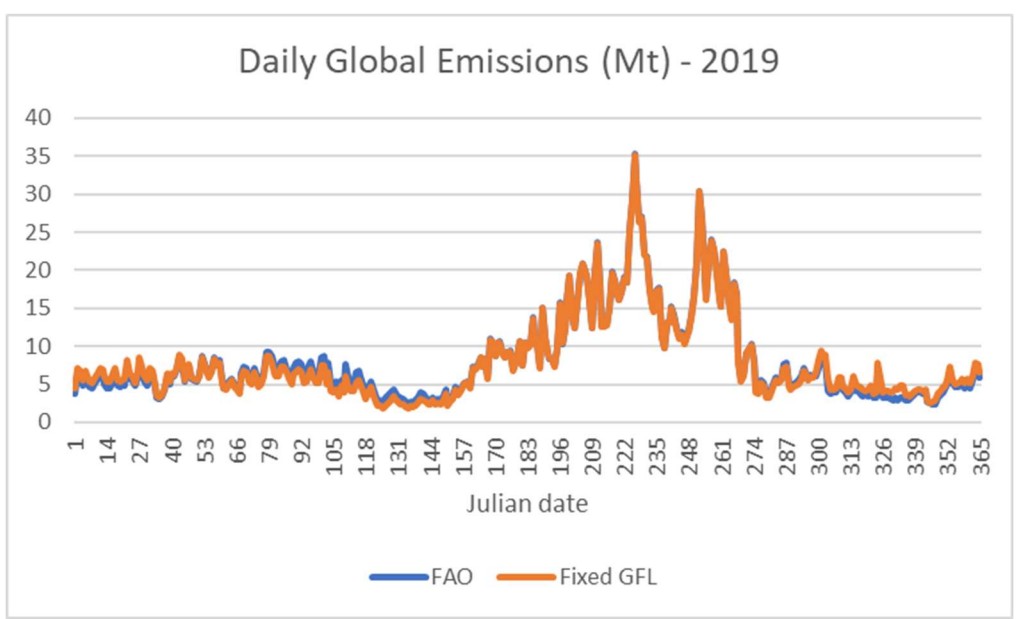

**Figure B.4. Time series of daily global emissions for 2019 using the FAO statistical approach versus a fixed grass fuel load (*GFL*) of 0.60 kg m$^{-2}$ for agriculture.**

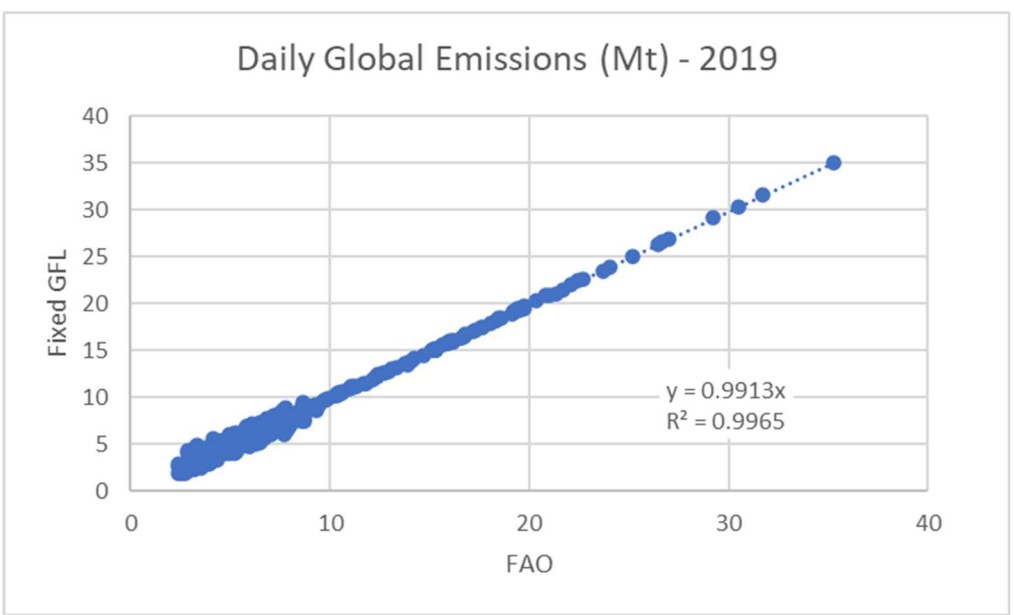

**Figure B.5. Scatter plot of daily global emissions for 2019 using the FAO statistical approach versus a fixed grass fuel load (*GFL*) of 0.60 kg m$^{-2}$ for agriculture.**

Locally and regionally, however, the agricultural burning methodology has a larger impact. Figure B.5 shows most variation between the methods occurs near the origin, and closer examination reveals this variation occurring primarily in the agricultural regions. Examining the regional differences within agricultural areas we find that in Europe, which has a large fraction of agricultural land though a small contribution to total emissions, the FAO approach used by GFFEPS produced 4.7 times the emissions produced using the average fuel load (Fig. B.6). Similarly, the FAO approach relative to the fixed values generates in TENA 2.9, in CEAS 2.3 and in MIDE 2.1 times the emissions. These are similar to recently published results by Hall et al. (2024), who reported a 2.7-fold increase in annual average cropland burned area (2003–2020) in cropland regions using the new global cropland area burned dataset (GloCAB) over the MCD64A1 product.

While the use of a single, fixed fuel load may be simplistic, this variation shown cannot simply be attributed to denser crop fuel loads. Wooded areas embedded in agricultural fields could contribute to larger fuel loads but the likely explanation is that these larger values are a result of smaller, undetected fires. This indicates the importance of properly modelling small fires in agricultural regions, and this would have an impact on air quality forecasting in these regions.

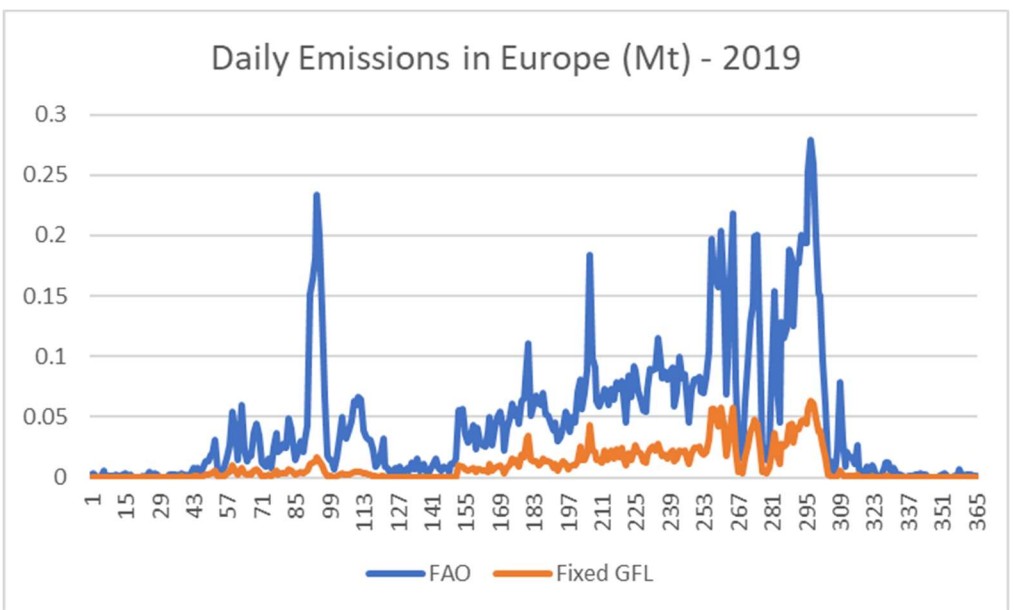

**Figure B.6. Daily emissions in Europe (Mt) for 2019 using the FAO statistical approach versus a fixed grass fuel load (*GFL*) of 0.60 kg m$^{-2}$ for agriculture.**

**B.3 Daily Weather**

The use of daily weather to predict fire behaviour and emissions is central to the GFFEPS model, due to its intended use in real time air-quality forecasting. Along with daily observed hotspots to determine burned area, the weather and fuel

type drives fuel consumption as predicted by the FBP system, the Growing Season Index (GSI) restricts fuel consumption in deciduous and grass fuels and the Foliar Moisture Content (FMC) affects the crown fuel consumption.

The relative sensitivity to daily weather variation was assessed by comparing the standard GFFEPS model predictions to those generated using a fixed consumption completeness, which when multiplied by the fuel load determines the amount of fuel consumption per area (similar to the FBP's total fuel consumption). This latter simulation thus eliminates the impact of meteorological variability. Consumption completeness values per GLC2000 land classification were not available so general values were assigned to forest (50%), grassland (75%) and peatland (25%) fuel types, based on average values for these categories from van Leeuwen et al. (2014).

Figures B.7 and B.8 present the daily values of global emissions using daily weather to drive FBP fuel consumption versus a constant combustion completeness, shown as a time series and as a scatter plot. These show close agreement between the two approaches with an $r^2$ of 0.979. The slope of 0.95 suggests that by using daily weather, the emissions drop by 5%, but this is an unreliable conclusion as the emissions are largely dependent on the general value used for combustion completeness.

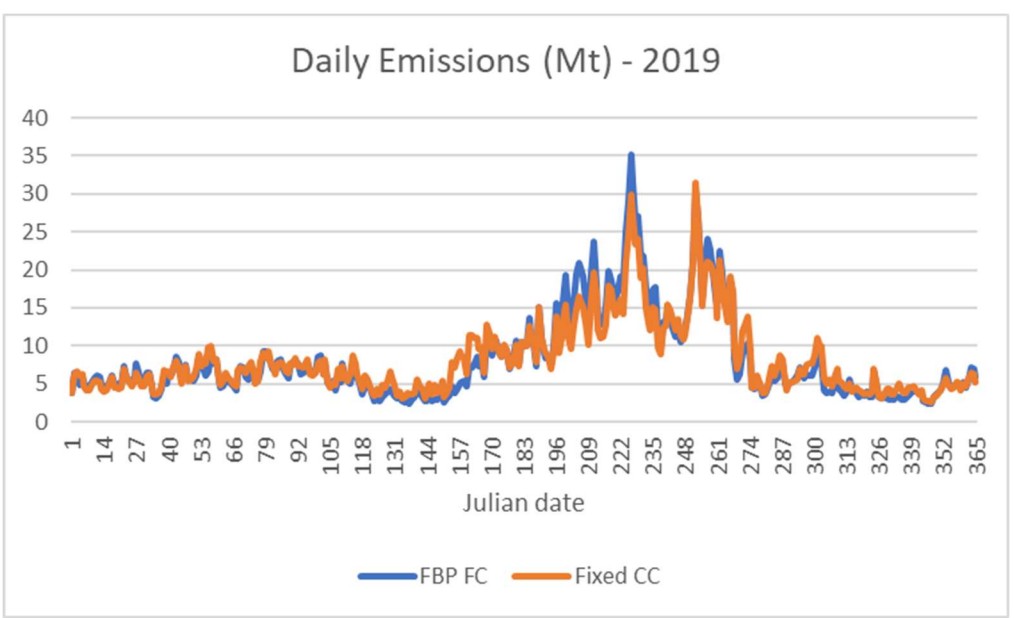

**Figure B.7. Time series of daily global emissions for 2019 using the daily weather to drive FBP fuel consumption versus a constant consumption completeness.**

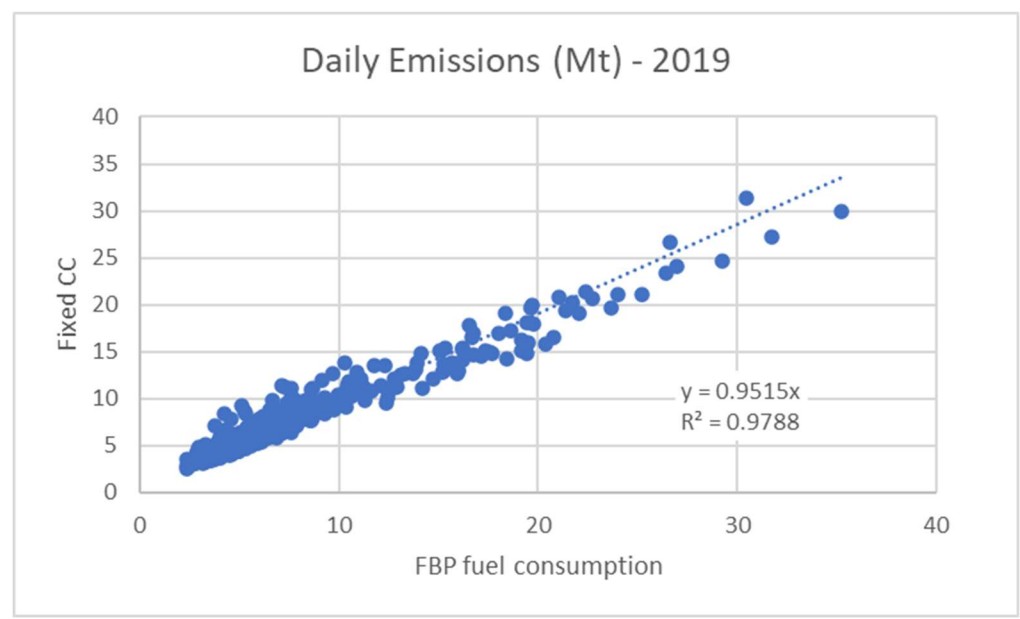

**Figure B.8. Scatter plot of daily global emissions for 2019 using the daily weather to drive FBP fuel consumption versus a constant consumption completeness.**

The variation around the emissions, especially at the lower end again suggests regional differences. In North America, emissions rates were lower when daily weather was employed: 71% in BONA, 75% in CEAM and 85% in TENA. In Boreal Asia (Fig. B.9), emissions were higher (298%) when daily weather was employed, due to the strong impact of weather on smoke estimates from burning peatlands, while in Australia emissions were 149% using the daily weather, reflecting the impact of El Niño. This indicates the impact of daily weather on air quality forecasting in these regions.

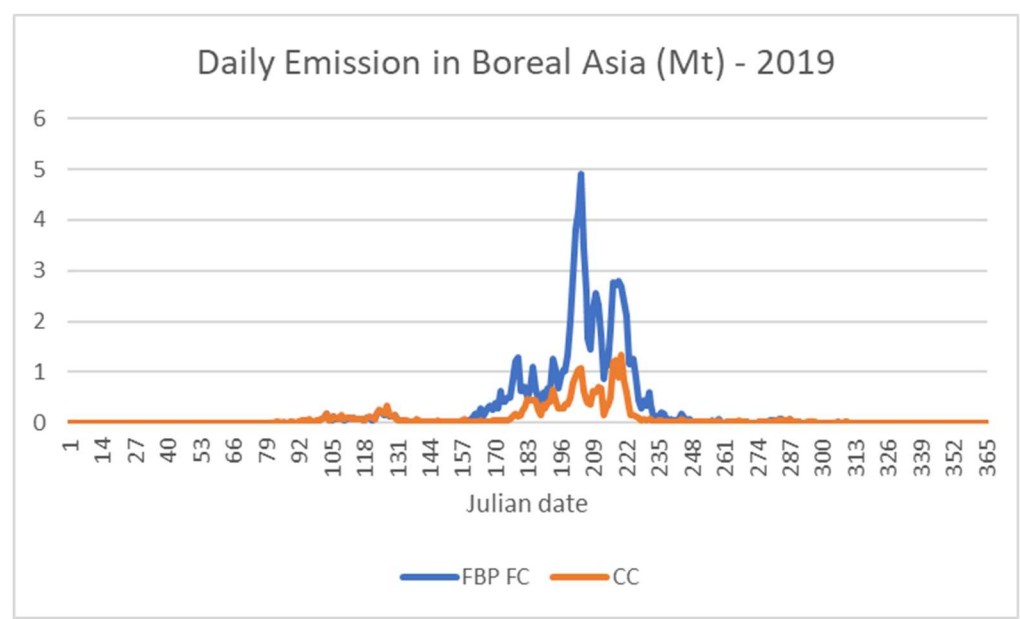

**Figure B.9. Daily emissions in Boreal Asia for 2019 using the daily weather to drive FBP fuel consumption versus a constant combustion completeness**.

*Code and data availability*. The data used in the analysis presented herein, and the GFFEPS code, are available online at https://zenodo.org/doi/10.5281/zenodo.10710452.

*Author contributions*. K.A.: research and development of GFFEPS, testing of code, manuscript preparation.  J.C.: preparation of meteorological data, compile and execute GFFEPS for the study period, manuscript review and editorial contributions.  P.E.: preparation of hotspot fields, development and documentation of hotspot methodology.  D.G.: manuscript review and editorial contributions.  P.A.M.: project management, manuscript review and editorial contributions. D.T.: review and assistance in development of GFFEPS equations and methodology, manuscript review and editorial contributions.

*Competing interests*. The authors declare that they have no conflict of interest.

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
