# Peer review of "The Global Forest Fire Emissions Prediction System version 1.0"

_Geoscientific Model Development, 2024_

## Author Response (AR1)

*Response to Reviewer 1*

*We thank the reviewer for their time and efforts reviewing this manuscript. Their comments were very helpful in improving the paper.*

*We regret that the reviewer found the paper confusing and difficult to read. The comments were very helpful in determining where the issues were and we have made substantive efforts to improve the clarity of the paper. Regarding the comment that the study requires a major overhaul, we contend that by clarifying certain points both in the manuscript and in this response, the reviewer's concerns have been addressed.*

*The reviewer's comments are provided in regular font – our responses are shown in italics. Revised sections of the manuscript are captured in quotes with starting line references in the revised manuscript included in square brackets.*

Anderson et al. describe the development of the existing Canadian Forest Fire Emissions Prediction System (CFFEPS) into the Global Forest Fire Emissions Prediction System (GFFEPS). The goal is to provide new input data for air quality modelling, taking advantage of the sophisticated way meteorological conditions are taken into account in CFFEPS. I have a number of minor comments (see below) and a few major comments. Addressing the major comments requires a major overhaul of the study and the way it is described.

The first major comment is that in its current state the paper describes the model and output, including a comparison against other inventories. While most literature (Ramo et al., 2021 [https://www.pnas.org/doi/full/10.1073/pnas.2011160118]; Chen et al., 2024 [https://essd.copernicus.org/articles/15/5227/2023/essd-15-5227-2023.html]) points towards more burned area and higher emissions than what has been estimated in the past decade, this paper finds lower numbers. The authors mention that larger validation efforts will follow but I feel they should be part of this paper. I don't feel it is justified to publish a methodological paper that needs to be revised shortly after, it would be better to include the validation in this paper and modify the approach if needed. Related to that is that the authors need to show that their approach has advantages over existing datasets. For example, does the more dynamic way of modelling fire characteristics lead to better results than FINN (or any other dataset) when comparing to field data (fuels, burned area) or to large-scale top-down constraints (for example Zheng et al., 2023 [https://www.science.org/doi/10.1126/science.ade0805])?

> *We feel GFFEPS has reached a benchmark with version 1.0 and that it is worthy of publication. At the same time, we expect GFFEPS project will continue to evolve over the years to come. In that sense, we expect that further validation will follow as new components are introduced. The reviewer perhaps misunderstands this when questioning that "*larger validation will follow*" but this would only be natural when the GFFEPS model develops into a version 2.0 (and beyond). With that in mind, we have removed the*

*phrase* "as a first step towards larger validation efforts" *(line 109 in the originally submitted manuscript).*

*Regarding validation and having* "advantages over existing datasets", *this was presented in the original manuscript's results chapter, where we showed this impact of daily weather and satellite observations on fire emission. Additional work is now presented in Appendix B.3, showing the impact of choice of land use datasets, agricultural emissions, and daily weather on smoke emissions both globally and within certain regions.*

*Regarding* "comparing to field data", *this was captured in Appendix A of the submitted manuscript, where GFFEPS methodology of calculating fuel consumption was compared to published field work in Canada, Siberia, Indonesia, African and Brazilian savannah, and Australia eucalypt, and then compared to values predicted by GFED. These comparisons were quite favourable, with GFFEPS predicting the range of values present in many of these studies, while those of GFED provided a single invariant value across conditions. To emphasize the importance of these Appendices, a reference was added to the end of the introduction:*

> "Two appendices are included. Comparisons of GFFEPS to field data is presented in Appendix A, where the GFFEPS methodology of calculating fuel consumption is compared to published field work in Canada, Siberia, Indonesia, African and Brazilian savannah, and Australian eucalypt, and compared to values predicted by GFED. Appendix B provides a sensitivity analysis, examining the impact of landcover data sets, of agricultural burning and small fires and of daily weather." [123]

*We note that at different points in the submitted manuscript that "Supplemental Section" was used interchangeably with "Appendix" in the text. "Appendix A." is now used throughout the revised manuscript to reference the supplemental fuels discussion.*

*Regarding the recent literature pointing towards more area burned, we expect that this will be an evolving process. The GFFEPS results are based on MCD64A1 data as the underlying burned area as a benchmark and proof-of-concept. The MC64A1 burned area product has been assessed/validated in Giglio et al. (2018), which, in turn, was used by GFED4.1s and used to calibrate GFAS1.2; validation and comparative studies have been conducted based on this burned area product (Van der Werf et al., 2017; Pan et al. 2020). We note that the use of other databases is certainly feasible, and data available at the current time may further change in the future. Further, we have evaluated the relative impact of MODIS versus GCL2000 land use data on the model performance for the year 2019, in new Appendix B.*

Second, I find the format confusing and the paper is difficult to read. Chapter 4 describes the methodology, but the previous sections also detail about different methods. For a reader it is difficult to keep track of whether CFFEPS or GFFEPS is being discussed. Also, it takes a long time before the Seiler and Crutzen equation is mentioned, that could be done early on. How about

1) Introduction, including the background of CFFEPS and other inventories

2) Methods and datasets with subsections of burned area, fuel, and other parameters. The part based on CFFEPS can be described briefly with reference to CFFEPS papers

3) Results

4) Discussion

*We have clarified the reasoning behind our construction of the paper and its headings - at the end of the Introduction, we have added the following:*

"Chapter 1 provides an introduction with historical content and need for the work. Chapter 2 provides the underlying theory of the model and foundational work. Chapter 3 outlines the external data required to drive the model while Chapter 4 describes the internal calculations and methodology. Results are presented in Chapter 5, discussion in Chapter 6 and conclusion in Chapter 7.

"Two appendices are included. Comparisons of GFFEPS to field data is presented in Appendix A, where the GFFEPS methodology of calculating fuel consumption is compared to published field work in Canada, Siberia, Indonesia, African and Brazilian savannah, and Australian eucalypt, and compared to values predicted by GFED. Appendix B provides a sensitivity analysis, examining the impact of landcover data sets, of agricultural burning and small fires and of daily weather." [119]

*Regarding the reviewer's comment regarding difficulty of keeping track of "whether CFFEPS or GFFEPS is being discussed", we have removed all but one reference [451] to CFFEPS following the Theory (2) until it is reintroduced in the Discussion (6).*

*Regarding the placement of the Seiler and Crutzen equation, we feel it is best being introduced in the subchapter on Global Models (2.3), as it does not play a role in the previous subchapters on the CFFDRS and CFFEPS.*

The third one is the land cover data. GLC is used, and developed for the year 2000. In almost 25 years many regions in the world have undergone changes in land cover, these should be incorporated to not misclassify fuels and other fire characteristics

*The reviewer is correct in stating there are other, more recent land use classifications than GLC2000. While we do not rule out adding other land use classifications in future, the decision to use GLC2000 were based on the following:*

*(1) We needed a global land use of sufficient resolution that was easy to employ (for proof of concept) and GLC2000 was well suited, given that it provides a single map global coverage at a 1-km resolution. We note that MODIS could also be used, but would require a trade-off between using their lower resolution 5-km*

*resolution map (MCD12C1) or stitching together many maps at 500-m resolution (MCD12Q1).*

(2) *A factor in favour of using the GLC2000 is the national expertise and ground truthing involved in the construction of that dataset. While MODIS has been validated, we felt there was additional value added in the multinational approach used in creating the GCL2000.*

(3) *Regarding the changes that are occurring in the last 25 years, we agree that the GLC2000 dataset is "getting old". We note, however, that this is less critical than might be expected as vegetation classes rarely change (e.g. deciduous forests rarely change into coniferous) and most land use changes, whether they were a result of disturbance (fires, deforestation) or urbanization, would result in landscapes less fire prone and this in turn would be reflected by a reduced number of hotspots. That is, the changes that might be expected would reduce the likelihood of fires occurring in the regions undergoing a change. For example, there should be fewer hotspots (if any) appearing in a burn scar, or in a formerly forested area that has been cleared and converted to urban landscapes.*

*The specific question of the value added using recent land classification data was addressed in Appendix B.1, where GFFEPS model results were compared using MODIS MCD12C1 data over GLC2000 for the year 2019. The finding was that global impact was negligible, while in certain regions, specifically Africa, differences were noted but that these may be a result of poor classification of woody savannas.*

*Our intention in the future is to move towards additional land use classifications, but for our immediate purposes, GLC2000 was deemed adequate for the reasons stated above. This is captured in 3.1, where now we state:*

> "While other land use databases are available (e.g., MODIS, etc.), GLC2000 was selected for its global spatial resolution with 1-km at the equator, for its level of detail in the number of land use types, for the national-level ground-truthing data used in its construction, for ease of data usage, accessibility and for consistency throughout our analysis. While acknowledging the 25-year age of the GLC2000 dataset, we note that land use changes occurring subsequent to the year 2000 are unlikely to result in a significant change in biomass burning emissions in an on-line model such as GFFEPS. For example, vegetation classes rarely change (e.g. deciduous forests rarely change into coniferous) and most land use changes, whether they were a result of disturbance (fires, deforestation) or urbanization, would result in landscapes less fire prone and this in turn would be reflected by a reduced number of hotspots. In turn, reduced hotspot detection would result in less smoke emissions, capturing the impact of the land use change. However, we note that the same methodology developed here using GLC2000 may be used with other land-use databases, including time-varying databases such as those provided by satellite retrievals (e.g. MODIS). We present comparisons between GFFEPS configured for MODIS land use data versus GLC2000 in Appendix B.1" [241]

The fourth is somewhat related. If I understand correctly, you use lookup tables to assign fuels to each GLC class. In other words, fuels are not variable beyond their GLC class. I feel the science has progressed way beyond this and it would be good to have a spatial and temporal component for the fuels; they can vary a lot over space and time

*We have clarified this in the revised manuscript. The GLC2000 land use categories are certainly one determining factor in the choice of fuels used in GFFEPS, but not the only one. Another determining factor, and included in GFFEPS, is the region: peatlands in the tropics differ from those in northern latitudes (4.3.1, and Appendix A.3); coniferous forests differ between North America and Eurasia (Appendices A.1, A.3), while in Australia they were assigned to eucalypts with its own specialized fuel consumption (4.3.2, and Appendix A.5). We have modified the text in section 3.1 to make this clearer:*

"We note that both the land use classification (GLC2000) and the region classification are used in determining the fuel assignment. For example, peatlands in the tropics differ from those in northern latitudes (see 4.3.1 Peat Fires and Appendix section A.3); coniferous forests differ between North America, Eurasia and Australia (see Appendices A.1, A.3, section 4.3.2 and Appendix A.5)." [269]

*It is also worth noting that beyond GCL class and region, fuels vary by phenology. This is now emphasized as a separate subchapter (3.4).*

Minor

L25-30: You compare a low fire year (2021) with regrowth estimates from a longer time period. In general "net" fire carbon emissions are higher than the 0.1 Pg C one would refer from the difference between the quoted 1.84 and 1.75 Pg C

*At the time of writing, the AMS State of the Climate in 2021 was the most recent. The 2022 report is now available. The 2003–20 mean value (min−max) equals 2.062 (1.781−2.421) Pg C $yr^{-1}$, which we have now used instead. Adjusted values now suggest 0.3 Pg C $yr^-1$ net fire carbon emissions (2.062-1.75 = 0.312).*

L32: "Estimates show that between 2003 to 2017, biomass burning accounted for 1.68 to 2.27 Pg C yr-1 ". Well, that is one dataset and if the emissions were so well known then this work would not be necessary. Please include the range of estimates

*The range as reported in the AMS State of the Climate in 2021 (1.68 to 2.27 Pg C $yr^{-1}$) was provided in the original text. The updated range for 1.781−2.42 between 2003 and 2020 based on the most recent (2022) State of the Climate report is now included. If the reviewer is asking for a range of different global model predictions, they are presented in the Results chapter.*

*We note that our intent here is to provide a biomass burning emissions processing system for use in air quality forecasting; we disagree with the reviewers' contention that the*

*existence of a range of annual emissions estimates (from the given reference or others) in some way making* "this work would not be necessary."

L34: please change the bullets to a narrative

*This is a difference in writing styles. We feel that bullets are a clear and descriptive way of presenting the information. Also, the final bullet referring to the 2023 fires season in Canada has been updated by statistics from a recent publication in Nature. It now reads as follows:*

"Canada's burned area reaching a record 15.0 Mha for 2023, exceeding the previous record of 6.7 Mha set in 1989 (Kolden et al. 2024)." [43]

L64: FRE is the time integral of FRP, not the other way around

*Yes, thank you for spotting the error. The sentence now reads:*

"… fire radiative energy (FRE, the time integral of fire radiative power (FRP); Mota and Wooster, 2018), and biome-specific conversion factors …" [65]

L66: FINN does not use MCD64A1

*The reviewer is correct regarding FINN. Our claim was intended as a general statement. Reference specifically to FINN has been removed. Instead, a description of FINN's calculations is now included in the 5.2 Results section:*

"FINN 1.5 calculates burned area based on active fire pixels detected by the MODIS Aqua and Terra satellites at 1 km2 (0.75 km2 in grasslands/savannas) per detection, which is then adjusted by percent tree, non-tree vegetation, and bare cover at 500m as provided by MODIS Vegetation Continuous Fields (VCF). FINN 2.5 (Wiedinmyer et al., 2023) uses a more sophisticated approach, aggregating VIIRS hotspots to create burned area polygons.." [659]

L71: Burned area approaches are not fundamentally restricted by satellite-overpass times as what you label 'top-down' studies (in the literature top-down studies often refer to those that use atmospheric observations of for example CO to constrain emissions

*The sentence reads "The satellite-based fire detection used by both top-down and bottom-up methodologies are restricted by satellite-overpass times, …", which is correct. The sentence does not indicate that satellite-based detection is used exclusively by either methodology. Top-down approaches could use aircraft or geosynchronous satellites (with no overpass times). To reflect this, the sentence has been modified to "generally restricted". [73]*

*We also note that the reviewer's example of CO measurements is not used in real-time fire detection so much as evidence of recent fire activity (Zheng et al., 2023).*

L79: GFED also has spatially and temporally variable fuel loads and combustion completeness (also relevant for next paragraph)

> *There is a recognized variability of fuel load and combustion between GFED land classifications (van Leeuwen et al., 2014), while Van Wees et al. (2022) introduced monthly variability as temperature and water stress scalers in a simplified version. Our point is that these values don't change on a temporal or spatial scale as presented in the GFFEPS model (namely, in GFED, the daily weather variations between surface locations does not impact the fire emissions, while this variation is captured in GFFEPS).*
>
> *The sentence has been rewritten as follows:*
>
> > "The bottom-up approach used by GFED is based on observed burned area (MODIS MCD64A1 mapping algorithm), landscape maps for fuels (MODIS MCD12Q1 land cover type), along with fuel loads, combustions completeness and emission factors per biome typically collected from the literature (van Leeuwen et al., 2014)." [67]

L115: multiply -> multiplied

> *Corrected.*

L117: "Fuel consumed is used to calculate total heat flux from the combustion process and then used to calculate plume injection height". One would expect that a rate would be more useful here

> *The sentence has been rewritten as follows:*
>
> > "Fuel consumed per time step is used to calculate heat flux from the combustion process and then used to calculate plume injection height, …" [134]

L177: One would wonder whether using FRP would lead to better results than just hotspots

> *Line 177 states* "The application of CFFEPS calculations is conducted on each satellite-detected hotspot." *The purpose of our manuscript is to present the methodology used in GFFEPS and compare its emissions predictions to other global models. The fundamental, bottom-up approach used by CFFEPS (and carried into GFFEPS) is summarized in this paragraph.*
>
> *Other top-down models such as GFAS use FRP to calculate fire emissions and plume rise. Emissions from GFFEPS and GFAS are compared in the results chapter. Further comparison of CFFEPS plume rise predictions to other models was conducted in Griffin et al. (2020) and deemed unnecessary to add to this GFFEPS manuscript.*

L193: Please consider introducing this equation earlier as it is also used in GFFEPS

*As discussed earlier, we feel that presenting the Seiler and Crutzen equation in the subchapter on Global Models (2.3) was the appropriate placement.*

L210: Think this Table can go by referring to Giglio et al

*True. Table 2 was placed in the manuscript before Figure 2 was added. The two together are redundant so Table 2 has been removed.*

L230: In my opinion (see feel free to ignore) Figure 3 and Table 3 do not add much as it is basically unmodified input data. Figure 4 is useful

*Given the GLC2000 is the starting point for fuels classification, it is beneficial to present it, and useful for those unfamiliar with this dataset. As the text is too long for the figure legend, Table 3 was included.*

L269: That is a firm statement, can you back this up with references?

*The sentence states "In spite of these limitations, hotspot data from VIIRS provides a picture of global fire activity that is consistent, continuous, and sufficiently complete."*

*This was intended as a generalized, self-evident statement. We have rewritten it as follows:*

"In spite of these limitations, we selected VIIRS data because it is sub-daily, global, readily available, higher resolution than alternative sensors, available in near-real time, and expected to continue well into the 2030s." [299]

L290: Not clear what the map contributes, can be written down in words

*The importance of the BUI and other factors is addressed in the description of the Canadian Forest Fire Danger Rating System (2.2). With that said, the importance of this was not captured later in the text, as the reviewer is indicating. The following text has been added:*

"Daily FWI is central to the GFFEPS system: noon values are used to calculate the FWI values, which are then used to predict fire behaviour and resulting smoke emissions. For example, the Buildup Index (BUI) is one of the FWI indices and a principal driver in calculating fuel consumption in the FBP system." [317]

Section 3.3: Maybe some of these parameters are useful to develop variability in fuels, as long as they can be constrained. Equations 2 and 3 seem very ad hoc without references"

*As discussed in the previous comment, the greenness of the landscape (captured by the GSI) and the foliar moisture content (FMC) are significant factors in fuel consumption. These are repeated in 4.3 Combustion Completeness.*

*To reinforce the importance of phenological effects, the sections on Growing Season Index (3.3.1) and on Foliar Moisture Content (3.3.2) have been separated from Global Weather (3.3) and a new subchapter entitled 3.4 Plant Phenology has been added. The introduction reads as follows:*

> "Seasonal cycles in plant characteristics, known as phenologies, significantly influence the timing and quantity of live vegetative growth. These phenological changes influence overall fuel moisture levels (considering both live and dead fuels) and consequently impact fire behaviour. In temperate and boreal ecosystems during spring, deciduous trees in the temperate zones emerge from winter dormancy, leafing out through the growing season before shedding leaves as they return to dormancy in autumn (Alexander, 2010a; Quintillio et al., 1991). Similarly, grasses green-up in the spring and reach maturity, then desiccate in the summer heat, either dying off or re-entering dormancy in warm temperate, Mediterranean, and tropical climates with a strong wet-dry seasonality such as Australia (Cheney and Sullivan, 2008). Grasses as well as trees in cool temperate and boreal regions are controlled by a combination of photoperiod and freezing temperatures initiating grass curing (Jolly et al., 2005). Coniferous crowns undergo an important seasonal dip in foliar moisture during the spring as needles transpire while the roots are still frozen that has impacts on the initiation of crown fire (Alexander, 2010a). These effects have an important impact on smoke emissions and hence have been addressed within GFFEPS." [329]

*Equations 2 and 3, used to calculate the date of minimum FMC in Eurasia, were derived for this manuscript and hence have no reference. They are a direct extension of the derivation used in North America, which is referenced in the preceding paragraph (Forestry Canada Fire Danger Group 1992).*

Section 3.4: There is a new burned area dataset that may be useful here to validate the Tier 1 approach (https://essd.copernicus.org/articles/16/867/2024/essd-16-867-2024.html)

*Thank you for the recent publication on global cropland burning.*

*We have now added a sensitivity analysis to the manuscript as Appendix B.2. In this analysis, we tested the impact of specific input parameters on the GFFEPS results for 2019. In B.2, we replaced our FAO statistical approach (as presented in 3.4), which accounts for small agricultural fires, with a default fuel load of 0.6 kg m$^{-2}$ and a burned area per hotpot (as described in 4.1) with no accounting for small fires. The results are strikingly similar to those presented in Hall et al., (2024), where we say*

> "Examining the regional differences within agricultural areas we find that in Europe, which has a large fraction of agricultural land though a small contribution to total emissions, the FAO approach used by GFFEPS produced 4.7 times the emissions produced using the average fuel load. Similarly, the FAO approach relative to the fixed values generates in TENA 2.9, in CEAS 2.3 and in MIDE 2.1 times the emissions. These are similar to recently published results by Hall et al.

(2024), who reported a 2.7-fold increase in annual average cropland burned area (2003–2020) in cropland regions using the new global cropland area burned dataset (GloCAB) over the MCD64A1 product." [1198]

L 400: Giglio et al. 2016 is about active fires, their 2018 paper is more relevant here

*Thank you for the correction. Updated in the revised manuscript.*

L 427: The VIIRS fire community must have a database with stationary sources to help cleaning up the dataset so you only capture landscape fires

*Perhaps but to our knowledge, this has not been done.*

Section 4.3. This section is titled combustion completeness but much of the text is about fuel consumption. It becomes increasingly difficult as a reader to follow your workflow, please consider a major overhaul of the structure of the paper.

*As mentioned in the opening paragraph of 4. Methodology, "the sections described here under the methodology follow each of the equation variables". To make this clearer, the following has been added to the introduction to chapter 4.*

"GFFEPS follows the same methodology as CFFEPS (Chen et al. 2019) but uses additional data sets and alterations described in this section. Likewise, GFFEPS follows Eq. (1) and the section titles described under the methodology follow each of the Seiler and Crutzen equation variables: burned area (BA), fuel load (FL), combustion completeness (CC) and emission factors (EF)." [451]

*Regarding combustion completeness (CC), GFFEPS does not calculate CC as a percent of the fuel load (4.2) consumed. Instead, the FBP fuel consumed is equal to the product of Seiler and Crutzen's fuel load and combustion completeness (FL × CC). The opening line of this section (4.3) now reads:*

"The combustion completeness (CC) used in GFFEPS is captured by the total fuel consumption (TFC) as calculated by FBP, which is equal to the product of Seiler and Crutzen's fuel load and combustion completeness (FL × CC)." [502]

Section 4.4: this is a novel aspect of this work, would be nice to see a comparison between your approach and the existing datasets (Akagi, Andreae)

*The calculation of emissions factors (4.4) is a sophisticated process involving a convolution function of hourly emission and area burned. The three stages (flaming, smoldering and residual) and the duff layer characteristics also play a significant role in determining hourly emissions. These are described in detail in Chen et al. (2019).*

*Chen et al (2019) further describes the emissions per species as based on Urbanski (2014), which in turn uses work by Akagi et al. (2013). A reference to Urbanski (2014) has been added to the current manuscript.*

*Akagi, S. K., Yokelson, R. J., Wiedinmyer, C., Alvarado, M. J., Reid, J. S., Karl, T., Crounse, J. D., and Wennberg, P. O.: Emission factors for open and domestic biomass burning for use in atmospheric models, Atmos. Chem. Phys., 11, 4039–4072, https://doi.org/10.5194/acp-11-4039-2011, 2011.*

*Urbanski, S.: Wildland fire emissions, carbon, and climate: Emission factors, Forest Ecol. Manag., 317, 51–60, https://doi.org/10.1016/J.FORECO.2013.05.045, 2014.*

L529: How nice would it be to see comparisons beyond Canada and the US

*We agree with the reviewer, yet the issue is one of insufficient data appearing in the literature. We searched through various publications and news reports but could not find reliable burned area data beyond Canada and the USA. Most information was partial (e.g., statistics for specific European countries or for individual Australian states) or not compatible (e.g., deforestation in the Amazon Basin but not area burned specifically nor presented as the entire southern hemisphere of South America). If the reviewer has additional references, we can include them.*

L564: To some degree it is expected that your values match GFAS and GFED as you use more or less the same burned area and fuel consumption numbers. Would be nice to show where you make a difference

*The reviewer states that we "use more or less the same burned area and fuel consumption numbers." "More or less" is a subjective term that discredits how we calculate burned area per hotspot in near-real time based on historical data, while GFED uses burned area data accumulated over the course of a month. While it can be argued we leaned on van Leeuwen's fuel load data (for recalculation), GFFEPS fuel consumption is based on the FBP and FWI system calculations that uses daily observed weather, which neither GFED nor GFAS use. Appendix A. further shows differences between the fuel consumption calculation between GFFEPS and GFED. Appendix B shows the relative impact of daily weather versus constant fuel consumption on GFFEPS' predictions.*

*The impact of these differences is presented in 5.2. Figure 11 shows that GFFEPS average global annual emissions are lower than GFAS/GFED (80%/74%), similar to FINN1.5 (97%) and much lower than FINN2.5 (38%). Similarly, Figure 12 shows the regional differences between GFED4.1s and GFFEPS; Figure 13, the annual area burned between MCD64A1, GFFEPS and FINN1.5/2/5; and Figure 14 shows daily difference in area burned between GFFEPS and FINN1.5/2.5. Along with each figure is a discussion of the differences in the accompanying text.*

*Therefore, we contend that we do not "use more or less the same burned area and fuel consumption numbers". We feel that we have done a sufficient job of presenting the*

*differences, and the model's sensitivity to input parameters, in the Appendices in the revised manuscript.*

L615: Looks we agree on the above

*The line indicates that "the GFFEPS model is in general agreement with well-established models." This is a general statement (i.e., our calculations do not differ by a magnitude or more), while the differences between the models is discussed in 5.2. We then concede that "much of the GFFEPS methodology and input data is similar to that used in GFED4.1s" but then go into describing the differences saying*

> "Nonetheless, the key essential differences between the two models are that GFED4.1s uses static fuel loads and consumption completeness per biome, while GFFEPS models these dynamically, both spatially and temporally, achieved by using the well-established CFFDRS with FBP fuel consumption driven by FWI fire weather; that GFFEPS considers plant phenology not explicitly recognized in GFED; and that GFFEPS calculates real-time burned area based current hotspots and historical statistics, while GFED uses burned area data accumulated over the course of a month from remotely sensed data.." [699]

*Later, Figures 14 present differences in daily burned area between GFFEPS and FINN1.5/2.5. We now analyze the impact of three additional factors in Appendix B of the revised manuscript.*

L616: GFED also has variable fuels. It makes me again a bit wondering about the added value of this work but that could be partly because it is often difficult to grasp the methodology. With a better structure, validation, and clearer description of the unique aspects of this work this could be mitigated

*As commented on earlier, we recognize that GFED has fuel loads and combustions completeness values that vary between biomes/land classification (and modified the text to reflect this). GFFEPS includes these as well, but goes beyond this by including daily and spatial variability of weather and fire behaviour, the impacts of plant phenology and daily predicted area burned based on satellite detections and historical burned area statistics, all of which are not considered within the GFED calculations. This is the heart and soul of its added value, what we are describing as "spatial and temporal variability."*

*We have strengthened these claims throughout many points in the revised manuscript (as described in our response) and hope that this additional information allows the reviewer to appreciate the differences between the two approaches and recognize the benefits of including daily weather and fire behaviour in real-time smoke forecasting.*

*Response to Reviewer 2*

*We thank the reviewer for their time and efforts reviewing this manuscript. Their comments were very helpful in improving the paper. The comments were very helpful in determining where the issues were and we have made substantive efforts to improve the the paper.*

*We disagree with the reviewer's thesis that the paper is premature. This addressed in a summary at the end.*

*The reviewer's comments are provided in regular font – our responses are shown in italics. Revised sections of the manuscript are captured in quotes with starting line references in the revised manuscript included in square brackets.*

Anderson et al. set out to create a global version of the Canadian Forest Fire Danger Rating System. Overall, this is an interesting proof of concept; however, I believe the publication of this paper is premature. Please see specific comments below.

**Major Comments**

Section 3.2: Do the authors have to worry about double counting fires when all the VIIRS sensors are used? Was any preprocessing done on the fire data? Why did the authors not filter out the presumed non-vegetation fire using the type flag? The science quality dataset includes this information.

> *We only used one VIIRS sensor (S-NPP). We used hotspots from 2012-2019, during most of which there was only one VIIRS in orbit. In future, with multiple satellites, double- and multiple-counting will be handled by the "times burned" metric and methodology.*
>
> *Upon investigation, we have found that the type flag mentioned by the reviewer unfortunately is not very reliable. We cross-compared our own industrial sites mask for Canada, to the categories present in the type flag: that is, we have ground truthed data upon which to evaluate the type flag. We found that the hotspots associated with these known industrial sites were classified in the VIIRS type flag as a mixture of type 0 (vegetation fires), type 2 (static), and type 3 (offshore). Most of the hotspots from smaller industrial sites were categorized by the VIIRS type flag as vegetation fires. Meanwhile an oddly large number of hotspots from wildfires are classified as offshore, though they are certainly on land, from their latitude and longitude coordinates. However, it is true that almost all the type 2 hotspots are from industrial sources, and these could have been removed from the analysis.*

Lines 310 – 315: It seems the author's primary justification for using GSI instead of NDVI is the ease of use because remote sensing data requires an extra step to mosaic the data together. How different would the results be using NDVI? Is there a more substantial scientific justification for using GSI instead of NDVI that can be added here?

*One of the goals of GFFEPS is running in real time or forecasting, for which NDVI would not be available. NDVI would probably be better than GSI for deciduous leaf phenology [and grass curing]. However, obtaining and processing NDVI data is significantly more difficult. To [re-]run GFFEPS for the 2012-2019 testing period using the VIIRS/NPP and JPSS1 Vegetation Indices 16-day global 1km product would involve downloading and processing some 80,000 files totalling 2TB – not impossible, but a significant undertaking.*

*Note that the referenced GSI paper Jolley . (2005) has an extensive comparison of GSI against NDVI. The GSI is currently being used by the US Forest Service as part of the National Fire Danger Rating System ([https://www.firelab.org/project/national-fire-danger-rating-system](https://www.firelab.org/project/national-fire-danger-rating-system)). We hope this addresses the reviewer's comment.*

Section 3.4: The authors may want to add some additional caveats including the fact that the FAO stats may not be accurate in countries where agricultural burning is illegal but widespread. For example, in Ukraine and Russia. Please see the following paper for further information: https://iopscience.iop.org/article/10.1088/1748-9326/abfc04

*Thank you for providing the reference on illegal burning in the Ukraine (Hall et al., 2021). We acknowledged that the approach presented may have issues, especially in developing countries and, in the case of Ukraine, there will be additional caveats to our approach.*

*Hall et al. (2021) that indicates "that cropland BA [in Ukraine] was significantly underestimated (by 30%–63%) in the widely used Moderate Resolution Imaging Spectroradiometer-based MCD64A1 BA product", yet this would be a small slice of global emissions. Figure 7a shows that on average, Boreal Asia accounts for 28 MgC yr$^{-1}$, 2% of the 1479 MgC yr$^{-1}$ global average. Also, many of the fires in Boreal Asia are forest fires in Siberia.*

*Regardless, as the reviewer suggests, we have added the following*

"With that said, the Tier 1 methodology used by the FAO to determine this value may not be rigorous in developing countries (Tubiello et al., 2014) or where illegal agricultural burning is widespread (Hall et al., 2021); nevertheless, its application in GFFEPS seemed a direct and practical solution for real-time smoke forecasting while addressing the small fire issue specific to agriculture activities." [446]

Furthermore, the Global Cropland Burned Area dataset (Hall et al., 2024) was recently released. It represents the cropland burned area within GFED5 (https://essd.copernicus.org/articles/16/867/2024/). I suggest using either this product or another product specifically designed to map agricultural burned area and compare some of the burned area statistics. The above-mentioned paper focused on Ukraine uses VIIRS active fires, so that is more in line with the author's methodology.

*Thank you for the recent publication on global cropland burning.*

*We have now added a sensitivity analysis to the manuscript as Appendix B. In this analysis, we tested the impact of specific input parameters on the GFFEPS results for 2019. In B.2, we replaced our FAO statistical approach (as presented in 3.4), which accounts for small agricultural fires, with a default fuel load of 0.6 kg m$^{-2}$ and a burned area per hotpot (as described in 4.1) with no accounting for small fires. The results are strikingly similar to those presented in Hall et al (2024), where we say*

> "Examining the regional differences within agricultural areas we find that in Europe, which has a large fraction of agricultural land though a small contribution to total emissions, the FAO approach used by GFFEPS produced 4.7 times the emissions produced using the average fuel load. Similarly, the FAO approach relative to the fixed values generates in TENA 2.9, in CEAS 2.3 and in MIDE 2.1 times the emissions. These are similar to recently published results by Hall et al. (2024), who reported a 2.7-fold increase in annual average cropland burned area (2003–2020) in cropland regions using the new global cropland area burned dataset (GloCAB) over the MCD64A1 product." [1198]

Section 5.1: The GFFEPS model underestimates the burned area in BONA (two areas with large burned area scars) and TENA. The authors then go on to say that the $R^2$ shows that the BA methodology is appropriate. Surely, the authors require an appropriate accuracy assessment to make this claim.

*In the text we stated that the $r^2$ values "suggest[s] the methodology for estimating burned area is appropriate." This is not a strong claim as there is uncertainty in the physical mapping of the fires. The detail in the criticisms of the national statistics is indicative of the complexities involved in accepting reported number and why we can't provide an accurate assessment.*

*The point of this paragraph is to compare GFFEPS calculation to real-world data. Regretfully, the latter has its own issues.*

*The portion of the paragraph has been rewritten as*

> "This suggests the methodology for estimating burned area used by GFFEPS is appropriate, though with a bias. On the other hand, reported national statistics of burned area have their own sources of error. For example, the level of rigour in mapping varies between Canadian provincial and territorial agencies, where unburned areas within fire perimeters may be captured by some agencies and not by others. This variable quality is then passed onto the national statistics. Similar issues are likely occurring in US statistics. The issue of mapping irregularities was also recognized by Fraser et al. (2004), who indicated the coarse resolution burned-area (approx. 1-km) provided by SPOT VEGETATION and NOAA AVHRR imagery produced burned-area estimates 72 percent larger than the crown fire burned area mapped at 30 m using Landsat TM (11,039 versus 6,403 ha average area). This bias

was attributed to spatial aggregation effects.  In summary, it is difficult to make clear conclusions from national statistics but these indicate the GFFEPS methodology is producing realistic results." [604]

*Fraser RH, Hall RJ, Landry R, Lynham TJ, Lee BS, Li Z (2004) Validation and calibration of Canada-wide coarse-resolution satellite burned area maps. Photogrammetric Engineering and Remote Sensing 70, 451–460.*

Line 569 and 579: The authors should compare their BA against GFED5 BA before making this claim since GFFEPS does not account for smaller fires and uses FAO stats for agricultural burning. Also, Africa is dominated by small fires in general, not just agricultural fires. Small fires include the smaller burned patches around larger burn scars, not just an actual small fire (which seems to be how the author interprets them based on lines 665 onwards).

*Line 569 states* "The lower values are largely attributed to the inclusion of daily fire behaviour in the combustion completeness calculations, not accounted for in the other models" *while 579 states* "These are areas dominated by agricultural burning, highlighting the impact of using FAO's crop-burning statistics."

*The reviewer states that* "GFFEPS does not account for smaller fires". *This is not correct: Agricultural Burning (3.4) of the submitted manuscript specifically describes how GFFEPS accounts for small fires.  The reviewer correctly states that GFFEPS* "uses FAO stats for agricultural burning", *without describing why this is a concern.  The FAO stats are a direct reporting of all agricultural burning, and hence is directly relevant for accounting for emissions from small, undetected fires.*

*With that said, the reviewer references African fires, where FAO statistics may not be handled as well as those in developed countries, and where small fires may be occurring in non-agricultural regions.  While this may account for the lower values predicted by GFFEPS in the region (Figure 12), it likely goes beyond agricultural fires.  This is discussed in the new Appendix B.1, where discrepancies were prevalent in Africa when substituting MCD12C1 for GLC2000 land classification*

"These differences are likely to poor matching of coniferous versus deciduous forests, a distinction not captured in MCD12C1 classifications Savannas and Woody savannas (as previously described).  The difference between coniferous and deciduous fuels is critical in the FBP fire behaviour calculations and any misclassification would have an impact on predictions.  Also, difficulties mapping fire emissions and land classifications in Africa have been discussed in various papers (Ramo et al., 2021; Nguyen and Wooster, 2020; Zhang et al. 2018), possibly accounting for the discrepancy shown in this comparison." [1163]

*As for comparing GFFEPS burned area against GFED5, reference was made to Chen et al 2023 in the original manuscript (line 76) as a method to account for the small fire boost.  This was published as our text was being prepared and would require a new analysis (see summary).*

*Finally, the methodology presented in the submitted manuscript, using FAO data, is the most recent (and we feel, the most accurate) of a series of approaches we examined to address the issue of small fires in GFFEPS. The first approach included the small fire boost documented in Randerson et al. (2012). The second approach assumed that fire size follows a power law (Cumming 2001; Hantson et al. 2016; Reed et al., 2002), using data from the Global Fire Atlas (Andela et al. 2019). Insufficient information was available to give us confidence in the first approach, and the second, while promising, resulted in extrapolations of small fires that often produced unacceptable results. With that said, the second approach pointed to a clear distinction of agricultural verses non-agricultural small fires, allowing us to confidently proceed with our assumption that small fires are inconsequential in non-agricultural regions. This was further supported in the Appendix B.2, where we now say*

> "Figure B.4 shows most variation between the methods occurs near the origin, and closer examination reveals this variation occurring primarily in the agricultural regions. Examining the regional differences within agricultural areas we find that in Europe, which has a large fraction of agricultural land though a small contribution to total emissions, the FAO approach used by GFFEPS produced 4.7 times the emissions produced using the average fuel load. Similarly, the FAO approach relative to the fixed values generates in TENA 2.9, in CEAS 2.3 and in MIDE 2.1 times the emissions. These are similar to recently published results by Hall et al. (2024), who reported a 2.7-fold increase in annual average cropland burned area (2003–2020) in cropland regions using the new global cropland area burned dataset (GloCAB) over the MCD64A1 product." [1196]

*Andela, N., Morton, D.C., Giglio, L., Paugam, R., Chen, Y., Hantson, S., Van Der Werf, G.R. and Randerson, J.T., 2019. The Global Fire Atlas of individual fire size, duration, speed and direction. Earth System Science Data, 11(2), pp.529-552.*

*Cumming, S.G., 2001. A parametric model of the fire-size distribution. Canadian Journal of Forest Research, 31(8), pp.1297-1303.*

*Hantson, S., Pueyo, S. and Chuvieco, E., 2016. Global fire size distribution: from power law to log-normal. International journal of wildland fire, 25(4), pp.403-412.*

*Reed, W.J. and McKelvey, K.S., 2002. Power-law behaviour and parametric models for the size-distribution of forest fires. Ecological Modelling, 150(3), pp.239-254.*

Section 6: Why is there no accuracy assessment/ validation on the burned area product? Since you are using MODIS and VIIRS, the authors can use the BARD dataset (https://edatos.consorciomadrono.es/dataset.xhtml?persistentId=doi:10.21950/BBQQU7). I don't recommend publishing a new product paper without an adequate burned area validation assessment since that is the primary input into the emissions calculations.

*The GFFEPS system is primarily intended as a near real time forecasting system, building on the Canadian model. To emphasize this, we extended (shown in italics) our motivation statement in the introduction to now say:*

> *"The motivation for this work was the recognized need in extending FireWork's current North American air-quality forecasting to the global domain, thus improving Canadian forecasts by introducing near real time global simulations of smoke emissions external to the original North American domain." [108]*

*Through the historic burned-area per hotspot approach, GFFEPS calculates burned area and emissions in near-real time on a daily basis. Other models such as GFED depend on month-end summaries of burned area, thus providing emissions in an historical context.*

*In Burned Area (4.1), we described our approach of using the historical MCD64A1 data to determine a historic burned area per hotspot, which is then used to predict burned area on a daily basis. This can be summed into global burned-area estimate (essential this is what we are doing in 5.2), yet this would ultimately come back to the original burned-area values from MC64A1.*

*The GFFEPS results are based on MCD64A1 data as its underlying, historical burned area. The MC64A1 burned area product has been assessed/validated in Giglio et al. (2018), which, in turn, was used by GFED4.1s and used to calibrate GFAS1.2; validation and comparative studies have been conducted based on this burned area product (Van der Werf et al., 2017; Pan et al. 2020). This is what we chose as our benchmark for comparing GFFEPS against other models, using common burned area products to allow us the ability to focus our attention on the impact of spatial and temporal variability.*

*Ultimately, MCD64A1 will be replaced by other, newer products. With that in mind, we feel that conducting a formal assessment/validation of various burned area products is best provided by the producers of these new products. Instead, the GFFEPS model, as a design requirement, focuses on providing operational, real-time calculation of fuel consumption, which drives the forecasted emissions predicted in the model. This is the product of fuel load and combustion completeness (FL × CC) of the Seiler and Crutzen equation. This is a component under-evaluated in most global models; where instead, a static value per biome is frequently used.*

**Minor comments:**

Title: Why is the product called a "Forest Fire" product when the authors are mapping burning in all land cover types?

> *The model's title is in keeping with other models in the Canadian Forest Fire Danger Rating System (CFFDRS), including the Canadian Forest Fire Weather Index (FWI) System and the Canadian Forest Fire Behavior Prediction (FBP) System.*

Line 16: change to "showing"

*Done*

Line 64: FRE is the time integral of FRP

*Corrected.  Now sentence reads*

"… measurements of fire radiative energy (FRE, the time integral of fire radiative power (FRP)…" [65]

Lines 74 - 79: The GFED5 Burned Area product has incorporated the GloCAB data (Hall et al., 2024) which provides a cropland burning-specific dataset. https://essd.copernicus.org/articles/16/867/2024/

*Thank you for the recent publication on global cropland burning.*

*GloCAB represents a recently published dataset that is not inline with the methodology using FAO statistics as presented in this manuscript; we've referenced it in our revised manuscript as another methodology of determining cropland burning estimates.  It may be of value in future versions of the GFFEPS model, this would require a specific comparison between the approaches which is beyond the scope of this paper (see summary).*

Line 115: change to "multiplied"

*Corrected*

Section 3.1: The justification for using the GLC2000 is weak, especially now that it is almost 25 years old. Have the authors run a sensitivity analysis with other land cover datasets to see how well the GLC2000 dataset has held up in recent years?

*A sensitivity analysis is now included as Appendix B.1.  It compares GFFEPS emissions for the 2019 using GLC2000 land cover to those using MCD12C1. Although there were regional differences, global results indicated near equality.*

"The scatter plot shows near equality between the two model predictions (a slope of 0.98) when forced through the origin, with an r2 of 0.93. Total annual emissions were 2,957 and 3,028 Mt as predicted by GLC2000 and MCD12C1 respectively."  [1145]

Line 269: "sufficiently complete" is quite a strong statement. VIIRS does not include the morning overpass compared to MODIS so the fire location data is already missing a large number of fires. I would remove the last portion of that sentence. It is also worth mentioning that since you are using the active fire product that you only have the afternoon snapshot of fire pixels as opposed to MODIS which has morning and afternoon.

*The sentence states* "In spite of these limitations, hotspot data from VIIRS provides a picture of global fire activity that is consistent, continuous, and sufficiently complete."

*This was intended as a generalized, self-evident statement.* "Sufficiently complete" *was referring to global coverage. We have rewritten it as follows:*

> "In spite of these limitations, we selected VIIRS data because it is sub-daily, global, readily available, higher resolution than alternative sensors, available in near-real time, and expected to continue well into the 2030s.." [299]

*We acknowledge that, lacking a morning overpass, VIIRS hotspot data is less able to characterize diurnal patterns of fire behavior - though not completely unable, since VIIRS also does nighttime detection (as does MODIS). In spite of this limitation, because of its higher resolution and sensitivity, VIIRS detects more fires, smaller fires, and more burned area than MODIS, and the hotspots have better geolocational accuracy (references below). Also, it is well established that fire behaviour peaks in the afternoon (Countryman 1972) and hence, afternoon detections would more likely and reliable. Finally, GFFEPS has a diurnal pattern to model area growth per time step as documented for CFFEPS in Chen et al. (2019).*

*Schroeder, W., Oliva, P., Giglio, L. & Csiszar, I. 2014. "The New VIIRS 375 m active fire detection data product: Algorithm description and initial assessment." Remote Sensing of the Environment, 143(2014). doi:10.1016/j.rse.2013.12.008*

*Fu, Y.; Li, R.; Wang, X.; Bergeron, Y.; Valeria, O.; Chavardès, R.D.; Wang, Y.; Hu, J. Fire Detection and Fire Radiative Power in Forests and Low-Biomass Lands in Northeast Asia: MODIS versus VIIRS Fire Products. Remote Sens. 2020, 12, 2870. https://doi.org/10.3390/rs12182870*

Figure 6 and Figure 7 captions: It would be helpful for readers unfamiliar with these indices to have a brief description of the meaning of the value in the caption.

*Addressed. Captions now read*

> "Figure 6. Buildup Index (BUI) for September 1, 2019 as interpolated to the 63,566 hotspot locations observed on that date. The BUI, a principal driver in calculating fuel consumption in the FBP system, is calculated using meteorological data from Environment and Climate Change Canada's Global Environmental Multiscale (GEM) model." [323]

> "Figure 7. Growing Season Index (GSI) for September 1, 2019 as interpolated to the 63,566 hotspot locations observed on that date. The GSI provides a method to estimate the greenness of deciduous forests and degree of grass curing, both important factors in fuel consumption. The 21-day average GSI is calculated using meteorological data from Environment and Climate Change Canada's Global Environmental Multiscale (GEM) model." [356]

Line 374: There is a newly released dataset that compiles all the crop-specific emission coefficients from the literature. It is available here: https://doi.org/10.5281/zenodo.7013656

*Thank you for the updated information. The reported values range from 0.89 to 0.92, with a generic crops value of 0.92. This is very close to the value we used (0.90). New crop specific values will be included in future runs.*

Line 387: Determining the other small fires as "inconsequential" is not an adequate justification for not developing a methodology to improve the representation of small fires, especially given the numerous papers showing how many small fires there are on the landscape. I suggest rephrasing.

*Line 387 states* "while assuming small fires in other, non-agricultural, landscapes were deemed inconsequential". *The reviewer has not acknowledged that the FAO approach we presented in subchapter 3.4 deals specifically with agricultural fires and presents a methodology that captures the emissions from large and small in agricultural zones. This is further discussed in the new Appendix B.2. This was also answered earlier in response to the reviewer's questions regarding lines 569 and 579.*

*As for non-agricultural fires, our experience in Canada, as referenced on the Canadian Wildland Fire Information System is that* "Fires of all sizes are included in the database, but only those greater than 200 hectares in final size are shown in the map above — these represent a small percentage of all fires but account for most of the area burned (usually more than 97%)." *To support this, we added the following text*

"The approach does assume small fires in other, non-agricultural, landscapes are inconsequential, which we see as acceptable. This is certainly the case in Canada, where the National Forestry Database (https://cwfis.cfs.nrcan.gc.ca/ha/nfdb, last accessed 2024-05-28) indicates that between 1980-2021, fires less than 1 ha, which constituted 73% of fires, account for only 0.03% of the burned area nationally; that fires less than 10 ha, which account constituted 87% of fires, account for only 0.18% of the burned area." [440]

Line 427: Why not just remove the persistent sources?

*In some cases it is difficult to determine which persistent sources are fires. Slow-burning fires that burn within the same pixel for more than one day, or areas that are frequently burned for agricultural purposes, could be considered "persistent"; e.g. agricultural burning may take place through transporting the material to be burned to a common site. However, the suggestion is a good one, as there are many locations that are clearly not fires that could have been removed. However, because our method minimizes the emissions from persistent sources, the impact on smoke emissions results would be insignificant.*

*Summary*

*We disagree with the review's thesis is that we are premature in publishing the GFFEPS model; that there is newer data that should be incorporated (NDVI; GFED 5; Global Cropland Burned Area); and that several comparative studies need to be conducted (GSI vs NVDI; Global Cropland Burned Area vs FAO data; GFFEPS versus BARD; GLC2000 vs MODIS landcover datasets). We note that the intended use of GFFEPS in the context of real-time forecasting was not clear to the reviewer, that this purpose governs many of the choices made regarding model inputs, and we have modified the manuscript accordingly to clarify this constraint.*

*We added a sensitivity analysis as Appendix B, where we carried out comparisons using of MODIS land use types versus GLC2000, using a default crop burning value and burned area per hotspot as opposed to the FAO agricultural burning, using fuel consumption based on fixed combustion completeness for grass, forest and peatland versus fuel consumption based on daily weather and fire behaviour. Other requests such as comparison of the GSI vs NVDI have taken place in the literature – and we note that in the context of real-time forecasting, the former represents a much smaller processing time requirement. As new data become available, they may be incorporated into GFFEPS, provided that they may be used in an operational forecasting environment, where minimizing processing time while generating emissions is a key requirement.*

*Our position is that we have reached a clear benchmark and feel that the methodology and our results based on common published datasets is worthy of publication. The model is compared to other models (GFAS1.2, GFED4.1s, FINN1.5/2.5), using results that have been published for each. Many of these models (e.g., GFED4.1s) use older burned area products (MC64A1) and it is only fitting that model comparisons should be conducted with similar input. We also note that historical burned area data must be used in the context of real-time forecasting.*

*This manuscript presents the initial version of GFFEPS. The model will continue to evolve with new methodologies and improved underlying data.*